# Including the efficacy of land ice changes in deriving climate sensitivity from paleodata

Lennert B. Stap[1], Peter Köhler[1], and Gerrit Lohmann[1]

[1]Alfred-Wegener-Institut, Helmholtz-Zentrum für Polar- und Meeresforschung, Am Handelshafen 12, 27570 Bremerhaven, Germany

**Correspondence:** L.B. Stap (lennert.stap@awi.de)

**Abstract.** The equilibrium climate sensitivity (ECS) of climate models is calculated as the equilibrium global mean surface air warming resulting from a simulated doubling of the atmospheric $CO_2$ concentration. In these simulations, long-term processes in the climate system, such as land ice changes, are not incorporated. Hence, climate sensitivity derived from paleodata has to be compensated for these processes, when comparing it to the ECS of climate models. Several recent studies found that the impact these long-term processes have on global temperature cannot be quantified directly through the global radiative forcing they induce. This renders the prevailing approach of deconvoluting paleotemperatures through a partitioning based on radiative forcings inaccurate. Here, we therefore implement an efficacy factor $\varepsilon_{[LI]}$, that relates the impact of land ice changes on global temperature to that of $CO_2$ changes, in our calculation of climate sensitivity from paleodata. We apply our refined approach to a proxy-inferred paleoclimate dataset, using $\varepsilon_{[LI]} = 0.45^{+0.34}_{-0.20}$ based on a multi-model assemblage of simulated relative influences of land ice changes on the Last Glacial Maximum temperature anomaly. The implemented $\varepsilon_{[LI]}$ is smaller than unity, meaning that per unit of radiative forcing the impact on global temperature is less strong for land ice changes than for $CO_2$ changes. Consequently, our obtained ECS estimate of $5.8 \pm 1.3\,\mathrm{K}$, where the uncertainty reflects the implemented range in $\varepsilon_{[LI]}$, is ~50% higher than when differences in efficacy are not considered.

# 1 Introduction

Equilibrium climate sensitivity (ECS) expresses the simulated equilibrated surface air temperature response to an instantaneous doubling of the atmospheric $CO_2$ concentration. The simulated effect of the applied $CO_2$ radiative forcing anomaly includes the Planck response, as well as the fast feedbacks such as those involving changes to snow, sea ice, lapse rate, clouds and water vapour. ECS varies significantly between different state-of-the-art climate models, for instance the CMIP5 ensemble shows a range of 1.9 to 4.4 K (Vial et al., 2013). Several ways have been put forward to constrain ECS, for example through the usage of paleoclimate data (e.g. Covey et al., 1996; Edwards et al., 2007), which is also the focus of this study. However, unlike results of models, temperature reconstructions based on paleoclimate proxy data always contain a mixed signal of all processes active in the climate system. Among these are long-term processes (or slow feedbacks) such as changes in vegetation, dust, and, arguably most importantly, land ice changes, which are kept constant in the climate model runs used to calculate ECS. Therefore, it is necessary to correct paleotemperature records for the influence of these processes, in order to make a meaningful comparison to ECS calculated by climate models.

In a co-ordinated community effort, the PALAEOSENS project proposed to relate the temperature response caused by these long-term processes to the globally averaged radiative forcing they induce (PALAEOSENS Project Members, 2012). Consequently, the paleotemperature record can be disentangled on the basis of the separate radiative forcings of these long-term processes (e.g. von der Heydt et al., 2014; Martínez-Botí et al., 2015; Köhler et al., 2015, 2017b, 2018; Friedrich et al., 2016). If all processes are accounted for in this manner, the effect of $CO_2$ changes and the accompanying short-term feedbacks, as described by the ECS, can be estimated. However, several studies have shown that, depending on the type of radiative forcing, the same global average radiative forcing can lead to different global temperature changes (e.g. Stuber et al., 2005; Hansen et al., 2005; Yoshimori et al., 2011). For instance, in a previous article (Stap et al., 2018) we simulated the separate and combined effects of $CO_2$ changes and land ice changes on global surface air temperature using the intermediate complexity climate model CLIMBER-2 and showed that the specific global temperature change per unit radiative forcing change depends on which process is involved. As a possible solution to this problem, Hansen et al. (2005) formulated the concept of 'efficacy' factors, which express the impact of radiative forcing by a certain process in comparison to the effect of radiative forcing by $CO_2$ changes.

Based on the concept of Hansen et al. (2005), here we introduce an efficacy factor for radiative forcing by albedo changes due to land ice variability, in our method of deriving climate sensitivity from paleodata. We first illustrate our refined approach by applying it to transient simulations over the past 5 Myr using CLIMBER-2 (Stap et al., 2018), obtaining a quantification of the effect on global temperature of $CO_2$ changes and the accompanying short-term feedbacks from a simulation forced by both land ice and $CO_2$ changes. We compare this result to a simulation where $CO_2$ changes are the only operating long-term process. In this manner, we can assess the error resulting from using a constant efficacy factor. Thereafter, we refine a previous estimate of climate sensitivity based on a paleoclimate dataset of the past 800 kyr (Köhler et al., 2015, 2018). In this dataset, the sole effect of $CO_2$ is not a-priori known. We therefore investigate the influence of the introduced efficacy factor on the calculated climate sensitivity. To do so, we appraise the influence of land ice changes and the associated efficacy using a range

that is given by different modelling efforts of the Last Glacial Maximum (LGM; ∼21 kyr ago) (Shakun, 2017). The climate sensitivity resulting from applying this range provides a quantification of the consequence of the uncertain efficacy of land ice changes.

## 2 Material and methods

In this section, we first summarize the approach to obtaining climate sensitivity from paleodata, that has been used in numerous earlier studies (e.g. PALAEOSENS Project Members, 2012; von der Heydt et al., 2014; Martínez-Botí et al., 2015; Köhler et al., 2015, 2017b, 2018; Friedrich et al., 2016). We then discuss our main refinement to that approach, which is the inclusion of the efficacy of land ice changes, and a further small refinement that is meant to unify the dependent variable in cross-plots of
10 radiative forcing and global temperature anomalies.

### 2.1 Approach to obtain climate sensitivity from paleodata

In climate model simulations used to quantify ECS, fast feedbacks, i.e. processes in the climate system with timescales of less than ∼100 yrs, are accounted for. However, slower processes, such as those involving changes to ice sheets, vegetation and dust, are commonly kept constant. The resulting response is also sometimes called 'Charney' sensitivity (Charney et al.,
1979). Following the notation of PALAEOSENS Project Members (2012), the ratio of the temperature change ($\Delta T_{[CO_2]}$) to the radiative forcing due to the $CO_2$ change ($\Delta R_{[CO_2]}$), yields $S^a$ (in $\mathrm{K\,W^{-1}\,m^2}$, and where $a$ stands for *actuo*):

$$S^a = \frac{\Delta T_{[CO_2]}}{\Delta R_{[CO_2]}}. \tag{1}$$

The subscript denotes that $CO_2$ is the only long-term process involved. Analogously, paleoclimate sensitivity ($S^p$) can be deduced from paleo-temperature reconstructions and paleo-$CO_2$ records as

$$S^p = \frac{\Delta T_g}{\Delta R_{[CO_2]}}. \tag{2}$$

In this case, the average global paleotemperature anomaly with respect to the pre-industrial (PI) ($\Delta T_g$) is, however, also affected
by the long-term processes that are typically neglected in climate simulations. Therefore, a correction to the paleotemperature record is needed to obtain $\Delta T_{[CO_2]}$ from $\Delta T_g$:

$$\Delta T_{[CO_2]} = \Delta T_g (1 - f), \tag{3}$$

or equivalently $S^a$ from $S^p$:

$$S^a = S^p (1 - f) = \frac{\Delta T_g}{\Delta R_{[CO_2]}} (1 - f). \tag{4}$$

Here, $f$ represents the effect of the slow feedbacks on paleotemperature (e.g. van de Wal et al., 2011). To obtain $f$, PALAEOSENS Project Members (2012) proposed an approach, which has subsequently been used in numerous studies aiming to constrain

climate sensitivity from paleodata (e.g. von der Heydt et al., 2014; Martínez-Botí et al., 2015; Köhler et al., 2015, 2017b, 2018; Friedrich et al., 2016), and paleoclimate modelling studies (e.g. PALAEOSENS Project Members, 2012; Friedrich et al., 2016; Chandan and Peltier, 2018). They suggested to quantify the influence of the long-term processes (X) by the radiative forcing change they induce ($\Delta R_{[X]}$), relative to the total radiative forcing perturbation:

$$f = \frac{\Delta R_{[X]}}{\Delta R_{[CO_2]} + \Delta R_{[X]}} = 1 - \frac{\Delta R_{[CO_2]}}{\Delta R_{[CO_2]} + \Delta R_{[X]}} \tag{5}$$

Combining Eqs. 4 and 5 and following the PALAEOSENS nomenclature, we can then derive the 'specific' paleoclimate sensitivity $S_{[CO_2,X]}$, where X represents the processes that are accounted for in the calculation of $f$:

$$S_{[CO_2,X]} = \frac{\Delta T_g}{\Delta R_{[CO_2]}}(1 - \frac{\Delta R_{[X]}}{\Delta R_{[CO_2]} + \Delta R_{[X]}}) = \frac{\Delta T_g}{\Delta R_{[CO_2]} + \Delta R_{[X]}} = \frac{\Delta T_g}{\Delta R_{[CO_2,X]}}. \tag{6}$$

If, for instance, only the most important slow feedback in the climate system, namely radiative forcing anomalies induced by albedo changes due to land ice (LI) variability is taken into account, then one can correct $S^p$ to derive the following specific climate sensitivity:

$$S_{[CO_2,LI]} = \frac{\Delta T_g}{\Delta R_{[CO_2]} + \Delta R_{[LI]}} = \frac{\Delta T_g}{\Delta R_{[CO_2,LI]}}. \tag{7}$$

Using this approach, several studies performed a least-squares regression through scattered data from paleotemperature and radiative forcing records (Martínez-Botí et al., 2015; Friedrich et al., 2016; Köhler et al., 2015, 2017b, 2018) relating $\Delta T_g$ to $\Delta R_{[CO_2,LI]}$ in a time-independent manner, from which $S_{[CO_2,LI]}$ could be determined. In the course of those studies, a state dependency of $S_{[CO_2,LI]}$ as function of background climate has been deduced for those data which are best approximated by a non-linear function. Furthermore, the quantification of $S_{[CO_2,LI]}$ for those state-dependent cases has been formalized in Köhler et al. (2017b). A synthesis of estimates of $S_{[CO_2,LI]}$ from both colder- and warmer-than-present climates has been compiled by von der Heydt et al. (2016).

## 2.2 Refinement 1: Taking the efficacy of land ice changes into account

The validity of the PALAEOSENS approach to calculate $f$ is contingent on the notion that identical global-average radiative forcing changes lead to identical global temperature responses, regardless of the processes involved. However, it has been demonstrated that the horizontal and vertical distribution of the radiative forcing affects the resulting temperature response (e.g. Stuber et al., 2005; Hansen et al., 2005; Yoshimori et al., 2011; Stap et al., 2018) because, e.g. different fast feedbacks are triggered depending on the location of the forcing. To address this issue, Hansen et al. (2005) introduced the concept of 'efficacy' factors, which we will explore further in this study. These factors ($\varepsilon_{[X]}$) relate the strength of the temperature response to radiative forcing caused by a certain process X ($\Delta T_{[X]}/\Delta R_{[X]}$), to a similar ratio caused by $CO_2$ radiative forcing ($\Delta T_{[CO_2]}/\Delta R_{[CO_2]}$). This introduction of efficacy requires a reformulation of $f$ as $f_\varepsilon$:

$$f_\varepsilon = \frac{\varepsilon_{[X]}\Delta R_{[X]}}{\Delta R_{[CO_2]} + \varepsilon_{[X]}\Delta R_{[X]}} = 1 - \frac{\Delta R_{[CO_2]}}{\Delta R_{[CO_2]} + \varepsilon_{[X]}\Delta R_{[X]}}, \tag{8}$$

and hence also of $S_{[CO_2,X]}$ as $S^\varepsilon_{[CO_2,X]}$:

$$S^\varepsilon_{[CO_2,X]} = \frac{\Delta T_g}{\Delta R_{[CO_2]} + \varepsilon_{[X]}\Delta R_{[X]}}. \quad (9)$$

In these reformulations, where in principle $\varepsilon_{[X]}$ can take any value, we introduce the superscript $\varepsilon$. This serves to clearly
distinguish these newly-derived sensitivities from those of the PALAEOSENS approach in which efficacy was not taken into
account, implying that identical radiative forcing of different processes leads to identical temperature changes.

To calculate $S^\varepsilon_{[CO_2,LI]}$, we constrain the efficacy factor for radiative forcing by land ice changes ($\varepsilon_{[LI]}$), using the following
formulation, which is based on, but slightly modified from Hansen et al. (2005):

$$\frac{\Delta T_{[LI]}}{\Delta R_{[LI]}} = \varepsilon_{[LI]} \frac{\Delta T_g - \Delta T_{[LI]}}{\Delta R_{[CO_2]}}. \quad (10)$$

This leads to:

$$\varepsilon_{[LI]} = \frac{\omega}{1-\omega} \frac{\Delta R_{[CO_2]}}{\Delta R_{[LI]}}, \quad (11)$$

where $\omega$ represents the fractional relative influence of land ice changes on the global temperature change ($\omega = \Delta T_{[LI]}/\Delta T_g$). If
$\varepsilon_{[LI]}$ is assumed to be constant in time (see Sect. 3.2), it can be calculated using Eq. 11 from data of any time step in the record
of $\Delta R_{[CO_2]}$ and $\Delta R_{[LI]}$, and consequently applied to the whole record (Fig. 1a,c). As before, with this $\varepsilon_{[LI]}$ a quantification of
$S^\varepsilon_{[CO_2,LI]}$ can be obtained by performing a least-squares regression through scattered data from paleotemperature and radiative
forcing records, now relating $\Delta T_g$ to ($\Delta R_{[CO_2]} + \varepsilon_{[LI]}\Delta R_{[LI]}$) in a time-independent manner.

Note that apart from the formulation based on Hansen et al. (2005) followed here, other formulations of the efficacy factor
are possible. For instance, one can define an alternative efficacy factor ($\varepsilon_{[LI],alt}$) such that it relates the effect of land ice changes
on global temperature directly to the radiative forcing anomaly caused by $CO_2$ changes, leading to:

$$S^\varepsilon_{[CO_2,X],alt} = \frac{\Delta T_g}{\Delta R_{[CO_2]} + \varepsilon_{[LI],alt}\Delta R_{[CO_2]}}. \quad (12)$$

In this alternative case, the efficacy factor $\varepsilon_{[LI],alt}$ relates to our original $\varepsilon_{[LI]}$ as:

$$\varepsilon_{[LI],alt} = \varepsilon_{[LI]} \frac{\Delta R_{[LI]}}{\Delta R_{[CO_2]}}. \quad (13)$$

This implies that if $\varepsilon_{[LI]}$ is indeed constant, any non-linearity in the relation between $\Delta R_{[CO_2]}$ and $\Delta R_{[LI]}$ would demand a
more complex formulation of the alternative efficacy factor $\varepsilon_{[LI],alt}$ (e.g. via a higher-order polynomial). Since we find such
a non-linearity in our data (Fig. 2), using an F test to determine that a second order polynomial is a significantly (p value <
0.0001) better fit to the data than a linear function, we refrain from following this alternative formulation further.

## 2.3 Refinement 2: Unifying the dependent variable

To calculate $S^\varepsilon_{[CO_2,LI]}$, previous studies have used cross-plots of global temperature anomalies and radiative forcing. The
latter is caused by a combination of $CO_2$ and land-ice changes, which is cumbersome if one wants to compare $S^\varepsilon_{[CO_2,LI]}$ to

other specific paleoclimate sensitivities $S^{\varepsilon}_{[\mathrm{CO_2,X}]}$, where more and/or different long-term processes are considered. Here, we therefore reformulate our quantification of $S^{\varepsilon}_{[\mathrm{CO_2,LI}]}$ to unify the dependent variable as $\Delta R_{[\mathrm{CO_2}]}$.

$$S^{\varepsilon}_{[\mathrm{CO_2,X}]} = \frac{\Delta T_{\mathrm{g}}}{\Delta R_{[\mathrm{CO_2}]} + \varepsilon_{[\mathrm{X}]}\Delta R_{[\mathrm{X}]}} = \frac{\Delta T_{\mathrm{g}}}{\Delta R_{[\mathrm{CO_2}]}}\frac{\Delta R_{[\mathrm{CO_2}]}}{\Delta R_{[\mathrm{CO_2}]} + \varepsilon_{[\mathrm{X}]}\Delta R_{[\mathrm{X}]}} = \frac{\Delta T^{\varepsilon}_{[-\mathrm{X}]}}{\Delta R_{[\mathrm{CO_2}]}}. \tag{14}$$

Here, $\Delta T^{\varepsilon}_{[-\mathrm{X}]}$ is the global temperature change (with respect to PI) stripped of the inferred influence of processes X, defined as:

$$\Delta T^{\varepsilon}_{[-\mathrm{X}]} := \Delta T_{\mathrm{g}}\frac{\Delta R_{[\mathrm{CO_2}]}}{\Delta R_{[\mathrm{CO_2}]} + \varepsilon_{[\mathrm{X}]}\Delta R_{[\mathrm{X}]}}. \tag{15}$$

Hence, for the calculation of $S^{\varepsilon}_{[\mathrm{CO_2,LI}]}$ we use:

$$\Delta T^{\varepsilon}_{[-\mathrm{LI}]} := \Delta T_{\mathrm{g}}\frac{\Delta R_{[\mathrm{CO_2}]}}{\Delta R_{[\mathrm{CO_2}]} + \varepsilon_{[\mathrm{LI}]}\Delta R_{[\mathrm{LI}]}}. \tag{16}$$

Now, we quantify $S^{\varepsilon}_{[\mathrm{CO_2,LI}]}$ by performing a least-squares regression (regfunc) through scattered data from $\Delta T^{\varepsilon}_{[-\mathrm{LI}]}$ and $\Delta R_{[\mathrm{CO_2}]}$. We use the precondition that no change in $CO_2$ is related to no change in $\Delta T^{\varepsilon}_{[-\mathrm{LI}]}$, meaning the regression intersects

the y-axis at the origin $((x,y) = (0,0))$. Following Köhler et al. (2017b), for any non-zero $\Delta R_{[\mathrm{CO_2}]}$, we calculate $S^{\varepsilon}_{[\mathrm{CO_2,LI}]}$ as:

$$S^{\varepsilon}_{[\mathrm{CO_2,LI}]}\bigg|_{\Delta R_{[\mathrm{CO_2}]}} = \frac{\mathrm{regfunc}}{\Delta R_{[\mathrm{CO_2}]}}\bigg|_{\Delta R_{[\mathrm{CO_2}]}}. \tag{17}$$

If $\Delta R_{[\mathrm{CO_2}]} = 0\,\mathrm{W\,m^{-2}}$, as is among others the case for pre-industrial conditions, $S^{\varepsilon}_{[\mathrm{CO_2,LI}]}$ is quantified as:

$$S^{\varepsilon}_{[\mathrm{CO_2,LI}]}\bigg|_{\Delta R_{[\mathrm{CO_2}]}=0} = \frac{\delta(\mathrm{regfunc})}{\delta(\Delta R_{[\mathrm{CO_2}]})}\bigg|_{\Delta R_{[\mathrm{CO_2}]}=0}. \tag{18}$$

Equations 17 and 18 yield a quantification of $S^{\varepsilon}_{[\mathrm{CO_2,LI}]}$, which can be compared to the value obtained for $S^{\varepsilon}_{[\mathrm{CO_2,LI}]}$ using the PALAEOSENS approach that does not consider efficacy differences (equivalent to using $\varepsilon_{[\mathrm{LI}]} = 1$) (Köhler et al., 2018).

To obtain $S^a$, one needs to multiply $S_{[\mathrm{CO_2,LI}]}$ by a conversion factor $\phi = 0.64 \pm 0.07$ ($1\sigma$-uncertainty) that accounts for the influence of other long-term processes, namely vegetation, aerosol and non-$CO_2$ greenhouse gas changes (PALAEOSENS Project Members, 2012). Note that this multiplication by $\phi$ ignores any possible state-dependencies in $\phi$ and assumes unit efficacy for processes other than land ice changes. Because a comprehensive analysis of the efficacy and state-dependency of these other processes is beyond the scope of this study, it is a source of uncertainty to be investigated in future research. Finally,

we obtain the equivalent ECS by multiplying $S^a$ by $\Delta R_{\mathrm{2xCO_2}} = 3.71 \pm 0.37\,\mathrm{W\,m^{-2}}$ ($1\sigma$-uncertainty), the radiative forcing perturbation representing a $CO_2$ doubling (Myhre et al., 1998).

## 3   Illustration of the approach using model simulations

In this section, we illustrate our refined approach, which considers efficacy differences, by applying it to transient simulations over the past 5 Myr using CLIMBER-2 (Stap et al., 2018). We obtain a quantification of the effect on global temperature of

$CO_2$ changes and the accompanying short-term feedbacks from a simulation forced by both land ice and $CO_2$ changes. We compare this result to a simulation where $CO_2$ changes are the only operating long-term process. By doing so, we assess the error resulting from using a constant efficacy factor.

### 3.1 CLIMBER-2 model simulations

Currently, long ($\sim 10^5$ to $\sim 10^6$ years) integrations of state-of-the-art climate models, such as general circulation models and Earth system models, are not yet not feasible due to limited computer power. This gap can be filled by using models of reduced complexity (Claussen et al., 2002; Stap et al., 2017). Using the intermediate complexity climate model CLIMBER-2
(Petoukhov et al., 2000; Ganopolski et al., 2001), climate simulations over the past 5 Myr were performed and analysed in Stap et al. (2018). CLIMBER-2 combines a 2.5-dimensional statistical-dynamical atmosphere model, with a 3-basin zonally averaged ocean model (Stocker et al., 1992), and a model that calculates dynamic vegetation cover based on the temperature and precipitation (Brovkin et al., 1997). The simulations could be forced by solar insolation changes due to orbital (O) variations (Laskar et al., 2004), by land ice (I) changes on both hemispheres (based on de Boer et al., 2013), and by $CO_2$ (C) changes
(based on van de Wal et al., 2011). In the reference experiment (OIC) all these factors are varied, while in other model integrations the land ice (experiment OC) or the $CO_2$ concentration (experiment OI) is kept fixed at PI level. The synergy of land ice and $CO_2$ changes is negligibly small, meaning their induced temperature changes add approximately linearly when both forcings are applied. Furthermore, the influence of orbital variations is also very small, so that experiment OC approximately yields the sole effect of $CO_2$ changes on global temperature ($\Delta T_{[OC]}$). As in Stap et al. (2018), we use the
simple energy balance model of Köhler et al. (2010) to analyse the applied radiative forcing of land ice albedo and $CO_2$ changes and simulated global temperature changes, after averaging to 1,000 year temporal resolution (Fig. 1a,b).

### 3.2 Analysis

First, we analyse experiment OC, which will serve as a target for our approach as deployed later in this section. We use a least-squares regression through scattered data of $\Delta R_{[CO_2]}$ and $\Delta T_{[OC]}$ to fit a second order polynomial (Fig. 3a). Using a
higher order polynomial rather than a linear function allows us to capture state dependency of paleoclimate sensitivity. Fitting even higher order polynomials leads to negligible coefficients for the higher powers, and is not pursued further. From the fit, we calculate a specific paleoclimate sensitivity $S^{\varepsilon}_{[CO_2,LI]}$ of 0.74 $K\,W^{-1}\,m^2$ for PI conditions ($\Delta R_{[CO_2]} = 0$ $W\,m^{-2}$) using Eq. 18. Note that, in this case, $S^{\varepsilon}_{[CO_2,LI]}$ is equal to $S^{\varepsilon}_{[CO_2]}$, $S_{[CO_2,LI]}$ and $S_{[CO_2]}$ as there are no land ice changes and therefore also no efficacy differences. The fit further shows decreasing $S^{\varepsilon}_{[CO_2,LI]}$ for rising $\Delta R_{[CO_2]}$.
Now, we apply our approach to the results of experiment OIC, in which both $CO_2$ and land ice cover vary over time, with the aim of deducing the sole effect of $CO_2$ changes on global temperature. We calculate the efficacy of land ice changes for the Last Glacial Maximum (21 kyr ago; LGM) from experiment OI, in which the $CO_2$ concentration is kept constant. We obtain $\omega = \Delta T_{[LI]}/\Delta T_g = \Delta T_{[OI]}/\Delta T_{[OIC]} = 0.54$. Consequently, we find $\varepsilon_{[LI]} = 0.58$ from Eq. 11, and apply this value to the whole record of $\Delta R_{[CO_2]}$ and $\Delta R_{[LI]}$. In this manner, we calculate $\Delta T^{\varepsilon}_{[-LI]}$ using Eq. 16. In principle, $\varepsilon_{[LI]}$ can be obtained using
data from any time step of the record, preferably when the radiative forcing anomalies are large to prevent outliers resulting

from divisions by small numbers. For example, using the results from all glacial marine isotope stages of the past 810 kyr (MIS 2, 6, 8, 10, 12, 14, 16, 18, and 20), instead of only the LGM, leads to a mean ($\pm 1\sigma$) $\varepsilon_{[LI]}$ of $0.56 \pm 0.09$.

We then fit a second order polynomial to the scattered data of the thusly obtained $\Delta T^{\varepsilon}_{[-LI]}$ from the results of experiment OIC, and $\Delta R_{[CO_2]}$ (Fig. 3b,c). Between $\Delta R_{[CO_2]} = -0.5\,\mathrm{W\,m^{-2}}$ and $\Delta R_{[CO_2]} = 0.5\,\mathrm{W\,m^{-2}}$, outliers resulted from division by small numbers (not shown in Fig. 3b). To remove these outliers, we first calculate the root mean square error (RMSE) between the fit and the data in the remainder of the domain. We then exclude all values from the range $\Delta R_{[CO_2]} = -0.5\,\mathrm{W\,m^{-2}}$ to $\Delta R_{[CO_2]} = 0.5\,\mathrm{W\,m^{-2}}$ where the fit differs from the data by more than $3 \times$ RMSE, and perform the regression again. This yields an $S^{\varepsilon}_{[CO_2,LI]}$ of $0.72\,\mathrm{K\,W^{-1}\,m^2}$ for PI (Fig. 3b) in the LGM-only case, and $0.73^{+0.06}_{-0.05}\,\mathrm{K\,W^{-1}\,m^2}$ in the case where all glacial periods are used (Fig. 3c). This supports our approach since it is only slightly lower than the $S^{\varepsilon}_{[CO_2,LI]}$ of $0.74\,\mathrm{K\,W^{-1}\,m^2}$ obtained from experiment OC, which it should approximate. The relationship between $\Delta T^{\varepsilon}_{[-LI]}$ and $\Delta R_{[CO_2]}$ (Fig. 3b) is less linear than that between $\Delta T_{[OC]}$ and $\Delta R_{[CO_2]}$ (Fig. 3a), hence the state dependency of $S^{\varepsilon}_{[CO_2,LI]}$ is enhanced. However, the difference between the $S^{\varepsilon}_{[CO_2,LI]}$ obtained from both experiments remains smaller than $0.07\,\mathrm{K\,W^{-1}\,m^2}$ through the entire 5-Myr interval in the LGM-only case, indicating that a constant efficacy is an acceptable assumption which only introduces a negligible additional uncertainty. However, the possible time-dependency of efficacy could be investigated more rigorously in future research using more sophisticated climate models.

The PALAEOSENS approach that does not consider efficacy differences ($\varepsilon_{[LI]} = 1$) yields a PI $S_{[CO_2,LI]}$ of $0.54\,\mathrm{K\,W^{-1}\,m^2}$ (Fig. 3d). This is clearly much more off-target than the results of our approach, signifying the importance of considering efficacy.

## 4   Application to proxy-inferred paleoclimate data

In this section, we compare our refined approach to calculate $S^{\varepsilon}_{[CO_2,LI]}$ incorporating efficacy, to our previous quantification of $S_{[CO_2,LI]}$ (Köhler et al., 2018), by reanalysing same paleoclimate dataset (introduced in Köhler et al., 2015). Other than for climate model simulations, in proxy-based datasets the influence of land ice changes on global temperature perturbations cannot be directly obtained, and is hence a-priori unknown. We therefore base the value of $\varepsilon_{[LI]}$ we implement here on a multi-model assemblage of simulated relative influences of land ice changes on the Last Glacial Maximum (LGM) temperature anomaly (Shakun, 2017).

### 4.1   Proxy-inferred paleoclimate dataset

The dataset to be investigated contains reconstructions of $\Delta T_g$, $\Delta R_{[CO_2]}$, and $\Delta R_{[LI]}$ for the past 800 kyr. Although the dataset covers the past 5 Myr, here we focus only on the past 800 kyr (Fig. 1c,d) because over this period $\Delta R_{[CO_2]}$ is constrained by high-fidelity measurements of $CO_2$ within ice cores, whereas Pliocene and Early Pleistocene $CO_2$ levels are still heavily debated (e.g. Badger et al., 2013; Martínez-Botí et al., 2015; Willeit et al., 2015; Stap et al., 2016, 2017; Chalk et al., 2017; Dyez et al., 2018). Radiative forcing by $CO_2$ is obtained from Antarctic ice core data compiled by Bereiter et al. (2015), using $\Delta R_{[CO_2]} = 5.35\,\mathrm{W\,m^{-2}} \cdot \ln(CO_2/(278\,\mathrm{ppm}))$ (Myhre et al., 1998). The revised formulation for $\Delta R_{[CO_2]}$ from Etminan et al.

(2016) leads to very similar results with less than $0.01\,\mathrm{W\,m^{-2}}$ differences between the approaches for typical late Pleistocene

30   $CO_2$ values (Köhler et al., 2017a). Radiative forcing caused by land ice albedo changes, as well as the global surface air temperature record ($\Delta T_g$), are based on results of the 3D ice-sheet model ANICE (de Boer et al., 2014) forced by northern hemispheric temperature anomalies with respect to a reference PI climate. The ANICE results are here considered to be proxy-inferred, because, unlike climate models, ANICE is not constrained by climatic boundary conditions such as insolation and greenhouse gases. The temperature anomalies follow directly from a benthic $\delta^{18}O$ stack (Lisiecki and Raymo, 2005) using an inverse technique. Nevertheless, the results are model-dependent and therefore subject to uncertainty. ANICE provided

geographically specific land ice distributions, and hence radiative forcing due to albedo changes with respect to PI on both hemispheres. In Köhler et al. (2015), the northern hemispheric (NH) temperature anomalies ($\Delta T_{\mathrm{NH}}$) are translated into global temperature perturbations ($\Delta T_{g1}$ in Köhler et al. (2015)) using polar amplification factors ($f_{\mathrm{PA}} = \Delta T_{\mathrm{NH}}/\Delta T_g$) as follows: at the LGM, $f_{\mathrm{PA}} = 2.7$ is taken from the average of PMIP3 model data (Braconnot et al., 2012), while at the mid-Pliocene Warm Period (mPWP, about 3.2 Myr ago), $f_{\mathrm{PA}} = 1.6$ is calculated from the average of PlioMIP results (Haywood et al., 2013). At

all other times, $f_{\mathrm{PA}}$ is linearly varied as a function of the NH temperature. In Appendix A, we investigate the influence of the chosen polar amplification factor on our results. The temporal resolution of the dataset is 2,000 years.

    Analysing this dataset, Köhler et al. (2018) found a temperature-$CO_2$ divergence appearing mainly during, or in connection with, periods of decreasing obliquity related to land ice growth or sea level fall. For these periods, a significantly different $S_{[\mathrm{CO2,LI}]}$ was obtained than for the remainder of the time frame. However, in the future we expect sea level to rise, hence these

intervals of strong temperature-$CO_2$ divergence should not be considered for the interpretation of paleodata in the context of future warming, e.g. by using paleodata to constrain ECS. In the following analysis, we therefore exclude these times with strong temperature-$CO_2$ divergence, leaving 217 data points as indicated in Fig. 1c,d.

## 4.2 Analysis

Shakun (2017) compiled model-based estimates of the relative impact of land ice changes on the LGM temperature anomaly

($\omega$ in Eq. 11) using an ensemble of 12 climate models, and estimated $\omega$ to be $0.46 \pm 0.14$ (mean $\pm 1\sigma$, full range $0.20 - 0.68$). Applying these values, in combination with the LGM values (taken here as the mean of the data at 20 and 22 kyr ago) $\Delta R_{[\mathrm{CO2}]} = -2.04\,\mathrm{W\,m^{-2}}$ and $\Delta R_{[\mathrm{LI}]} = -3.88\,\mathrm{W\,m^{-2}}$, yields $\varepsilon_{[\mathrm{LI}]} = 0.45^{+0.34}_{-0.20}$. Implementing this range for $\varepsilon_{[\mathrm{LI}]}$ in Eq. 16, we calculate $\Delta T^{\varepsilon}_{[-\mathrm{LI}]}$ over the whole 800-kyr period. Fitting second order polynomials by least-squares regression to the scattered data of $\Delta T^{\varepsilon}_{[-\mathrm{LI}]}$ and $\Delta R_{[\mathrm{CO2}]}$, we infer a PI $S^{\varepsilon}_{[\mathrm{CO2,LI}]}$ of $2.45^{+0.53}_{-0.56}\,\mathrm{K\,W^{-1}\,m^2}$ (Fig. 4a). The substantial uncertainty

given here only reflects the $1\sigma$ uncertainty in $\varepsilon_{[\mathrm{LI}]}$. Similar to Köhler et al. (2018), we also detect a state dependency with decreasing $S^{\varepsilon}_{[\mathrm{CO2,LI}]}$ towards colder climates for this dataset, more strongly so in case of lower $\varepsilon_{[\mathrm{LI}]}$. This state dependency is opposite to the one found in the CLIMBER-2 results (Sect. 3). The difference may be related either to the fact that fast climate feedbacks are too linear, or that some slow feedbacks are underestimated in intermediate complexity climate models like CLIMBER-2 (see Köhler et al., 2018, for a detailed discussion). At $\Delta R_{[\mathrm{CO2}]} = -2.04\,\mathrm{W\,m^{-2}}$, the LGM value, $S^{\varepsilon}_{[\mathrm{CO2,LI}]}$

is only $1.45^{+0.33}_{-0.37}\,\mathrm{K\,W^{-1}\,m^2}$. The PALAEOSENS approach, which does not consider efficacy and is therefore equivalent to our approach using $\varepsilon_{[\mathrm{LI}]} = 1$, yields $S_{[\mathrm{CO2,LI}]} = 1.66\,\mathrm{K\,W^{-1}\,m^2}$ for PI, and $S_{[\mathrm{CO2,LI}]} = 0.93\,\mathrm{K\,W^{-1}\,m^2}$ for the LGM (Fig. 4b).

The specific paleoclimate sensitivities we find using the refined approach are hence generally larger than those obtained when neglecting efficacy differences. This is because, for the range of the impact of land ice changes on the LGM temperature anomaly implemented ($\omega = 0.46 \pm 0.14$), the efficacy factor $\varepsilon_{[\mathrm{LI}]}$ is smaller than unity. In other words, these land ice changes contribute comparatively less per unit radiative forcing to the global temperature anomalies than the $CO_2$ changes.

Our inferred PI $S^\varepsilon_{[\mathrm{CO_2,LI}]}$ is equivalent to an $S^a$ of $1.6^{+0.3}_{-0.4}\,\mathrm{K\,W^{-1}\,m^2}$, when only considering the uncertainty caused by the implemented range in $\varepsilon_{[\mathrm{LI}]}$, and to an $S^a$ of $1.6^{+0.1}_{-0.2}\,\mathrm{K\,W^{-1}\,m^2}$, when only considering the uncertainty in the conversion factor

$\phi$. The equivalent ECS is $5.8 \pm 1.3\,\mathrm{K}$ per $CO_2$ doubling, when only considering the uncertainty caused by the implemented range in $\varepsilon_{[\mathrm{LI}]}$, and $5.8 \pm 0.6\,\mathrm{K}$ per $CO_2$ doubling, when only considering the uncertainty in the conversion factor $\Delta R_{2\mathrm{xCO_2}}$. The ECS we find is thus on the high end of the results of other approaches to obtain ECS (Knutti et al., 2017), e.g. the 2.0 to 4.3 K 95%-confidence range from a large model ensemble (Goodwin et al., 2018), and the 2.2 to 3.4 K 66% confidence range from an emerging constraint from global temperature variability and CMIP5 (Cox et al., 2018). Hence, the low end of

our ECS estimate is in the best agreement with these other estimates. This could mean that the relative influence of land ice changes on the LGM temperature anomaly is on the high side, or possibly higher than, the $0.46 \pm 0.14$ range we consider here. Alternatively, the conversion factor $\phi = 0.64 \pm 0.07$ we use to convert $S_{[\mathrm{CO_2,LI}]}$ to $S^a$ is an overestimation, which could be caused by a larger-than-unity efficacy of long-term processes besides $CO_2$ and land ice changes. We have focused primarily on the effect of $\varepsilon_{[\mathrm{LI}]}$ on $S^\varepsilon_{[\mathrm{CO_2,LI}]}$ in this analysis, and therefore we have for simplicity ignored uncertainties in the investigated

proxy-inferred records themselves. A comprehensive description of these uncertainties and their influence on the calculated climate sensitivity can be found in Köhler et al. (2015).

## 5   Conclusions

We have incorporated the concept of a constant efficacy factor (Hansen et al., 2005), that interrelates the global temperature responses to radiative forcing caused by land ice changes and $CO_2$ changes, into our framework of calculating specific pale-

oclimate sensitivity $S^\varepsilon_{[\mathrm{CO_2,LI}]}$. The aim of this effort has been to overcome the problem that land ice and $CO_2$ changes can lead to significantly different global temperature responses, even when they induce the same global-average radiative forcing. Firstly, we have assessed the usefulness of considering efficacy differences by applying our refined approach to results of 5-Myr CLIMBER-2 simulations (Stap et al., 2018), where the separate effects of land ice changes and $CO_2$ changes can be isolated. In the results of these simulations, the error from assuming the efficacy factor to be constant in time is negligible.

Thereafter, we have used our approach to reanalyse an 800-kyr proxy-inferred paleoclimate dataset (Köhler et al., 2015). We have inferred a range in the land ice change efficacy factor $\varepsilon_{[\mathrm{LI}]}$ from the relative impact of land ice changes on the LGM temperature anomaly simulated by a 12-member climate model ensemble (Shakun, 2017). The thusly obtained efficacy factor $\varepsilon_{[\mathrm{LI}]} = 0.45^{+0.34}_{-0.20}$ is smaller than unity, implying that the impact on global temperature per unit of radiative forcing is less strong for land ice changes than for $CO_2$ changes. Consequently, our derived PI $S^\varepsilon_{[\mathrm{CO_2,LI}]}$ of $2.45^{+0.53}_{-0.56}\,\mathrm{K\,W^{-1}\,m^2}$ is $\sim 50\%$ larger than

when efficacy differences are neglected. The equivalent $S^a$ and ECS corresponding to this $S^\varepsilon_{[\mathrm{CO_2,LI}]}$ are $1.6^{+0.3}_{-0.4}\,\mathrm{K\,W^{-1}\,m^2}$

and $5.8 \pm 1.3\,\mathrm{K}$ per $CO_2$ doubling respectively. The uncertainty in these estimates is only caused by the implemented range in $\varepsilon_{[\mathrm{LI}]}$.

*Data availability.* The CLIMBER-2 dataset is available at https://doi.pangaea.de/10.1594/PANGAEA.887427, and the proxy-inferred paleoclimate dataset is available at https://doi.pangaea.de/10.1594/PANGAEA.855449, from the PANGAEA database. For more information or data, please contact the authors.

## 5 Appendix A: Influence of the polar amplification factor

In the analysis performed in Sect. 4.2, we have used a global temperature record that was obtained from northern high-latitude temperature anomalies using a polar amplification factor $f_{\mathrm{PA}}$ that varies from 2.7 at the coldest to 1.6 at the warmest conditions (Sect. 4.1). However, recent climate model simulations of the Pliocene using updated paleogeographic boundary conditions show that in warmer times polar amplification could have been nearly the same as in colder times (Kamae et al., 2016; Chandan and Peltier, 2017). We therefore repeat the analysis using the same range in $\varepsilon_{[\mathrm{LI}]}$ and the same dataset, but with an applied constant $f_{\mathrm{PA}} = 2.7$ over the entire past 800 kyr to generate $\Delta T_g$ ($\Delta T_{g2}$ in Köhler et al. (2015)).

The constant polar amplification used here counteracts increasing state dependency towards low temperatures, as the temperature differences are no longer amplified by changing polar amplification. Hence, $S^{\varepsilon}_{[\mathrm{CO_2,LI}]}$ is smaller at PI, $1.96^{+0.42}_{-0.44}\,\mathrm{K\,W^{-1}\,m^2}$ compared to $2.45^{+0.53}_{-0.56}\,\mathrm{K\,W^{-1}\,m^2}$ using the variable $f_{\mathrm{PA}}$, but diminishes less strongly towards colder conditions (Fig. A1a cf. Fig. 4a). As before, the PALAEOSENS approach (equivalent to our approach using $\varepsilon_{[\mathrm{LI}]} = 1$), yields a lower PI $S_{[\mathrm{CO_2,LI}]}$ of $1.34\,\mathrm{K\,W^{-1}\,m^2}$ (Fig. A1b). The PI $S^{\varepsilon}_{[\mathrm{CO_2,LI}]}$ inferred here using our refined approach corresponds to an $S^a$ of $1.3^{+0.2}_{-0.3}\,\mathrm{K\,W^{-1}\,m^2}$, and an ECS of $4.6^{+1.0}_{-1.3}\,\mathrm{K}$ per $CO_2$ doubling.

*Author contributions.* L.B.S. designed the research. L.B.S. and P.K. performed the analysis. L.B.S. drafted the paper, with input from all co-authors.

*Competing interests.* The authors declare that they have no conflict of interest.

*Acknowledgements.* This work is institutionally funded at AWI via the research program PACES-II of the Helmholtz Association. We thank Roderik van de Wal for commenting on an earlier draft of the manuscript, and two anonymous referees for their constructive comments, which have helped to improve the quality of the manuscript.

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

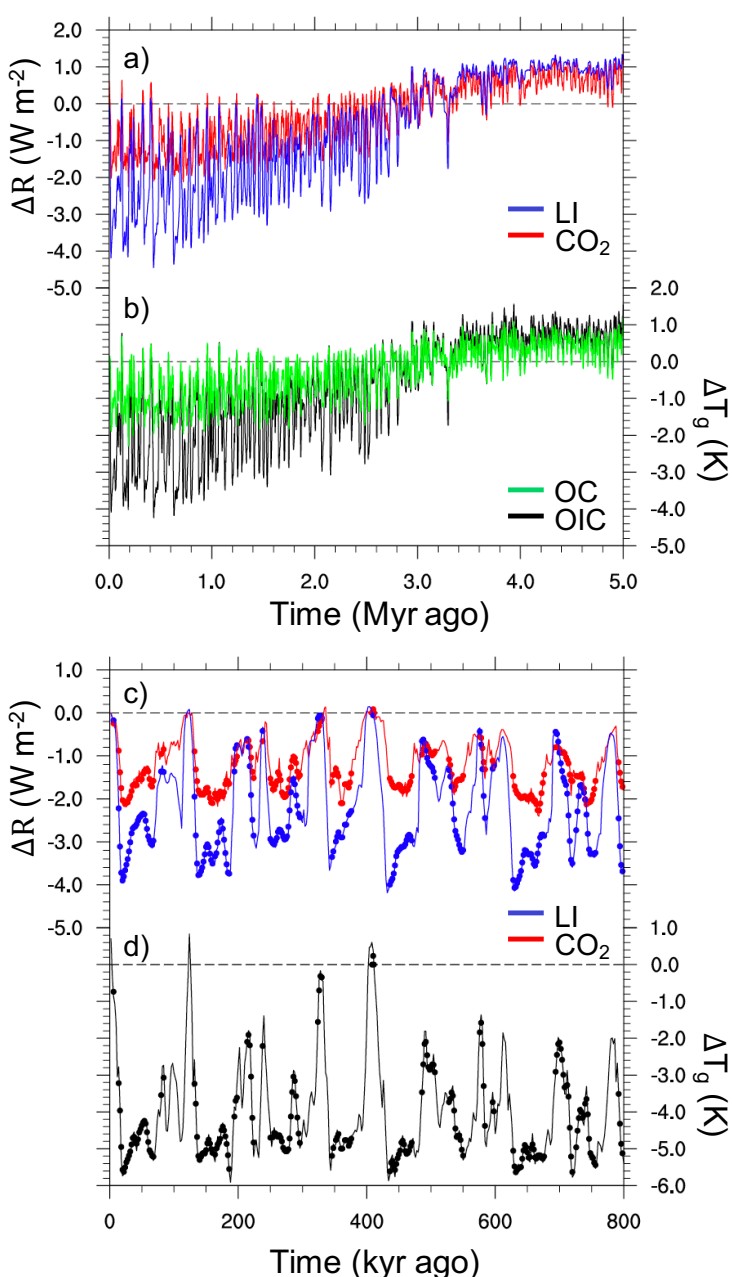

**Figure 1.** Timeseries of radiative forcing anomalies ($\Delta R$) caused by $CO_2$ (red) changes and land ice changes (blue), and global temperature anomalies ($\Delta T_g$) with respect to PI, from **a-b)** the CLIMBER-2 model dataset (Stap et al., 2018), with temperature data for experiment OIC in black and for experiment OC in green, and from **c-d)** the proxy-inferred dataset (Köhler et al., 2015), with solid lines for the whole dataset, and dots for the data used in this study which exclude times with strong temperature-$CO_2$ divergence (see Sect. 4.1). Note the differing axis scales.

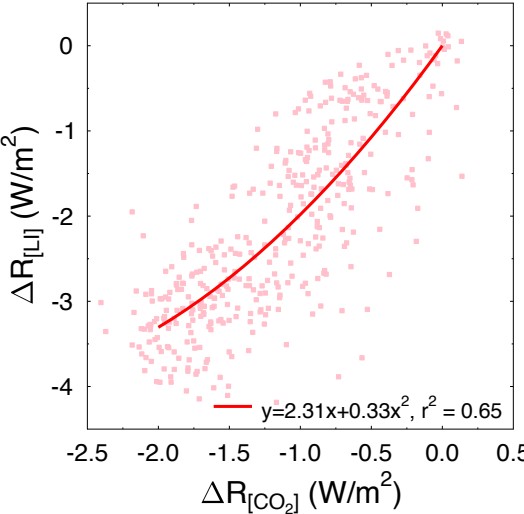

**Figure 2.** The relation between radiative forcing anomalies caused by $CO_2$ changes ($\Delta R_{[CO_2]}$) and land ice changes ($\Delta R_{[LI]}$) from the whole proxy-inferred dataset (Köhler et al., 2015) (pink dots). The red line represents a second order polynomial least-squares regression through the scattered data.

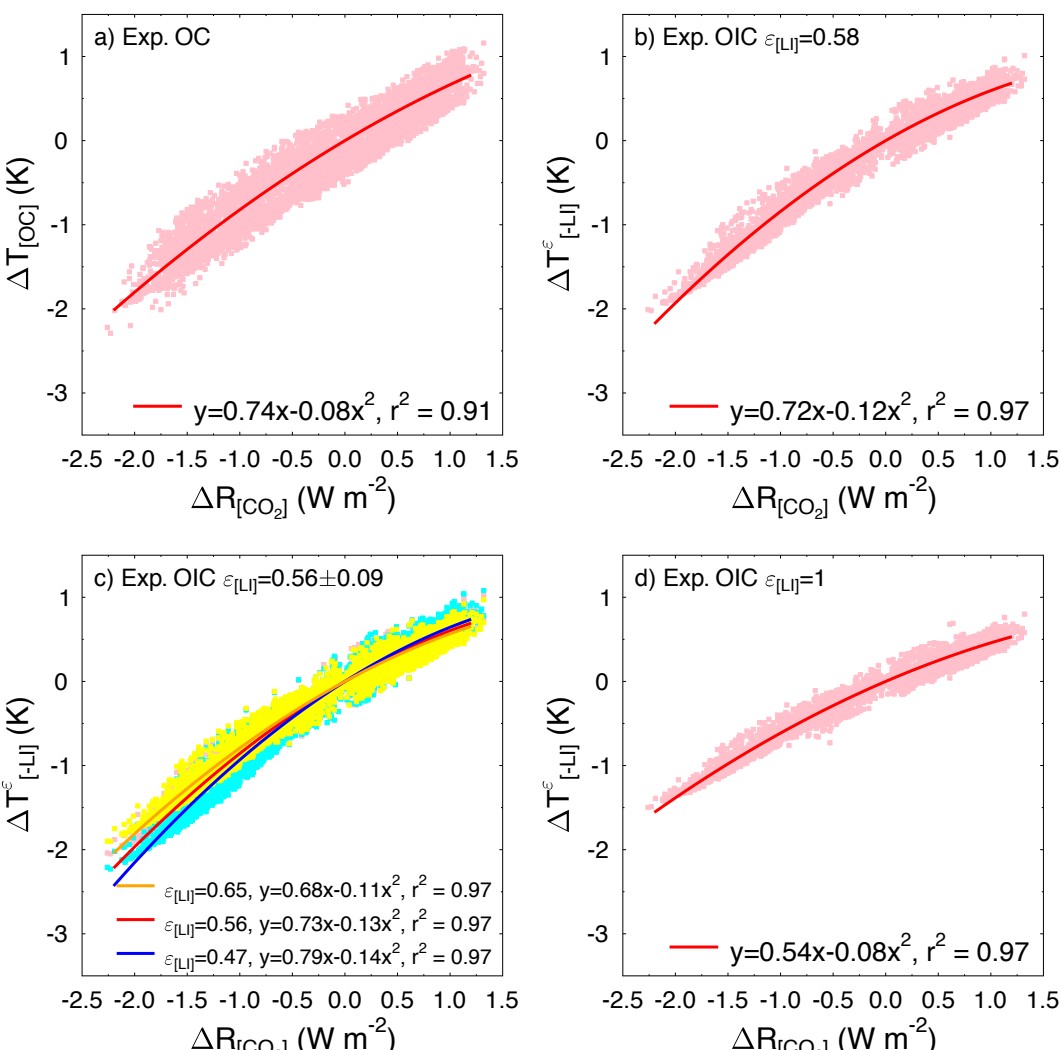

**Figure 3.** Temperature anomalies with respect to PI over the last 5 Myr from CLIMBER-2 (Stap et al., 2018) against imposed radiative forcing of $CO_2$. **a)** Simulation with fixed PI land ice distribution (experiment OC) ($\Delta T_{[OC]}$). **b)** Calculated global temperature perturbations from experiment OIC stripped of the inferred influence of land ice ($\Delta T_{[-LI]}^{\varepsilon}$) using Eq. 16 with $\varepsilon_{[LI]} = 0.58$. Here, $\varepsilon_{[LI]}$ is obtained from matching climate sensitivity with the target value at the LGM. **c)** Same as in (b), but using $\varepsilon_{[LI]} = 0.47$ (cyan dots), $\varepsilon_{[LI]} = 0.56$ (pink dots), and $\varepsilon_{[LI]} = 0.65$ (yellow dots), Here, $\varepsilon_{[LI]}$ is obtained from the mean ($\pm 1\sigma$) of matching climate sensitivity with the target value at all glacial marine isotope stages of the past 810 kyr (MIS 2, 6, 8, 10, 12, 14, 16, 18, and 20). **d)** Same as in (b), but using $\varepsilon_{[LI]} = 1$, which is equivalent to the PALAEOSENS approach where efficacy differences were not considered. The red lines - and in **(c)** also the orange and blue lines - represent second order polynomial least-squares regressions through the scattered data.

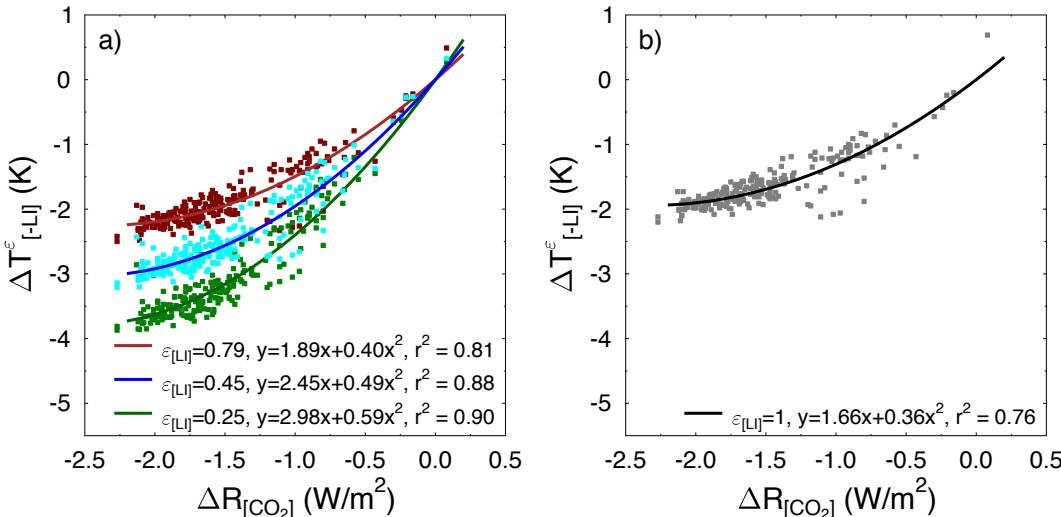

**Figure 4.** The global temperature perturbations stripped of the inferred influence of land ice ($\Delta T^{\varepsilon}_{[-\text{LI}]}$) calculated using Eq. 16 against $\Delta R_{[\text{CO}_2]}$ from the proxy-inferred paleoclimate dataset (Köhler et al., 2015), using: **a)** $\varepsilon_{[\text{LI}]} = 0.79$ (maroon dots), $\varepsilon_{[\text{LI}]} = 0.45$ (cyan dots), and $\varepsilon_{[\text{LI}]} = 0.25$ (green dots). Here, $\varepsilon_{[\text{LI}]}$ is obtained by converting the multi-model assemblage of simulated relative influences of land ice changes on the LGM temperature anomaly ($0.46 \pm 0.14$) (Shakun, 2017). **b)** Same as in (a), but using $\varepsilon_{[\text{LI}]} = 1$ (grey dots), which is equivalent to the PALAEOSENS approach. The brown, blue, dark green (**a**), and black lines (**b**) represent second order polynomial least-squares regressions through the data.

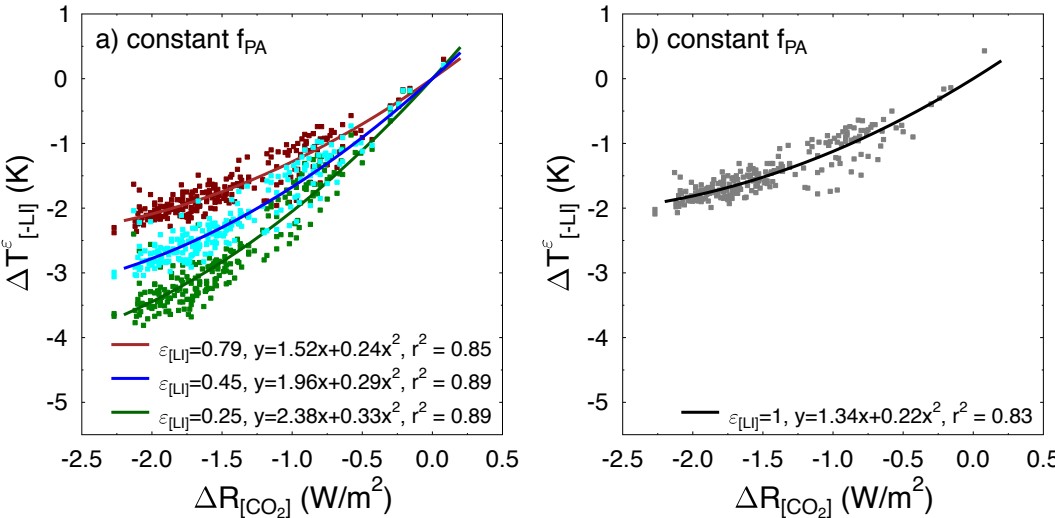

**Figure A1.** The global temperature perturbations stripped of the inferred influence of land ice ($\Delta T^{\varepsilon}_{[-LI]}$) calculated using Eq. 16 against $\Delta R_{[CO_2]}$ from the proxy-inferred paleoclimate dataset (Köhler et al., 2015), using: **a)** $\varepsilon_{[LI]} = 0.79$ (maroon dots), $\varepsilon_{[LI]} = 0.45$ (cyan dots), and $\varepsilon_{[LI]} = 0.25$ (green dots). Here, $\varepsilon_{[LI]}$ is obtained from converting the multi-model assemblage of simulated relative influences of land ice changes on the LGM temperature anomaly ($0.46 \pm 0.14$) (Shakun, 2017). **b)** Same as in (a), but using $\varepsilon_{[LI]} = 1$ (grey dots), which is equivalent to the PALAEOSENS approach. The brown, blue, dark green (**a**), and black lines (**b**) represent second order polynomial least-squares regressions through the data. Here, the global temperature anomalies are derived from the northern high-latitude temperature anomaly reconstruction assuming a constant polar amplification factor ($f_{PA}$) of 2.7, as opposed to the variable $f_{PA}$ used in Fig. 4.