# Peer review of "Including the efficacy of land ice changes in deriving climate sensitivity from paleodata"

_Earth System Dynamics, 2018_

## Referee Comment (RC1) · Anonymous Referee #1 · 14 Jan 2019

**Review of Stap et al. 2018**

In this paper the authors try to address a matter of importance — that concerning the efficacy of the different radiative forcing — and which is directly relevant to the ongoing efforts by various modelling and proxy analysis groups to estimate the planet's Equilibrium Climate Sensitivity (ECS). I like the idea of the paper and I am quite sure the paper will be accepted, but I feel there is need for clarity and additional analysis before the paper is in publishable form.

**Points of broadest significance**

**Definition of the efficacy factor:** This paper builds upon the work by Hansen et al. 2005 and by PALEOSENS members (2012), but the way the authors introduce the efficacy factor in equations (8) and (9) is different from those employed in these other works. For example, according to the PALEOSENS approach, equation (9) should be expressed as (See sample calculation in PALEOSENS supplementary materials section B.2):

$$S^\epsilon_{[CO_2,LI]} = \frac{\Delta T_g}{\Delta R_{CO_2} + \Delta R_{LI}} = \frac{\Delta T_g}{\Delta R_{CO_2} + \epsilon_{[LI]}\Delta R_{CO_2}}$$

This says that the efficacy of the radiative forcing from land ice changes, $\Delta R_{[LI]}$ is related to the equivalent radiative forcing from changes in $CO_2$ through is a fractional parameter $\epsilon_{[LI]}$. This is what the efficacy is meant to serve: to help assess the radiative forcing from non-greenhouse gas sources by relating it to the better constrained forcing from $CO_2$. But the way the authors are using $\epsilon_{[X]}$ is quite strange and it doesn't make sense to me. It doesn't appear to be a typographical mistake. The climate sensitivity world is already overflowing with numerous different formulations and I think there should be a very good reason (and which should be made extremely clear in the paper) to define an existing concept differently.

**Regarding the sample calculations from CLIMBER experiment:** The authors try to apply their new formulation to compute $S^\epsilon_{[CO_2,LI]}$ from their CLIMBER data and compare it to $S_{[CO_2,LI]}$ that they have previously found. Using

$$S_{[CO_2,LI]} = \frac{\Delta T_g}{\Delta R_{CO_2} + \Delta R_{LI}}$$

the authors found $S_{[CO_2,LI]}$ to be 0.54. This formulation uses $\Delta T_g$, $\Delta R_{[CO_2]}$ and $\Delta R_{[LI]}$ all of which are available from their CLIMBER models (and shown in Fig 1). Their new formulation $S^\epsilon_{[CO_2,LI]}$, after substituting for $\epsilon_{[LI]}\Delta R_{[LI]}$ from equation (11) into equation (9) reduces to

$$S^\epsilon_{[CO_2,LI]} = \frac{\Delta T_g - \Delta T_{LI}}{\Delta R_{CO_2}}$$

in which all the terms are again derived from their CLIMBER models, the only difference from the original expression is that instead of $\Delta R_{[LI]}$ the new expression uses $\Delta T_{[LI]}$. I am quite confused why the new approach using temperature from land ice changes, instead of radiative forcing due to land ice changes, (both from the same set of models), and leading to a higher inferences of S is to be favoured (a sentiment expressed at the start of page 8)?

**Constant $\epsilon_{[LI]}$:** The authors have talked a lot about the state dependency of $S^{\epsilon}_{[CO_2,LI]}$, but they have barely discussed the state dependency of $\epsilon_{[LI]}$, which is the bread and butter of this paper. After all, $\epsilon_{[LI]}$ will likely depend on state and it can be readily computed for either their numerical model or the paleo data using their equation (11) and therefore the variability can be assessed in the manuscript. The conclusion says "the assumption that the efficacy factor is indeed constant in time could be tested more rigorously using more sophisticated climate models", but it can be tested in this manuscript using the models and data they are already employing. Furthermore, in the absence of this analysis, the usage of LGM specific $\epsilon_{[LI]}$ in calculations, and which is applied as a constant value to the entirety of the Pleistocene time series makes the analysis look very contrived. The reader does not know, if the results change a lot if $\epsilon_{[LI]}$ is derived, from say MIS5 and then kept constant for the entire interval of analysis? So the range of changes in $\epsilon_{[LI]}$ and the dependence of principle results on that should be included in the manuscript.

**Section 3.3:** A big shortcoming of this manuscript is section 3.3 which is extremely convoluted and difficult to follow. For an otherwise relatively clearly written paper, this section seems to have been put together haphazardly without the attention to detail that makes the rest of the paper readily readable. Though I have made a couple of specific comments for this section further down in my review, in general I have not been able to follow this section at all and therefore have not been able to provided the quality of feedback that I would have liked. A careful re-writing of this section by the authors is required.

**Scientific Comments**

1. The various sensitivities are quotes in two different units throughout the paper: K per doubling of $CO_2$ and $K\,W^{-1}m^2$. While the authors have been generally very clear about the units and about converting between them, as is the case on page 8, I do encourage them to use only one unit throughout the paper. This helps a reader to quickly compare various numbers from across the paper without having to convert the units. Alternatively, the authors could quote all sensitivities in both units, example: "so and so sensitivity was found to be 1.66 $K\,W^{-1}m^2$ or equivalently 5.6 K per doubling of $CO_2$" (similar to the last sentence in the conclusions section).
2. Sentence spanning lines 9–10 on page 3: I don't understand what is meant by this sentence, specifically by the part "it has been shown that simulations of models that have been integrated over a few centuries are not yet in equilibrium". Perhaps rephrasing this sentence could make it clearer.

3. Line 10 page 3: Regarding ECS the authors say "Another way to express" but no other way has been previously mentioned until that point in the article. The ECS has only been defined up to that point. I think it makes more sense to rephrase it as "One way to express".

4. Since the form of "f" is important for the rest of the paper, the authors should clearly articulate the motivations for f as given in equation 5.

5. Last para, page 4: So is $S_{[CO_2,LI]}$ to be considered as an estimate of $S_a$? Maybe the authors should clarify this explicitly. In the process of making this clarification the starting sentence of that paragraph will likely need to be modified to make the argument fit in seamlessly.

6. In the first paragraph on page 5 the authors say that they take a "further simplifying step" to more easily compare "$S^{\epsilon}_{[CO_2,LI]}$ to other specific paleoclimates sensitivities $S^{\epsilon}_{[CO_2,X]}$ by unifying the dependent variable". But all they have done is move the specific dependent variable $\Delta R_{[X]}$ into the newly defined $CO_2$-equivalent temperature and which doesn't in any way free someone of the need to compute that forcing or to compute the efficacy factor. So I fail to see the simplification here (besides a notational one) but more importantly I fail to see the practical usefulness. For any given $S_{[CO_2,LI]}$ by the time one has computed the $CO_2$-equivalent temperature, they might as well have just used equation 9.

7. First para, page 6: In the experiments OC, and OI, which as I understand are meant to assess the effects of land ice and $CO_2$ respectively, why are the orbital conditions also varied in conjunction? It seems that the authors answer this later on in the manuscript, at the beginning of section 3.1: "since the influence of orbital variations is very small". That comment should be moved closer to where these experiments OC and OI are discussed.

8. Line 21, page 6: "ANICE was forced by northern hemisphere temperatures obtained…" Northern hemispheres temperature or temperature anomaly? I think it should be the anomaly.

9. Page 6: regarding the discussion of the amplification factor for the Pliocene, new results coming from the revised paleo-geographic boundary conditions for PlioMIP2 (Kamae et al. 2016; Chandan and Peltier 2017; Hunter et al. 2019) that suggest that the amplification factor could have been larger. Models that were used in the previous PlioMIP and whose results were synthesized in Haywood et al. 2013 were consistently failing to produce the polar amplification that has been inferred from proxies. With the new results the polar amplification factor in the warm interval of the Pliocene is nearly the same as the amplification factor during the cold LGM. The authors should and cite the new papers add a comment/analysis regarding how the revised amplification factor for the warm interval affects their results.

10. Lines 15-17 page 7: the authors say they are inferring $S^{\epsilon}_{[CO_2,LI]}$ or $S^{\epsilon}_{[CO_2]}$ here but I think a bit of additional comment is required to clarify the appearance of $\epsilon$ in these sensitivities. These are after all inferred from experiment OC in which $\Delta R_{[LI]}$ is zero, so the meaning of land-ice radiative efficacy $\epsilon$ is not strictly defined. This is probably hair-splitting over notation but I think it is best to be as clear as possible since the climate sensitivity literature is already overflowing with (sometimes sloppily used) notation.

11. Line 1, page 8: "the new approach considering efficacies clearly leads to a more satisfactory result that the old approach." In the present form this sentence implies that for some reason the numerical value 0.74 is more satisfactory than the older value of 0.54. I am not sure if that is defensible or even that the authors themselves meant to imply that. I think the authors meant to

say something like "the new approach is more flexible/accommodating/physically accurate than the old approach". Please re-phrase this accordingly.

12. Lines 13–15, page 8: The authors have presented two results which lead to opposite conclusions. This needs to be addressed here directly instead of referring the reader to another publication. While the issue may have been more thoroughly assessed in Köhler et al. 2018, a brief comment should also be provided here so that the reader grasps the discordance in the author's results at a bare-minimum level without having to read up another paper.

13. Line 9, page 8: For the calculation of $\epsilon_{[LI]}$ using equation 11 please provide the values of $\Delta R_{CO_2}$ and $\Delta R_{LI}$ at LGM that were used.

14. Line 15, page 8: is the mean the value "of" years 20 and 22 kya or "between" those years?

15. Line 16, page 8: "The specific paleo climate sensitivities we find here are generally higher than calculated by the old approach" But the new sensitivity calculated is 1.39 which is lower than that by the old approach which was 1.66.

16. Line 12 page 9: "We correct the induced $\Delta T_{[CO_2]}$ of all individual models for this ratio" I don't follow.

17. Line 15 page 9: At this point I am lost. Why are you doing that regression? What it the motivation? And are you subtracting the global value $\Delta T_{[CO_2]}$ from $\Delta T_{NH}$?

18. The ECS given in Table 1 for the CCSM4 model is different from that usually cited. Bitz et al. 2012 using the NCAR-CCSM4 and recently Chandan and Peltier, 2018 using a related UofT-CCSM4 have deduced the ECS to be 3.2. The value in Table 1 is lower than that. Where did the authors get this from? Haywood et al. 2013 also use CCSM4 ECS (from Bitz et al) of 3.2. Do the numbers for the other models need to be checked as well?

19. The authors should cite all the original experiment design papers for the PMIP3 experiments listed in Table 1. This can be done readily by adding a new column to the table called "References".

20. The figure description for Figure 3 is completely wrong. It is talking about things that are not on the figure.

**Technical Comments**

1. Line 2 page 1:" to equilibrium"
2. Line 29 page 2: "are obtained from  various model setups"
3. Line 16 page 3: "In this case, the average global paleo temperature anomaly with respect to the pre-industrial (PI)  ($\Delta T_{[g]}$) is"
4. Line 17 page 3: "that are typically neglected in  climate simulations".
5. Lines 3-4 on page 4 incorporating the phrase "the calculated paleoclimate sensitivity" in the current form refers to some specific and as yet undefined sensitivity. It's best to rephrase it as "If, for instance, only the most important slow feedback in the climate system, namely radiative forcing anomalies induced by albedo changes due to land ice (LI) variability are taken into account, then one can correct $S^p$ to derive the following specific paleoclimate sensitivity."

6. The sentence on line 5, page 4, appears as a sharp interruption to the logic train before and after that sentence. It should instead be placed at the end of that paragraph and rephrased as " A synthesis of  estimates of $S_{[CO_2,LI]}$  from both …."

7. Line 15 page 4: " because, e.g."

8. Line 18 page 4: "through efficacy factors ($\epsilon_{[X]}$). This requires a reformulation"

9. Line 20 page 4: "to clearly distinguish  the sensitivities from  those of the PALAEOSENS project in which the radiative forcing of the different processes  were assigned identical efficacies."

10. Line 25 page 4: "by land ice changes ($\epsilon_{[LI]}$), using  the following formulation which is based on, but modified from Hansen et al. (2005)"

11. Line 22 page 5: The sentence "CLIMBER-2 combined a 2.5 statistical-dynamical…" seems something is missing after 2.5. Did the authors mean "2.5 degree"?

12. Line 5 page 5: Add comma after "Similarly"

13. Line 3 page 5: "leaving 217 data points as indicated in Fig 1c,d."

14. Sentence beginning on line 18 page 7: change it to something like "For our first attempt at compensating paleoclimates sensitivity for slow processes other than $CO_2$ changes we strive to deduce the same $S^\epsilon_{[CO_2,LI]}$, inferred above, from experiment OIC in which both $CO_2$ and land ice cover vary over time".

15. Line 23 page 7: "Between ….  some  outliers resulted from division  by small numbers (not shown on Fig. 2b)."

16. Line 29 page 7: "….is more linear than that  between …"

17. Line 31 page 7 " through the entire 5 million year interval."

18. Line 11 page 8: " Similar to Köhler et al., 2018, we too detect"

19. Line 15 page 8: the value of $\Delta R_{[CO_2]}$ should be -2.04

20. Line 15 page 8: "the LGM value ( taken here as the mean…)"

21. Line 21 page 8: "we first scale  it by a factor"

22. Line 23 page 8: "Note that this scaling still assumes unit efficacy for  process other than land ice changes"

23. Line 24 page 8: "Then, after  multiplying by"

24. Line 24 page 8: Units should be $Wm^{-2}$

25. Line 11 page 9: "that the ratio of the radiative forcing change $\Delta R_{[CO_2]}$ between the LGM (185 ppm $CO_2$) and the PI (280 ppm $CO_2$), to the change between the PI and  a $2 \times$ PI case is

26. Line 16 page 9: "significant  at the 95% level"

27. Conclusions section, Lines 26, 28, 30: $\epsilon_{[CO_2,LI]}$ is a new symbol not previously defined. It seems like a mistake and the authors likely meant $\epsilon_{[LI]}$

28. The yellow star in Fig 4 is barely visible against the cyan background. Please change it to something dark, maybe black.

**References**

Bitz, C., Shell, K.M., Gent, P.R., Bailey, D.A., Danabasoglu, G., Armour, K.C., Holland, M.M., Kiehl, J.T., 2012. Climate Sensitivity of the Community Climate System Model, Version 4. J. Climate 25, 3053–3070. doi:10.1175/JCLI-D-11-00290.1

Chandan, D., Peltier, W.R., 2017. Regional and global climate for the mid-Pliocene using the University of Toronto version of CCSM4 and PlioMIP2 boundary conditions. Clim. Past 13, 919–942. doi:10.5194/cp-13-919-2017

Chandan, D., Peltier, W.R., 2018. On the mechanisms of warming the mid-Pliocene and the inference of a hierarchy of climate sensitivities with relevance to the understanding of climate futures. Clim. Past 14, 825–856. doi:10.5194/cp-14-825-2018

Hansen, J., Sato, M., Ruedy, R.A., Nazarenko, L.S., Lacis, A.A., Schmidt, G.A., Russell, G., Aleinov, I., Bauer, M., Bell, N., Cairns, B., Canuto, V., Chandler, M.A., Cheng, Y., Del Genio, A., 2005. Efficacy of climate forcings. J. Geophys. Res. 110. doi:10.1029/2005JD005776

Haywood, A.M., Hill, D.J., Dolan, A.M., Otto-Bliesner, B.L., Bragg, F.J., Chan, W.-L., Chandler, M.A., Contoux, C., Dowsett, H.J., Jost, A., Kamae, Y., Lohmann, G., Lunt, D.J., Abe-Ouchi, A., Pickering, S.J., Ramstein, G., Rosenbloom, N.A., Salzmann, U., Sohl, L.E., Stepanek, C., Ueda, H., Yan, Q., Zhang, Z., 2013. Large-scale features of Pliocene climate: results from the Pliocene Model Intercomparison Project. Clim. Past 9, 191–209. doi:10.5194/cp-9-191-2013

Hunter, S. J., Haywood, A. M., Dolan, A. M., and Tindall, J. C.: The HadCM3 contribution to PlioMIP Phase 2 Part 1: Core and Tier 1 experiments, Clim. Past Discuss., https://doi.org/10.5194/cp-2018-180, in review, 2019

Kamae, Y., Yoshida, K., Ueda, H., 2016. Sensitivity of Pliocene climate simulations in MRI-CGCM2.3 to respective boundary conditions. Clim. Past 12, 1619–1634. doi:10.5194/cp-12-1619-2016

PALEOSENS Members, 2012. Making sense of palaeoclimate sensitivity. Nature 491, 683–691. doi: 10.1038/nature11574

---

## Referee Comment (RC2) · Anonymous Referee #2 · 15 Jan 2019

My initial thoughts on seeing this paper were very positive in the sense that, given the uncertainty of information in the paleorecord, and the difficulty of using state of the art models to make very long runs, all progress in the area of better defining climate sensitivity as it relates to past climates is worthwhile.

My optimism remained through the first parts of the paper, but by the end I have to admit that I am lost and really do not understand what the authors are trying to do and what they have discovered.

The authors introduce a variable, DTe[CO2 - equiv] but do not explain why this is useful or interesting. What I would have done is take equation (9), replace X with LI and then explore all the elements of that equation. This would show us how S varies with DRLI and DRCO2, as well a DTg and one could consider how much of the

state+forcing+efficacy dependence of S[CO2] is accounted for by considering land ice with and without considering efficacy. I can see that Figures 2 and 3 represent some kind of sensitivity-like variable, but I cannot grasp its meaning. Basically, DTe[CO2 - equiv] is not, as you suggest in equation 14 simply a function of DRCO2 but also depends on Tg and DRLI. I hope that the remedy is a better explanation of the reasons behind the derivations in section 1 and also better explanation of the insight that you gain from the results.

Other points.

P1L10 "Recently, it has been shown that simulations of models that have been integrated over a few centuries are not yet in equilibrium, and from longer climate simulations a higher ECS can be deduced (Knutti et al., 2017)."

This needs rephrasing. It has been well known since before dynamical oceans were included in climate models that the equilibrium time of the ocean is of the order of thousands of years. Since the invention of the AOGCM, ad-hoc methods have been introduced to try to estimate equilibrium climate sensitivity without running the models to equilibrium. What recent work has been doing is assessing the accuracy of such approximations.

P2L23 "likewise as several earlier studies"

-> "as in several earlier studies"

Eq(10) This equation suggests to me that DTg-DT[LI]=DT[CO2]. Maybe I misunderstand, but it seems to me that DTg=DT[LI]+DT[CO2]+DT[X]+Z, where DT[X] is the influence of all the other forcings and Z represents cross terms (ie nonlinearities).

P3L5 "Similarly as in the old approach," Not English

Eq(13) Looks like a minus sign between "CO2" and "equiv".

P5L12 "A functional relationship between TE[CO2−equiv] and R[CO2] (T[CO2−equiv]

= g(R[CO2])) can be obtained by least squares regressions of higher-order polynomial to the scattered data of these variables."

It is not clear which variables are "these variables".

Sections 2 and 3

I think the paper order should be 2.2.1, 3.1 then 2.2.2, 3.2. The way it is presented is just confusing. Present the whole of the simple modelling case and then move on to the data-based case.

P7L22 "and again fit a second order polynomial to the scattered data of T" [CO2−equiv]"

Which experiment?

P8L12 "Similarly as before" Not English

Table 1 State which paper each "published ECS" comes from.

I prefer to write reviews before reading what other reviewers have posted, as I feel I will be too easily influenced, so I did not read the other reviewer's comment until now. I am encouraged to see that the other reviewer also found the paper very difficult to follow. This increases my optimism that there is hope that with better explanation in critical areas, and reorganisation to improve the storyline, that the paper may become both comprehensible and publishable.

---

## Author Comment (AC1) · 25 Feb 2019

*'Including the efficacy of land ice changes in deriving climate sensitivity from paleodata'*
*by L.B. Stap, P. Köhler and G. Lohmann.*
*Submitted for potential publication by Earth System Dynamics*

REPLY TO THE COMMENTS BY THE REVIEWERS

Color coding:
Black – comments by reviewers
Green – reply by authors

**Reviewer #1**

In this paper the authors try to address a matter of importance — that concerning the efficacy of the different radiative forcing — and which is directly relevant to the ongoing efforts by various modelling and proxy analysis groups to estimate the planet's Equilibrium Climate Sensitivity (ECS). I like the idea of the paper and I am quite sure the paper will be accepted, but I feel there is need for clarity and additional analysis before the paper is in publishable form.

We thank the reviewer for a careful examination of our work. We are pleased that the reviewer likes the idea of the paper. In the revised manuscript, we have largely followed the provided comments to improve the clarity of the paper, and we have included additional analysis, as described below.

**Points of broadest significance**

**Definition of the efficacy factor:** This paper builds upon the work by Hansen et al. 2005 and by PALEOSENS members (2012), but the way the authors introduce the efficacy factor in equations (8) and (9) is different from those employed in these other works. For example, according to the PALEOSENS approach, equation (9) should be expressed as (See sample calculation in PALEOSENS supplementary materials section B.2):

$$S^{\varepsilon}_{[CO2,LI]} = \frac{\Delta T_g}{\Delta R_{CO2} + \Delta R_{LI}} = \frac{\Delta T_g}{\Delta R_{CO2} + \varepsilon_{LI}\Delta R_{CO2}}$$

This says that the efficacy of the radiative forcing from land ice changes, $\Delta R_{[LI]}$ is related to the equivalent radiative forcing from changes in $CO_2$ through is a fractional parameter $\varepsilon_{[LI]}$. This is what the efficacy is meant to serve: to help assess the radiative forcing from non-greenhouse gas sources by relating it to the better constrained forcing from $CO_2$. But the way the authors are using $\varepsilon_{[X]}$ is quite strange and it doesn't make sense to me. It doesn't appear to be a typographical mistake. The climate sensitivity world is already overflowing with numerous different formulations and I think there should be a very good reason (and which should be made extremely clear in the paper) to define an existing concept differently.

We would like to argue that the way we have implemented the efficacy factor in our approach is the most natural extension to (our) earlier studies. Indeed, the PALAEOSENS approach, which we have used so far, employs:

$$S_{[CO2,LI]} = \frac{\Delta T_g}{\Delta R_{[CO2]} + \Delta R_{[LI]}}.$$

Mind though: no superscript $\varepsilon$ here, because efficacy differences are not considered.

In Köhler et al. (2010), radiative forcing records over the past 800 kyr of many different processes including $CO_2$ and land ice changes, were analyzed. This is not the issue we consider in this manuscript. Instead, we try to overcome the problem that the strength of the response of global-average temperature to global-average radiative forcing can be different, depending on the generating process (in this case land ice changes or $CO_2$ changes), so in general:

$$\frac{\Delta T_{[CO2]}}{\Delta R_{[CO2]}} \neq \frac{\Delta T_{[LI]}}{\Delta R_{[LI]}}.$$

In our opinion, the most logical approach to include this difference in efficacy is to multiply the radiative forcing of land ice changes by an appropriate factor so that the strength of the temperature response is the same as when $CO_2$ would be the generating process. Indeed, Hansen et al. (2005) compared the effects of several different processes, expressing the efficacy of these processes X as:

$$\varepsilon_{[X]} = \frac{\Delta T_{[X]} / \Delta R_{[X]}}{\Delta T_{[CO2]} / \Delta R_{[CO2]}}, \text{ so in our case: } \varepsilon_{[LI]} = \frac{\Delta T_{[LI]} / \Delta R_{[LI]}}{\Delta T_{[CO2]} / \Delta R_{[CO2]}}.$$

Our implementation,

$$\frac{\Delta T_{[LI]}}{\Delta R_{[LI]}} = \varepsilon_{[LI]} \frac{\Delta T_g - \Delta T_{[LI]}}{\Delta R_{[CO2]}},$$

follows this approach very closely. The only difference is that we relate the effect of land ice changes (left hand side) to the effect of all processes except land ice changes (right hand side), because we calculate specific climate sensitivity $S_{[CO2,LI]}^{\varepsilon}$, which does not account for the effect of these other processes.

In principal, it is also possible to relate the impact of land ice changes on global temperature directly to $\Delta R_{[CO2]}$, as the reviewer proposes:

$$S_{[CO2,LI],\text{alt}}^{\varepsilon} = \frac{\Delta T_g}{\Delta R_{[CO2]} + \varepsilon_{[LI],\text{alt}} \Delta R_{[CO2]}},$$

where the efficacy factor in this alternative case ($\varepsilon_{[LI],\text{alt}}$) relates to the one used in our approach ($\varepsilon_{[LI]}$) as:

$$\varepsilon_{[LI],\text{alt}} = \varepsilon_{[LI]} \frac{\Delta R_{[LI]}}{\Delta R_{[CO2]}}.$$

However, from the records of $\Delta R_{[CO2]}$ and $\Delta R_{[LI]}$ of our dataset, we infer a non-linear relationship between these two quantities (see the figure below, included as the new Fig. 2 in the revised manuscript). This would introduce a cumbersome state dependency of $\varepsilon_{[LI]}$, which is avoided by our approach. This has now been elaborated upon in the revised manuscript.

[Figure]

*Figure: Relation between radiative forcing anomalies caused by CO$_2$ changes ($\Delta R_{[CO2]}$) and land ice changes ($\Delta R_{[LI]}$) from the proxy-inferred dataset (pink dots). The red line represents a second order polynomial least-squares regression through the scattered data.*

Furthermore, for clarification we have split the method section (Section 2.1) into three parts in the revised manuscript, describing 1) the PALAEOSENS approach used in earlier studies, 2) our main refinement: the inclusion of the efficacy of land ice changes, and 3) a small refinement that unifies the dependent variable in cross-plots of radiative forcing and global temperature anomalies.

**Regarding the sample calculations from CLIMBER experiment:** The authors try to apply their new formulation to compute $S^{\varepsilon}_{[CO2,LI]}$ from their CLIMBER data and compare it to $S_{[CO2,LI]}$ that they have previously found. Using

$$S_{[CO2,LI]} = \frac{\Delta T_g}{\Delta R_{[CO2]} + \Delta R_{[LI]}}$$

the authors found $S_{[CO2,LI]}$ to be 0.54. This formulation uses $\Delta T_g$, $\Delta R_{[CO2]}$ and $\Delta R_{[LI]}$ all of which are available from their CLIMBER models (and shown in Fig 1). Their new formulation $S^{\varepsilon}_{[CO2,LI]}$, after substituting for $\varepsilon_{[LI]}\Delta R_{[LI]}$ from equation (11) into equation (9) reduces to

$$S^{\varepsilon}_{[CO2,LI]} = \frac{\Delta T_g - \Delta T_{LI}}{\Delta R_{CO2}}$$

in which all the terms are again derived from their CLIMBER models, the only difference from the original expression is that instead of $\Delta R_{[LI]}$ the new expression uses $\Delta T_{[LI]}$. I am quite confused why the new approach using temperature from land ice changes, instead of radiative forcing due to land ice changes, (both from the same set of models), and leading to a higher inferences of S is to be favoured (a sentiment expressed at the start of page 8)?

The goal of this section is to validate our refined approach by applying it to the idealized CLIMBER-2 simulations. Here, the effect of $CO_2$ is a-priori known from the results of experiment OC:

$$S^{\varepsilon}_{[CO2,LI]} = \frac{\Delta T_{OC}}{\Delta R_{[CO2]}}.$$

This result functions as the target for our approach of obtaining the sole effect of $CO_2$ changes on global temperature from the results of experiment OIC, where land ice cover and $CO_2$ levels are both varied over time. Our refined approach considers the efficacy of land ice changes:

$$S^{\varepsilon}_{[CO2,LI]} = \frac{\Delta T_g}{\Delta R_{[CO2]} + \varepsilon_{[LI]}\Delta R_{[LI]}}.$$

We calculate the efficacy factor $\varepsilon_{[LI]}$ as:

$$\varepsilon_{[LI]} = \frac{\omega}{1-\omega}\frac{\Delta R_{[CO2]}}{\Delta R_{[LI]}},$$

where

$$\omega = \left.\frac{\Delta T_{[LI]}}{\Delta T_g}\right|_{\text{spec. time}}.$$

Note here that the parameter $\omega$ is obtained from temperatures at a specific time (for instance, the LGM), constituting the assumption that $\varepsilon_{[LI]}$ is constant in time. Therefore, the simplification that the reviewer makes by substituting equation (11) into equation (9) is not generally valid. Otherwise, the refined approach would indeed by construction always yield the target value for $S^{\varepsilon}_{[CO2,LI]}$ (apart from a negligible contribution by the synergy of $CO_2$ and land ice changes). Instead, $S^{\varepsilon}_{[CO2,LI]}$ is only matched by construction at the LGM. The comparison we make between our approach and the target, provides a quantification of the error yielded by assuming a time-invariant $\varepsilon_{[LI]}$, which has been clarified in the revised manuscript (see also our answer to the next general comment).

We do not favour a higher or lower value for $S^{\varepsilon}_{[CO2,LI]}$, but the fact that our refined approach gives a quantification of $S^{\varepsilon}_{[CO2,LI]}$ that is much closer to the target, stresses the importance of including efficacy differences.

**Constant $\varepsilon_{[LI]}$:** The authors have talked a lot about the state dependency of $S^{\varepsilon}_{[CO2,LI]}$, but they have barely discussed the state dependency of $\varepsilon_{[LI]}$, which is the bread and butter of this paper. After all, $\varepsilon_{[LI]}$ will likely depend on state and it can be readily computed for either their numerical model or the paleo data using their equation (11) and therefore the variability can be assessed in the manuscript. The conclusion says "the assumption that the efficacy factor is indeed constant in time could be tested more rigorously using more sophisticated climate models", but it can be tested in this manuscript using the models and data they are already employing. Furthermore, in the absence of this analysis, the usage of LGM specific $\varepsilon_{[LI]}$ in calculations, and which is applied as a constant value to the entirety of the Pleistocene time series makes the analysis look very contrived. The reader does not know, if the results change a lot if $\varepsilon_{[LI]}$ is derived, from say MIS5 and then kept constant for the entire interval of analysis? So the range of changes in $\varepsilon_{[LI]}$ and the dependence of principle results on that should be included in the manuscript.

As explained in the answer to the previous general comment of the reviewer, the analysis of the CLIMBER-2 results gives a quantification of the error made by assuming the efficacy factor to be constant in time. This is now explained more clearly in the revised manuscript. CLIMBER-2 is, however, not the most advanced model around; the results are very linear (small synergy of the effects of land ice and $CO_2$ changes), and important long-term feedbacks such as dust and non-$CO_2$ greenhouse gas changes are ignored in the simulations we analyze. We therefore maintain the sentence stating that the assumption of a time-constant efficacy factor can be investigated more rigorously using results of more sophisticated models. We have moved this sentence to the section where we present and discuss the CLIMBER-2 results.

So far, we derived $\varepsilon_{[LI]}$ using data from the LGM, because this is a well-studied time slice, that we also use in the analysis of our proxy-inferred dataset. In principal, however, $\varepsilon_{[LI]}$ can be obtained using data from any moment in time. Preferably, the radiative forcing anomalies should be large to prevent outliers resulting from divisions by small numbers, making MIS5 a less suited candidate. Instead, we now include an extra analysis of the CLIMBER-2 results, where we obtain $\varepsilon_{[LI]}$ from the mean value of all glacial marine isotope stages of the past 810 kyr (MIS 2, 4, 6, 8, 10, 12, 14, 16, 18, and 20). We find an $\varepsilon_{[LI]}$ of $0.56 \pm 0.09$ and a corresponding PI $S^{\varepsilon}_{[CO2,LI]}$ of $0.73^{+0.06}_{-0.05}$ K W$^{-1}$ m$^2$.

**Section 3.3:** A big shortcoming of this manuscript is section 3.3 which is extremely convoluted and difficult to follow. For an otherwise relatively clearly written paper, this section seems to have been put together haphazardly without the attention to detail that makes the rest of the paper readily readable. Though I have made a couple of specific comments for this section further down in my review, in general I have not been able to follow this section at all and therefore have not been able to provided the quality of feedback that I would have liked. A careful re-writing of this section by the authors is required.

We understand from the comments of this reviewer and reviewer #2, that Sect. 3.3 was not as easily understandable as we had hoped upon submission of the manuscript. During the process of revising the manuscript, we have come to the realization that this section only served as a further illustration of the importance of the effect of land ice changes that is already found in Sect. 3.2. As such, it is not essential to the main storyline of our manuscript. To improve the clarity of the paper as a whole, we have therefore decided to remove it from the manuscript.

**Scientific comments**

1. The various sensitivities are quotes in two different units throughout the paper: K per doubling of $CO_2$ and K W$^{-1}$ m$^2$. While the authors have been generally very clear about the units and about converting between them, as is the case on page 8, I do encourage them to use only one unit throughout the paper. This helps a reader to quickly compare various numbers from across the paper without having to convert the units. Alternatively, the authors could quote all sensitivities in both units, example: "so and so sensitivity was found to be 1.66 K W$^{-1}$ m$^2$ or equivalently 5.6 K per doubling of $CO_2$" (similar to the last sentence in the conclusions section).

As we now explain in the method section, we express $S^a$ in K W$^{-1}$ m$^2$ and ECS in K per doubling. We now convert $S^{\varepsilon}_{[CO2,LI]}$ to both quantities, and quote them conjointly.

2. Sentence spanning lines 9–10 on page 3: I don't understand what is meant by this sentence, specifically by the part "it has been shown that simulations of models that have been integrated over a few centuries are not yet in equilibrium". Perhaps rephrasing this sentence could make it clearer.

We have removed this sentence from the manuscript, since it was not essential to - and therefore distracting from - the storyline.

3. Line 10 page 3: Regarding ECS the authors say "Another way to express" but no other way has been previously mentioned until that point in the article. The ECS has only been defined up to that point. I think it makes more sense to rephrase it as "One way to express".

This sentence has been rephrased. ECS is expressed in K per doubling, and $S^a$ in K $W^{-1}$ $m^2$. They relate to each other as: ECS = $S^a$ * 3.7 W $m^{-2}$. This has now been made clear in the revised manuscript.

4. Since the form of "f" is important for the rest of the paper, the authors should clearly articulate the motivations for f as given in equation 5.

Equation 5 is part of the PALAEOSENS approach that has been used so far in numerous publications, and as such is not an equation we propose here. This has been made clear in the revised manuscript by splitting the method section into three parts (see our answer to the first general comment of the reviewer). The idea of the PALAEOSENS approach was that the influence of long-term processes on global temperature is directly proportional to the radiative forcing perturbation they induce, as is now mentioned in the revised manuscript.

5. Last para, page 4: So is $S_{[CO2,LI]}$ to be considered as an estimate of $S^a$? Maybe the authors should clarify this explicitly. In the process of making this clarification the starting sentence of that paragraph will likely need to be modified to make the argument fit in seamlessly.

No, we obtain an estimate for $S^a$ by multiplying $S^{\varepsilon}_{[CO2,LI]}$ by 0.64. This has now been clarified in the method section.

6. In the first paragraph on page 5 the authors say that they take a "further simplifying step" to more easily compare "$S^{\varepsilon}_{[CO2,LI]}$ to other specific paleoclimates sensitivities $S^{\varepsilon}_{[CO2,X]}$ by unifying the dependent variable". But all they have done is move the specific dependent variable $\Delta R_{[X]}$ into the newly defined $CO_2$-equivalent temperature and which doesn't in any way free someone of the need to compute that forcing or to compute the efficacy factor. So I fail to see the simplification here (besides a notational one) but more importantly I fail to see the practical usefulness. For any given $S_{[CO2,LI]}$ by the time one has computed the $CO_2$-equivalent temperature, they might as well have just used equation 9.

We realize now that calling this step 'simplifying' was somewhat confusing. In the revised manuscript, we have made a separate subsection describing this small refinement, which serves to unify the dependent variable in cross-plots of radiative forcing and global temperature anomalies. This makes our calculated $S^{\varepsilon}_{[CO2,LI]}$ more readily comparable to other

specific paleoclimate sensitivities, where more and/or different long-term processes are considered. We describe the newly introduced variable now more accurately as the global temperature change (with respect to PI) stripped of the inferred influence of processes X ($\Delta T_{[-X]}$), in our case land ice changes ($\Delta T_{[-LI]}$).

7. First para, page 6: In the experiments OC, and OI, which as I understand are meant to assess the effects of land ice and $CO_2$ respectively, why are the orbital conditions also varied in conjunction? It seems that the authors answer this later on in the manuscript, at the beginning of section 3.1: "since the influence of orbital variations is very small". That comment should be moved closer to where these experiments OC and OI are discussed.

This comment has been moved to the description of the model data, as suggested by the reviewer.

8. Line 21, page 6: "ANICE was forced by northern hemisphere temperatures obtained…" Northern hemispheres temperature or temperature anomaly? I think it should be the anomaly.

ANICE was indeed forced by the anomaly. We thank the reviewer for this careful observation.

9. Page 6: regarding the discussion of the amplification factor for the Pliocene, new results coming from the revised paleo-geographic boundary conditions for PlioMIP2 (Kamae et al. 2016; Chandan and Peltier 2017; Hunter et al. 2019) that suggest that the amplification factor could have been larger. Models that were used in the previous PlioMIP and whose results were synthesized in Haywood et al. 2013 were consistently failing to produce the polar amplification that has been inferred from proxies. With the new results the polar amplification factor in the warm interval of the Pliocene is nearly the same as the amplification factor during the cold LGM. The authors should and cite the new papers add a comment/analysis regarding how the revised amplification factor for the warm interval affects their results.

In the revised manuscript, we have included an appendix, in which we analyze the same proxy-data inferred dataset, but using a constant polar amplification factor of 2.7 over the past 800 kyr ($\Delta T_{g2}$ in Köhler et al. 2015). This is in our opinion a very interesting addition to our manuscript, but it does not affect the main results qualitatively.

10. Lines 15-17 page 7: the authors say they are inferring $S^\varepsilon_{[CO2,LI]}$ or $S^\varepsilon_{[CO2]}$ here but I think a bit of additional comment is required to clarify the appearance of $\varepsilon$ in these sensitivities. These are after all inferred from experiment OC in which $\Delta R_{[LI]}$ is zero, so the meaning of land-ice radiative efficacy $\varepsilon$ is not strictly defined. This is probably hair-splitting over notation but I think it is best to be as clear as possible since the climate sensitivity literature is already overflowing with (sometimes sloppily used) notation.

As the reviewer rightly points out, in the case of $\Delta R_{[LI]}=0$, $\Delta R_{[LI]}$ and $\varepsilon_{[LI]}$ have no effect on $S^\varepsilon_{[CO2,LI]}$, so $S^\varepsilon_{[CO2,LI]} = S^\varepsilon_{[CO2]} = S_{[CO2,LI]} = S_{[CO2]}$, which is now indicated in the revised manuscript.

11. Line 1, page 8: "the new approach considering efficacies clearly leads to a more satisfactory result that the old approach." In the present form this sentence implies that for some reason the numerical value 0.74 is more satisfactory than the older value of 0.54. I am

not sure if that is defensible or even that the authors themselves meant to imply that. I think the authors meant to say something like "the new approach is more flexible/accommodating/physically accurate than the old approach". Please re-phrase this accordingly.

This has been rephrased, because calling the results 'more satisfactory' could let readers believe we have a certain preference for a lower or higher result, which we of course do not have. We meant to say the new result (0.72 K W$^{-1}$ m$^2$) is much closer to the target value of 0.74 K W$^{-1}$ m$^2$ than the result of the old approach, stressing the importance of including efficacy. This has been clarified in the revised manuscript.

12. Lines 13–15, page 8: The authors have presented two results which lead to opposite conclusions. This needs to be addressed here directly instead of referring the reader to another publication. While the issue may have been more thoroughly assessed in Köhler et al. 2018, a brief comment should also be provided here so that the reader grasps the discordance in the author's results at a bare-minimum level without having to read up another paper.

A brief explanation of this result has been included in the revised manuscript.

13. Line 9, page 8: For the calculation of $\varepsilon_{[LI]}$ using equation 11 please provide the values of $\Delta R_{[CO2]}$ and $\Delta R_{[LI]}$ at LGM that were used.

The LGM values ($\Delta R_{[CO2]}$ = -2.04 W m$^{-2}$ and $\Delta R_{[LI]}$ = -3.88 W m$^{-2}$) are now provided. Upon including them and redoing the calculations, we realized we made a small mistake in the calculation of $\varepsilon$ in the former section 3.2, and the corresponding $S^{\varepsilon}_{[CO2,LI]}$. This has been corrected in the revised manuscript. We thank the reviewer for letting us double-check our calculations.

14. Line 15, page 8: is the mean the value "of" years 20 and 22 kya or "between" those years?

The temporal resolution of this dataset is 2,000 years, so we have values for 20 and 22 kyr ago. In that sense, it is indeed the mean 'of' these times. This has now been clarified.

15. Line 16, page 8: "The specific paleo climate sensitivities we find here are generally higher than calculated by the old approach" But the new sensitivity calculated is 1.39 which is lower than that by the old approach which was 1.66.

The new sensitivity of 1.39 K W$^{-1}$ m$^2$ (revised to 1.45 K W$^{-1}$ m$^2$, see our answer to scientific comment #13) holds for the LGM, and should be compared to 0.93 K W$^{-1}$ m$^2$ obtained by the old approach. The PI sensitivity of 1.66 K W$^{-1}$ m$^2$ of the old approach should be compared to our new PI sensitivity of 2.45 K W$^{-1}$ m$^2$. This has now been clarified in the revised manuscript.

16. Line 12 page 9: "We correct the induced $\Delta T_{[CO2]}$ of all individual models for this ratio" I don't follow.

17. Line 15 page 9: At this point I am lost. Why are you doing that regression? What it the motivation? And are you subtracting the global value $\Delta T_{[CO2]}$ from $\Delta T_{NH}$?

18. The ECS given in Table 1 for the CCSM4 model is different from that usually cited. Bitz et al. 2012 using the NCAR-CCSM4 and recently Chandan and Peltier, 2018 using a related UofT-CCSM4 have deduced the ECS to be 3.2. The value in Table 1 is lower than that. Where did the authors get this from? Haywood et al. 2013 also use CCSM4 ECS (from Bitz et al) of 3.2. Do the numbers for the other models need to be checked as well?

19. The authors should cite all the original experiment design papers for the PMIP3 experiments listed in Table 1. This can be done readily by adding a new column to the table called "References".

Answer to points 16 to 19: Section 3.3 has been removed from the manuscript, see our answer to the fourth general comment of the reviewer.

20. The figure description for Figure 3 is completely wrong. It is talking about things that are not on the figure.

Figure 3 and its caption have been corrected.

**Technical comments**

We are very grateful for these technical comments by the reviewer. We have implemented all the suggestions, except where indicated.

1. Line 2 page 1:" to equilibrium"

This sentence has been rewritten completely.

2. Line 29 page 2: "are obtained from  various model setups"

This sentence has been rewritten completely.

3. Line 16 page 3: "In this case, the average global paleo temperature anomaly with respect to the pre-industrial (PI)  ($\Delta T_g$) is"
4. Line 17 page 3: "that are typically neglected in  climate simulations".
5. Lines 3-4 on page 4 incorporating the phrase "the calculated paleoclimate sensitivity" in the current form refers to some specific and as yet undefined sensitivity. It's best to rephrase it as "If, for instance, only the most important slow feedback in the climate system, namely radiative forcing anomalies induced by albedo changes due to land ice (LI) variability are taken into account, then one can correct $S^p$ to derive the following specific paleoclimate sensitivity."
6. The sentence on line 5, page 4, appears as a sharp interruption to the logic train before and after that sentence. It should instead be placed at the end of that paragraph and rephrased as " A synthesis of  estimates of $S_{[CO2,LI]}$  from both …."
7. Line 15 page 4: " because, e.g."
8. Line 18 page 4: "through efficacy factors ($\varepsilon_{[LI]}$), . This requires a reformulation"
9. Line 20 page 4: "to clearly distinguish  the sensitivities from  those of the PALAEOSENS project in which the radiative forcing of the different processes  were assigned identical efficacies."

This sentence has been rephrased to:

'This serves to clearly distinguish these newly-derived sensitivities from those of the PALAEOSENS project in which efficacy was not taken into account, implying that identical radiative forcing of different processes leads to identical temperature changes.'

10. Line 25 page 4: "by land ice changes ($\varepsilon_{[LI]}$), using  the following formulation which is based on, but modified from Hansen et al. (2005)"

11. Line 22 page 5: The sentence "CLIMBER-2 combined a 2.5 statistical-dynamical…" seems something is missing after 2.5. Did the authors mean "2.5 degree"?

Corrected to '… 2.5-dimensional …'

12. Line 5 page 5: Add comma after "Similarly"

Corrected to 'As before, …'

13. Line 3 page 5: "leaving 217 data points as indicated in Fig 1c,d."

14. Sentence beginning on line 18 page 7: change it to something like "For our first attempt at compensating paleoclimates sensitivity for slow processes other than $CO_2$ changes we strive to deduce the same $S^\varepsilon_{[CO2,LI]}$, inferred above, from experiment OIC in which both $CO_2$ and land ice cover vary over time".

This sentence has been rewritten as:

'Now, we apply our approach to the results of experiment OIC, in which both $CO_2$ and land ice cover vary over time, with the aim of deducing the sole effect of $CO_2$ changes on global temperature.'

15. Line 23 page 7: "Between ….  some  outliers resulted from division  by small numbers (not shown on Fig. 2b)."

16. Line 29 page 7: "….is more linear than that  between …"

17. Line 31 page 7 " through the entire 5 million year interval."

18. Line 11 page 8: " Similar to Köhler et al., 2018, we too detect"

19. Line 15 page 8: the value of $\Delta R_{[CO2]}$ should be -2.04

20. Line 15 page 8: "the LGM value ( taken here as the mean…)"

21. Line 21 page 8: "we first scale  it by a factor"

This sentence has been removed.

22. Line 23 page 8: "Note that this scaling still assumes unit efficacy for  process other than land ice changes"

This sentence has been corrected as suggested by the reviewer, and replaced to the method section.

23. Line 24 page 8: "Then, after  multiplying by"

This sentence has been removed.

24. Line 24 page 8: Units should be $Wm^{-2}$

This sentence has been removed.

25. Line 11 page 9: "that the ratio of the radiative forcing change  between the LGM (185 ppm $CO_2$) and the PI (280 ppm $CO_2$), to the change between the PI and  a 2 X PI case is
26. Line 16 page 9: "significant  at the 95% level"

Answer to points 25 and 26: Section 3.3 has been removed from the manuscript, see our answer to the fourth general comment of the reviewer.

27. Conclusions section, Lines 26, 28, 30: $\varepsilon_{[CO2,LI]}$ is a new symbol not previously defined. It seems like a mistake and the authors likely meant $\varepsilon_{[LI]}$
28. The yellow star in Fig 4 is barely visible against the cyan background. Please change it to something dark, maybe black.

All the colors in Fig. 4 have been changed for better visibility.

REFERENCES:
Forster, P. M., Andrews, T., Good, P., Gregory, J. M., Jackson, L. S., and Zelinka, M.: Evaluating adjusted forcing and model spread for historical and future scenarios in the CMIP5 generation of climate models, Journal of Geophysical Research: Atmospheres, 118, 1139–1150, 2013.

Haywood, A. M., Hill, D. J., Dolan, A. M., Otto-Bliesner, B. L., Bragg, F., Chan, W.-L., Chandler, M. A., Contoux, C., Dowsett, H. J., Jost, A., et al.: Large-scale features of Pliocene climate: results from the Pliocene Model Intercomparison Project, Climate of the Past, 9, 191–209, 2013.

Hansen, J., Sato, M. K. I., Ruedy, R., Nazarenko, L., Lacis, A., Schmidt, G. A., Russell, G., Aleinov, I., Bauer, M., Bauer, S., et al.: Efficacy of climate forcings, Journal of Geophysical Research: Atmospheres, 110, 2005.

Köhler, P., Bintanja, R., Fischer, H., Joos, F., Knutti, R., Lohmann, G., and Masson-Delmotte, V.: What caused Earth's temperature variations during the last 800,000 years? Data-based evidences on radiative forcing and constraints on climate sensitivity, Quaternary Science Reviews, 29, 129–145, https://doi.org/10.1016/j.quascirev.2009.09.026, 2010.

Köhler, P., de Boer, B., von der Heydt, A. S., Stap, L. B., and van de Wal, R. S. W.: On the state-dependency of the equilibrium climate sensitivity during the last 5 million years, Climate of the Past, 11, 1801–1823, 2015.

Stap, L. B., van de Wal, R. S. W., de Boer, B., Köhler, P., Hoencamp, J. H., Lohmann, G., Tuenter, E., and Lourens, L. J.: Modeled influence of land ice and $CO_2$ on polar amplification and paleoclimate sensitivity during the past 5 million years, Paleoceanography and Paleoclimatology, 33, 381–394, 2018.

**Reviewer #2**

My initial thoughts on seeing this paper were very positive in the sense that, given the uncertainty of information in the paleorecord, and the difficulty of using state of the art models to make very long runs, all progress in the area of better defining climate sensitivity as it relates to past climates is worthwhile.

My optimism remained through the first parts of the paper, but by the end I have to admit that I am lost and really do not understand what the authors are trying to do and what they have discovered.

We thank the reviewer for considering our work. We are pleased the reviewer sees merit in the aim of our study. Along the helpful comments provided, we have thoroughly rewritten and restructured the manuscript to get our message across more clearly.

Most importantly we have improved the readability of the manuscript in the following manners:
- We have removed Section 3.3 from the manuscript, because this section only served as a further illustration of the importance of the effect of land ice changes that is already found in Sect. 3.2. As such, it is not essential to the main storyline of our manuscript.
- We have included a brief introduction to the method and results sections, in which we explain the aim of the section. The results sections end with a statement and discussion of the gained insights.
- We have split Sect. 2.1 (the method section) into three parts, describing 1) the PALAEOSENS approach used so far in earlier studies, 2) our main refinement: the inclusion of the efficacy of land ice changes, and 3) a small refinement that unifies the dependent variable in cross-plots of radiative forcing and global temperature anomalies.
- We have relocated Sects. 2.2.1 and 2.2.2, so that first the modelling results are introduced and analyzed straight away, and thereafter the proxy-inferred dataset is introduced and analyzed.
- We have renamed the variable 'CO2-equivalent temperature change' ($\Delta T_{[CO2-equiv]}$). In the revised manuscript, we have more accurately named it 'the global temperature change (with respect to PI) stripped of the influence of land ice changes ($\Delta T_{[-LI]}$)'.

The authors introduce a variable, DTe[CO2 - equiv] but do not explain why this is useful or interesting.

The introduction of this variable, named $\Delta T_{[-LI]}$ in the revised manuscript (see below), serves to unify the dependent variable in cross-plots of radiative forcing and global temperature anomalies. This makes our calculated $S^\varepsilon_{[CO2,LI]}$ more readily comparable to other specific paleoclimate sensitivities, where more and/or different long-term processes are considered. This step is a small refinement compared to our main refinement of including the efficacy of land ice changes. This has been clarified in the revised manuscript by splitting the method section into three parts.

What I would have done is take equation (9), replace X with LI and then explore all the elements of that equation. This would show us how S varies with DRLI and DRCO2, as well a DTg and one could consider how much of the state+forcing+efficacy dependence of S[CO2] is accounted for by considering land ice with and without considering efficacy.

This has been done in detail in Köhler et al. (2010) for the old approach (equivalent to $\varepsilon_{[LI]} = 1$ in the refined approach). The inclusion of an efficacy factor for land ice changes does not qualitatively change this analysis, it just linearly amplifies (when $\varepsilon_{[LI]} > 1$) or diminishes (when $\varepsilon_{[LI]} < 1$) the effect of radiative forcing by land ice changes. We therefore focus directly on the effect of $\varepsilon_{[LI]}$ on $S^{\varepsilon}_{[CO2,LI]}$.

I can see that Figures 2 and 3 represent some kind of sensitivity-like variable, but I cannot grasp its meaning.

Indeed, in Figures 2 and 3 we showed the main results of our manuscript: the influence of the deduced $\varepsilon_{[LI]}$ on $S^{\varepsilon}_{[CO2,LI]}$.

Basically, DTe[CO2 - equiv] is not, as you suggest in equation 14 simply a function of DRCO2 but also depends on Tg and DRLI.

What we meant here is that we make a regression to the scattered data of $\Delta R_{[CO2]}$ and the variable DTe[CO2 - equiv] (now called $\Delta T_{[-LI]}$). $\Delta T_{[-LI]}$ comprises the influences of $\Delta R_{[LI]}$, $\Delta R_{[CO2]}$ and $\Delta T_g$. To clarify this, in the revised manuscript we have named this function *regfunc* (instead of *g*).

I hope that the remedy is a better explanation of the reasons behind the derivations in section 1 and also better explanation of the insight that you gain from the results.

Other points.

P1L10 "Recently, it has been shown that simulations of models that have been integrated over a few centuries are not yet in equilibrium, and from longer climate simulations a higher ECS can be deduced (Knutti et al., 2017)."

This needs rephrasing. It has been well known since before dynamical oceans were included in climate models that the equilibrium time of the ocean is of the order of thousands of years. Since the invention of the AOGCM, ad-hoc methods have been introduced to try to estimate equilibrium climate sensitivity without running the models to equilibrium. What recent work has been doing is assessing the accuracy of such approximations.

We have removed this sentence from the manuscript, since it was not essential to - and therefore distracting from - the storyline.

P2L23 "likewise as several earlier studies"
-> "as in several earlier studies"

This sentence has been removed.

Eq(10) This equation suggests to me that DTg-DT[LI]=DT[CO2]. Maybe I misunderstand, but it seems to me that DTg=DT[LI]+DT[CO2]+DT[X]+Z, where DT[X] is the influence of all the other forcings and Z represents cross terms (ie nonlinearities).

In this manuscript, we aim to calculate specific climate sensitivity $S^\varepsilon_{[CO2,LI]}$, which only compensates paleoclimate sensitivity ($S^p$) for the influence of land ice cover changes. To deduce the efficacy of land ice changes, we relate its effect on global temperature changes to that of all other processes combined. This is in line with our calculation of $S^a$ from $S^\varepsilon_{[CO2,LI]}$ by multiplying by a factor of 0.64, which implies unit efficacy for all other processes than land ice changes. As stated in the text, this is a source of uncertainty to be investigated in future research.

P3L5 "Similarly as in the old approach," Not English

Corrected to: 'As before, …'

Eq(13) Looks like a minus sign between "CO2" and "equiv".

We realize that calling this variable 'CO2-equivalent temperature change' ($\Delta T_{[CO2\text{-}equiv]}$) was confusing. In the revised manuscript we have therefore more accurately named it 'the global temperature change (with respect to PI) stripped of the influence of land ice changes ($\Delta T_{[\text{-}LI]}$)'.

P5L12 "A functional relationship between TE[CO2–equiv] and R[CO2] (T[CO2–equiv]= g(R[CO2])) can be obtained by least squares regressions of higher-order polynomial to the scattered data of these variables."

It is not clear which variables are "these variables".

This sentence has been rephrased as:
'Now, we quantify $S^\varepsilon_{[CO2,LI]}$ by performing a least-squares regression (regfunc) through scattered data from $\Delta T^\varepsilon_{[\text{-}LI]}$ and $\Delta R_{[CO2]}$.'

Sections 2 and 3

I think the paper order should be 2.2.1, 3.1 then 2.2.2, 3.2. The way it is presented is just confusing. Present the whole of the simple modelling case and then move on to the data-based case.

We have followed the suggestion of the reviewer.

P7L22 "and again fit a second order polynomial to the scattered data of T" [CO2–equiv]"

Which experiment?

Here, we analyse the results of experiment OIC. We have clarified this in the revised manuscript.

P8L12 "Similarly as before" Not English

This sentence has been corrected.

Table 1 State which paper each "published ECS" comes from.

Section 3.3 has been removed from the manuscript, see our answer to the comment of the reviewer below.

I prefer to write reviews before reading what other reviewers have posted, as I feel I will be too easily influenced, so I did not read the other reviewer's comment until now. I am encouraged to see that the other reviewer also found the paper very difficult to follow. This increases my optimism that there is hope that with better explanation in critical areas, and reorganisation to improve the storyline, that the paper may become both comprehensible and publishable.

Reviewer #1 was mostly concerned about Sect. 3.3. During the process of revising the manuscript, we have come to the realization that this section only served as a further illustration of the importance of the effect of land ice changes that is already found in Sect. 3.2. As such, it is not essential to the main storyline of our manuscript. To improve the clarity of the paper as a whole, we have therefore decided to remove it from the manuscript.

---

## Author Response (AR1)

ALFRED-WEGENER-INSTITUT HELMHOLTZ-ZENTRUM FÜR POLAR-UND MEERESFORSCHUNG

Alfred Wegener Institute, PO Box 12 01 61, 27515 Bremerhaven, Germany

То

Dr. Daniel Kirk-Davidoff Handling Editor of Earth System Dynamics

14 March 2019 Subject: Resubmission of manuscript #esd-2018-88

**Dear Dr. Kirk-Davidoff,**

We wish to resubmit our manuscript entitled **'Including the efficacy of land ice changes in deriving climate sensitivity from paleodata'**, co-authored by L.B. Stap, P. Köhler and G. Lohmann, for consideration of publication in *Earth System Dynamics*. The manuscript represents a significant revision of our earlier submitted manuscript of the same name (MS No.: esd-2018-88) along the constructive reviewer comments.

Following the reviewers and your suggestions and to improve the readability of the manuscript, we have made the following major changes:

- We have removed Section 3.3 from the manuscript, because this section only served as a further illustration of the importance of the effect of land ice changes that is already found in Sect. 3.2. As such, it is not essential to the main storyline of our manuscript.
- We have included a brief introduction to the method and results sections, in which we explain the aim of each section. The results sections end with a statement and discussion of the gained insights.
- We have split Sect. 2.1 (the method section) into three parts, describing 1) the PALAEOSENS approach used so far in earlier studies, 2) our main refinement: the inclusion of the efficacy of land ice changes, and 3) a small refinement that unifies the dependent variable in cross-plots of radiative forcing and global temperature anomalies.
- We have relocated Sects. 2.2.1 and 2.2.2, so that first the modelling results are introduced and analyzed straight away, and thereafter the proxy-inferred dataset is introduced and analyzed.
- We have renamed the variable 'CO2-equivalent temperature change'  $(\Delta T_{[CO2-equiv]})$ . In the revised manuscript, we have more accurately named it 'the global temperature change (with respect to PI) stripped of the influence of land ice changes  $(\Delta T_{I-LII})$ '.

Dr. Lennert B. Stap Postdoctoral researcher Bussestraße 24 27570 Bremerhaven +49(471)4831-1721 lennert.stap@awi.de

Alfred Wegener Institute Helmholtz Centre for Polar and Marine Research

BREMERHAVEN

Am Handelshafen 12 27570 Bremerhaven Germany Phone +49 471 4831-0 Fax +49 471 4831-1149 www.awi.de

Public law institution

Head Office: Am Handelshafen 12 27570 Bremerhaven Germany Phone +49 471 4831-0 Fax +49 471 4831-1149 www.awi.de

Board of Governors: MinDir Dr. Karl Eugen Huthmacher Board of Directors: Prof. Dr. Antje Boetius (Director) Dr. Karsten Wurr (Administrative Director) Dr. Uwe Nixdorf (Vice Director) Prof. Dr. Karen H. Wiltshire (Vice Director)

Bank account: Commerzbank AG, Bremerhaven BIC/Swift COBADEFF292 IBAN DE12292400240349192500 Tax-Id-No. DE 114707273

ALFRED-WEGENER-INSTITUT HELMHOLTZ-ZENTRUM FÜR POLAR-UND MEERESFORSCHUNG

- In the last paragraph of Sect. 2.2, we discuss the alternative formulation of the efficacy factor suggested by reviewer #1, and why we opt for our formulation. At your suggestion, this includes a significance test for the (linear vs. non-linear) relation between  $\Delta R_{CO2}$  and  $\Delta R_{LI}$ .
- We have added an appendix, in which we analyze the same proxydata inferred dataset, but using a constant polar amplification factor of 2.7 over the past 800 kyr.

We submit a color-coded revised manuscript, as well as a detailed point-bypoint response to the comments of the reviewers. If you have any further questions, please do not hesitate to contact us.

Yours sincerely,

Lennert Stap

'Including the efficacy of land ice changes in deriving climate sensitivity from paleodata' by L.B. Stap, P. Köhler and G. Lohmann. Submitted for potential publication by Earth System Dynamics

**REPLY TO THE COMMENTS BY THE REVIEWERS**

Color coding: Black – comments by reviewers Green – reply by authors

**Reviewer #1**

In this paper the authors try to address a matter of importance — that concerning the efficacy of the different radiative forcing — and which is directly relevant to the ongoing efforts by various modelling and proxy analysis groups to estimate the planet's Equilibrium Climate Sensitivity (ECS). I like the idea of the paper and I am quite sure the paper will be accepted, but I feel there is need for clarity and additional analysis before the paper is in publishable form.

We thank the reviewer for a careful examination of our work. We are pleased that the reviewer likes the idea of the paper. In the revised manuscript, we have largely followed the provided comments to improve the clarity of the paper, and we have included additional analysis, as described below.

**Points of broadest significance**

**Definition of the efficacy factor:** This paper builds upon the work by Hansen et al. 2005 and by PALEOSENS members (2012), but the way the authors introduce the efficacy factor in equations (8) and (9) is different from those employed in these other works. For example, according to the PALEOSENS approach, equation (9) should be expressed as (See sample calculation in PALEOSENS supplementary materials section B.2):

$$S_{[CO2,LI]}^{\varepsilon} = \frac{\Delta T_g}{\Delta R_{CO2} + \Delta R_{LI}} = \frac{\Delta T_g}{\Delta R_{CO2} + \varepsilon_{LI} \Delta R_{CO2}}$$

This says that the efficacy of the radiative forcing from land ice changes,  $\Delta R_{[LI]}$  is related to the equivalent radiative forcing from changes in CO2 through is a fractional parameter  $\varepsilon_{[LI]}$ . This is what the efficacy is meant to serve: to help assess the radiative forcing from non-greenhouse gas sources by relating it to the better constrained forcing from CO2. But the way the authors are using  $\varepsilon_{[X]}$  is quite strange and it doesn't make sense to me. It doesn't appear to be a typographical mistake. The climate sensitivity world is already overflowing with numerous different formulations and I think there should be a very good reason (and which should be made extremely clear in the paper) to define an existing concept differently.

We would like to argue that the way we have implemented the efficacy factor in our approach is the most natural extension to (our) earlier studies. Indeed, the PALAEOSENS approach, which we have used so far, employs:

$$S_{[CO2,LI]} = \frac{\Delta T_g}{\Delta R_{[CO2]} + \Delta R_{[LI]}}$$

Mind though: no superscript  $\varepsilon$  here, because efficacy differences are not considered.

In Köhler et al. (2010), radiative forcing records over the past 800 kyr of many different processes including  $CO_2$  and land ice changes, were analyzed. This is not the issue we consider in this manuscript. Instead, we try to overcome the problem that the strength of the response of global-average temperature to global-average radiative forcing can be different, depending on the generating process (in this case land ice changes or  $CO_2$  changes), so in general:

$$\frac{\Delta T_{[CO2]}}{\Delta R_{[CO2]}} \neq \frac{\Delta T_{[LI]}}{\Delta R_{[LI]}}.$$

In our opinion, the most logical approach to include this difference in efficacy is to multiply the radiative forcing of land ice changes by an appropriate factor so that the strength of the temperature response is the same as when  $CO_2$  would be the generating process. Indeed, Hansen et al. (2005) compared the effects of several different processes, expressing the efficacy of these processes X as:

$$\varepsilon_{[X]} = \frac{\Delta T_{[X]} / \Delta R_{[X]}}{\Delta T_{[CO2]} / \Delta R_{[CO2]}}, \text{ so in our case: } \varepsilon_{[LI]} = \frac{\Delta T_{[LI]} / \Delta R_{[LI]}}{\Delta T_{[CO2]} / \Delta R_{[CO2]}}.$$

Our implementation,

$$\frac{\Delta T_{[LI]}}{\Delta R_{[LI]}} = \varepsilon_{[LI]} \frac{\Delta T_g - \Delta T_{[LI]}}{\Delta R_{[CO2]}},$$

follows this approach very closely. The only difference is that we relate the effect of land ice changes (left hand side) to the effect of all processes except land ice changes (right hand side), because we calculate specific climate sensitivity  $S^{\varepsilon}_{[CO2,LI]}$ , which does not account for the effect of these other processes.

In principal, it is also possible to relate the impact of land ice changes on global temperature directly to  $\Delta R_{[CO2]}$ , as the reviewer proposes:

$$S_{[CO2,LI],\text{alt}}^{\varepsilon} = \frac{\Delta I_g}{\Delta R_{[CO2]} + \varepsilon_{[LI],\text{alt}} \Delta R_{[CO2]}},$$

where the efficacy factor in this alternative case ( $\epsilon_{[LI],alt}$ ) relates to the one used in our approach ( $\epsilon_{[LI]}$ ) as:

$$\varepsilon_{[LI],alt} = \varepsilon_{[LI]} \frac{\Delta R_{[LI]}}{\Delta R_{[CO2]}}$$

However, from the records of  $\Delta R_{[CO2]}$  and  $\Delta R_{[LI]}$  of our dataset, we infer a non-linear relationship between these two quantities (see the figure below, included as the new Fig. 2 in the revised manuscript). This would introduce a cumbersome state dependency of  $\varepsilon_{[LI]}$ , which is avoided by our approach. This has now been elaborated upon in the revised manuscript.

Figure: Relation between radiative forcing anomalies caused by  $CO_2$  changes ( $\Delta R_{[CO2]}$ ) and land ice changes ( $\Delta R_{[LI]}$ ) from the proxy-inferred dataset (pink dots). The red line represents a second order polynomial least-squares regression through the scattered data.

Furthermore, for clarification we have split the method section (Section 2.1) into three parts in the revised manuscript, describing 1) the PALAEOSENS approach used in earlier studies, 2) our main refinement: the inclusion of the efficacy of land ice changes, and 3) a small refinement that unifies the dependent variable in cross-plots of radiative forcing and global temperature anomalies.

**Regarding the sample calculations from CLIMBER experiment:** The authors try to apply their new formulation to compute  $S^{\epsilon}_{[CO2,LI]}$  from their CLIMBER data and compare it to  $S_{[CO2,LI]}$  that they have previously found. Using

$$S_{[CO2,LI]} = \frac{\Delta T_g}{\Delta R_{[CO2]} + \Delta R_{[LI]}}$$

the authors found  $S_{[CO2,LI]}$  to be 0.54. This formulation uses  $\Delta T_g$ ,  $\Delta R_{[CO2]}$  and  $\Delta R_{[LI]}$  all of which are available from their CLIMBER models (and shown in Fig 1). Their new formulation  $S^{\epsilon}_{[CO2,LI]}$ , after substituting for  $\epsilon_{[LI]}\Delta R_{[LI]}$  from equation (11) into equation (9) reduces to

$$S_{[CO2,LI]}^{\varepsilon} = \frac{\Delta T_g - \Delta T_{LI}}{\Delta R_{CO2}}$$

in which all the terms are again derived from their CLIMBER models, the only difference from the original expression is that instead of  $\Delta R_{[LI]}$  the new expression uses  $\Delta T_{[LI]}$ . I am quite confused why the new approach using temperature from land ice changes, instead of radiative forcing due to land ice changes, (both from the same set of models), and leading to a higher inferences of S is to be favoured (a sentiment expressed at the start of page 8)?

The goal of this section is to validate our refined approach by applying it to the idealized CLIMBER-2 simulations. Here, the effect of  $CO_2$  is a-priori known from the results of experiment OC:

$$S_{[CO2,LI]}^{\varepsilon} = \frac{\Delta T_{OC}}{\Delta R_{[CO2]}}.$$

This result functions as the target for our approach of obtaining the sole effect of  $CO_2$  changes on global temperature from the results of experiment OIC, where land ice cover and  $CO_2$  levels are both varied over time. Our refined approach considers the efficacy of land ice changes:

$$S_{[CO2,LI]}^{\varepsilon} = \frac{\Delta T_g}{\Delta R_{[CO2]} + \varepsilon_{[LI]} \Delta R_{[LI]}}$$

We calculate the efficacy factor  $\epsilon_{\text{[LI]}}$  as:

$$\varepsilon_{[LI]} = \frac{\omega}{1-\omega} \frac{\Delta R_{[CO2]}}{\Delta R_{[LI]}},$$

where

$$\omega = \frac{\Delta T_{[LI]}}{\Delta T_g} \bigg|_{\text{spec. time}}.$$

Note here that the parameter  $\omega$  is obtained from temperatures at a specific time (for instance, the LGM), constituting the assumption that  $\varepsilon_{[LI]}$  is constant in time. Therefore, the simplification that the reviewer makes by substituting equation (11) into equation (9) is not generally valid. Otherwise, the refined approach would indeed by construction always yield the target value for  $S_{[CO2,LI]}^{\varepsilon}$  (apart from a negligible contribution by the synergy of CO2 and land ice changes). Instead,  $S_{[CO2,LI]}^{\varepsilon}$  is only matched by construction at the LGM. The comparison we make between our approach and the target, provides a quantification of the error yielded by assuming a time-invariant  $\varepsilon_{[LI]}$ , which has been clarified in the revised manuscript (see also our answer to the next general comment).

We do not favour a higher or lower value for  $S^{\varepsilon}_{[CO2,LI]}$ , but the fact that our refined approach gives a quantification of  $S^{\varepsilon}_{[CO2,LI]}$  that is much closer to the target, stresses the importance of including efficacy differences.

**Constant**  $\epsilon_{[LI]}$ : The authors have talked a lot about the state dependency of  $S^{\epsilon}_{[CO2,LI]}$ , but they have barely discussed the state dependency of  $\epsilon_{[LI]}$ , which is the bread and butter of this paper. After all,  $\epsilon_{[LI]}$  will likely depend on state and it can be readily computed for either their numerical model or the paleo data using their equation (11) and therefore the variability can be assessed in the manuscript. The conclusion says "the assumption that the efficacy factor is indeed constant in time could be tested more rigorously using more sophisticated climate models", but it can be tested in this manuscript using the models and data they are already employing. Furthermore, in the absence of this analysis, the usage of LGM specific  $\epsilon_{[LI]}$  in calculations, and which is applied as a constant value to the entirety of the Pleistocene time series makes the analysis look very contrived. The reader does not know, if the results change a lot if  $\epsilon_{[LI]}$  is derived, from say MIS5 and then kept constant for the entire interval of analysis? So the range of changes in  $\epsilon_{[LI]}$  and the dependence of principle results on that should be included in the manuscript.

As explained in the answer to the previous general comment of the reviewer, the analysis of the CLIMBER-2 results gives a quantification of the error made by assuming the efficacy factor to be constant in time. This is now explained more clearly in the revised manuscript. CLIMBER-2 is, however, not the most advanced model around; the results are very linear (small synergy of the effects of land ice and CO2 changes), and important long-term feedbacks such as dust and non-CO2 greenhouse gas changes are ignored in the simulations we analyze. We therefore maintain the sentence stating that the assumption of a time-constant efficacy factor can be investigated more rigorously using results of more sophisticated models. We have moved this sentence to the section where we present and discuss the CLIMBER-2 results.

So far, we derived  $\varepsilon_{[LI]}$  using data from the LGM, because this is a well-studied time slice, that we also use in the analysis of our proxy-inferred dataset. In principal, however,  $\varepsilon_{[LI]}$  can be obtained using data from any moment in time. Preferably, the radiative forcing anomalies should be large to prevent outliers resulting from divisions by small numbers, making MIS5 a less suited candidate. Instead, we now include an extra analysis of the CLIMBER-2 results, where we obtain  $\varepsilon_{[LI]}$  from the mean value of all glacial marine isotope stages of the past 810 kyr (MIS 2, 4, 6, 8, 10, 12, 14, 16, 18, and 20). We find an  $\varepsilon_{[LI]}$  of 0.56 ± 0.09 and a corresponding PI  $S_{[CO2,LI]}^{\varepsilon}$  of  $0.73^{+0.06}_{-0.05}$  K W-1 m2.

**Section 3.3:** A big shortcoming of this manuscript is section 3.3 which is extremely convoluted and difficult to follow. For an otherwise relatively clearly written paper, this section seems to have been put together haphazardly without the attention to detail that makes the rest of the paper readily readable. Though I have made a couple of specific comments for this section further down in my review, in general I have not been able to follow this section at all and therefore have not been able to provided the quality of feedback that I would have liked. A careful re-writing of this section by the authors is required.

We understand from the comments of this reviewer and reviewer #2, that Sect. 3.3 was not as easily understandable as we had hoped upon submission of the manuscript. During the process of revising the manuscript, we have come to the realization that this section only served as a further illustration of the importance of the effect of land ice changes that is already found in Sect. 3.2. As such, it is not essential to the main storyline of our manuscript. To improve the clarity of the paper as a whole, we have therefore decided to remove it from the manuscript.

**Scientific comments**

1. The various sensitivities are quotes in two different units throughout the paper: K per doubling of  $CO_2$  and K W-1 m2. While the authors have been generally very clear about the units and about converting between them, as is the case on page 8, I do encourage them to use only one unit throughout the paper. This helps a reader to quickly compare various numbers from across the paper without having to convert the units. Alternatively, the authors could quote all sensitivities in both units, example: "so and so sensitivity was found to be 1.66 K W-1 m2 or equivalently 5.6 K per doubling of  $CO_2$ " (similar to the last sentence in the conclusions section).

As we now explain in the method section, we express  $S^a$  in K W-1 m2 and ECS in K per doubling. We now convert  $S^{\varepsilon}_{[CO2,LI]}$  to both quantities, and quote them conjointly. 2. Sentence spanning lines 9–10 on page 3: I don't understand what is meant by this sentence, specifically by the part "it has been shown that simulations of models that have been integrated over a few centuries are not yet in equilibrium". Perhaps rephrasing this sentence could make it clearer.

We have removed this sentence from the manuscript, since it was not essential to - and therefore distracting from - the storyline.

3. Line 10 page 3: Regarding ECS the authors say "Another way to express" but no other way has been previously mentioned until that point in the article. The ECS has only been defined up to that point. I think it makes more sense to rephrase it as "One way to express".

This sentence has been rephrased. ECS is expressed in K per doubling, and Sa in K W-1 m2. They relate to each other as: ECS = Sa \* 3.7 W m-2. This has now been made clear in the revised manuscript.

4. Since the form of "f" is important for the rest of the paper, the authors should clearly articulate the motivations for f as given in equation 5.

Equation 5 is part of the PALAEOSENS approach that has been used so far in numerous publications, and as such is not an equation we propose here. This has been made clear in the revised manuscript by splitting the method section into three parts (see our answer to the first general comment of the reviewer). The idea of the PALAEOSENS approach was that the influence of long-term processes on global temperature is directly proportional to the radiative forcing perturbation they induce, as is now mentioned in the revised manuscript.

5. Last para, page 4: So is  $S_{[CO2,LI]}$  to be considered as an estimate of Sa? Maybe the authors should clarify this explicitly. In the process of making this clarification the starting sentence of that paragraph will likely need to be modified to make the argument fit in seamlessly.

No, we obtain an estimate for Sa by multiplying  $S^{\varepsilon}_{[CO2,LI]}$  by 0.64. This has now been clarified in the method section.

6. In the first paragraph on page 5 the authors say that they take a "further simplifying step" to more easily compare "Sɛ[CO2,LI] to other specific paleoclimates sensitivities Sɛ[CO2,X] by unifying the dependent variable". But all they have done is move the specific dependent variable  $\Delta R_{[X]}$  into the newly defined CO2-equivalent temperature and which doesn't in any way free someone of the need to compute that forcing or to compute the efficacy factor. So I fail to see the simplification here (besides a notational one) but more importantly I fail to see the practical usefulness. For any given S[CO2,LI] by the time one has computed the CO2-equivalent temperature, they might as well have just used equation 9.

We realize now that calling this step 'simplifying' was somewhat confusing. In the revised manuscript, we have made a separate subsection describing this small refinement, which serves to unify the dependent variable in cross-plots of radiative forcing and global temperature anomalies. This makes our calculated  $S_{[CO2,LI]}^{\varepsilon}$  more readily comparable to other

specific paleoclimate sensitivities, where more and/or different long-term processes are considered. We describe the newly introduced variable now more accurately as the global temperature change (with respect to PI) stripped of the inferred influence of processes X ( $\Delta T_{[-X]}$ ), in our case land ice changes ( $\Delta T_{[-LI]}$ ).

7. First para, page 6: In the experiments OC, and OI, which as I understand are meant to assess the effects of land ice and  $CO_2$  respectively, why are the orbital conditions also varied in conjunction? It seems that the authors answer this later on in the manuscript, at the beginning of section 3.1: "since the influence of orbital variations is very small". That comment should be moved closer to where these experiments OC and OI are discussed.

This comment has been moved to the description of the model data, as suggested by the reviewer.

8. Line 21, page 6: "ANICE was forced by northern hemisphere temperatures obtained..." Northern hemispheres temperature or temperature anomaly? I think it should be the anomaly.

ANICE was indeed forced by the anomaly. We thank the reviewer for this careful observation.

9. Page 6: regarding the discussion of the amplification factor for the Pliocene, new results coming from the revised paleo-geographic boundary conditions for PlioMIP2 (Kamae et al. 2016; Chandan and Peltier 2017; Hunter et al. 2019) that suggest that the amplification factor could have been larger. Models that were used in the previous PlioMIP and whose results were synthesized in Haywood et al. 2013 were consistently failing to produce the polar amplification that has been inferred from proxies. With the new results the polar amplification factor during the cold LGM. The authors should and cite the new papers add a comment/analysis regarding how the revised amplification factor for the warm interval affects their results.

In the revised manuscript, we have included an appendix, in which we analyze the same proxydata inferred dataset, but using a constant polar amplification factor of 2.7 over the past 800 kyr ( $\Delta T_{g2}$  in Köhler et al. 2015). This is in our opinion a very interesting addition to our manuscript, but it does not affect the main results qualitatively.

10. Lines 15-17 page 7: the authors say they are inferring  $S^{\epsilon}_{[CO2,LI]}$  or  $S^{\epsilon}_{[CO2]}$  here but I think a bit of additional comment is required to clarify the appearance of  $\epsilon$  in these sensitivities. These are after all inferred from experiment OC in which  $\Delta R_{[LI]}$  is zero, so the meaning of land-ice radiative efficacy  $\epsilon$  is not strictly defined. This is probably hair-splitting over notation but I think it is best to be as clear as possible since the climate sensitivity literature is already overflowing with (sometimes sloppily used) notation.

As the reviewer rightly points out, in the case of  $\Delta R_{[LI]}=0$ ,  $\Delta R_{[LI]}$  and  $\varepsilon_{[LI]}$  have no effect on  $S^{\varepsilon}_{[CO2,LI]}$ , so  $S^{\varepsilon}_{[CO2,LI]} = S^{\varepsilon}_{[CO2]} = S_{[CO2,LI]} = S_{[CO2]}$ , which is now indicated in the revised manuscript.

11. Line 1, page 8: "the new approach considering efficacies clearly leads to a more satisfactory result that the old approach." In the present form this sentence implies that for some reason the numerical value 0.74 is more satisfactory than the older value of 0.54. I am

not sure if that is defensible or even that the authors themselves meant to imply that. I think the authors meant to say something like "the new approach is more flexible/accommodating/physically accurate than the old approach". Please re-phrase this accordingly.

This has been rephrased, because calling the results 'more satisfactory' could let readers believe we have a certain preference for a lower or higher result, which we of course do not have. We meant to say the new result (0.72 K W-1 m2) is much closer to the target value of 0.74 K W-1 m2 than the result of the old approach, stressing the importance of including efficacy. This has been clarified in the revised manuscript.

12. Lines 13–15, page 8: The authors have presented two results which lead to opposite conclusions. This needs to be addressed here directly instead of referring the reader to another publication. While the issue may have been more thoroughly assessed in Köhler et al. 2018, a brief comment should also be provided here so that the reader grasps the discordance in the author's results at a bare-minimum level without having to read up another paper.

A brief explanation of this result has been included in the revised manuscript.

13. Line 9, page 8: For the calculation of  $\epsilon_{\text{[LI]}}$  using equation 11 please provide the values of  $\Delta R_{\text{[CO2]}}$  and  $\Delta R_{\text{[LI]}}$  at LGM that were used.

The LGM values ( $\Delta R_{[CO2]} = -2.04 \text{ W m}^{-2}$  and  $\Delta R_{[LI]} = -3.88 \text{ W m}^{-2}$ ) are now provided. Upon including them and redoing the calculations, we realized we made a small mistake in the calculation of  $\epsilon$  in the former section 3.2, and the corresponding  $S^{\epsilon}_{[CO2,LI]}$ . This has been corrected in the revised manuscript. We thank the reviewer for letting us double-check our calculations.

14. Line 15, page 8: is the mean the value "of" years 20 and 22 kya or "between" those years?

The temporal resolution of this dataset is 2,000 years, so we have values for 20 and 22 kyr ago. In that sense, it is indeed the mean 'of' these times. This has now been clarified.

15. Line 16, page 8: "The specific paleo climate sensitivities we find here are generally higher than calculated by the old approach" But the new sensitivity calculated is 1.39 which is lower than that by the old approach which was 1.66.

The new sensitivity of 1.39 K W-1 m2 (revised to 1.45 K W-1 m2, see our answer to scientific comment #13) holds for the LGM, and should be compared to 0.93 K W-1 m2 obtained by the old approach. The PI sensitivity of 1.66 K W-1 m2 of the old approach should be compared to our new PI sensitivity of 2.45 K W-1 m2. This has now been clarified in the revised manuscript.

16. Line 12 page 9: "We correct the induced  $\Delta T_{[CO2]}$  of all individual models for this ratio" I don't follow.

17. Line 15 page 9: At this point I am lost. Why are you doing that regression? What it the motivation? And are you subtracting the global value  $\Delta T_{[CO2]}$  from  $\Delta T_{NH}$ ?

18. The ECS given in Table 1 for the CCSM4 model is different from that usually cited. Bitz et al. 2012 using the NCAR-CCSM4 and recently Chandan and Peltier, 2018 using a related UofT-CCSM4 have deduced the ECS to be 3.2. The value in Table 1 is lower than that. Where did the authors get this from? Haywood et al. 2013 also use CCSM4 ECS (from Bitz et al) of 3.2. Do the numbers for the other models need to be checked as well?

19. The authors should cite all the original experiment design papers for the PMIP3 experiments listed in Table 1. This can be done readily by adding a new column to the table called "References".

Answer to points 16 to 19: Section 3.3 has been removed from the manuscript, see our answer to the fourth general comment of the reviewer.

20. The figure description for Figure 3 is completely wrong. It is talking about things that are not on the figure.

Figure 3 and its caption have been corrected.

**Technical comments**

We are very grateful for these technical comments by the reviewer. We have implemented all the suggestions, except where indicated.

1. Line 2 page 1:"with to equilibrium"

This sentence has been rewritten completely.

2. Line 29 page 2: "are obtained from different various model setups"

This sentence has been rewritten completely.

3. Line 16 page 3: "In this case, the average global paleo temperature anomaly with respect to the pre-industrial (PI) average ( $\Delta T_g$ ) is"

4. Line 17 page 3: "that are typically neglected in the climate simulations".

5. Lines 3-4 on page 4 incorporating the phrase "the calculated paleoclimate sensitivity" in the current form refers to some specific and as yet undefined sensitivity. It's best to rephrase it as "If, for instance, only the most important slow feedback in the climate system, namely radiative forcing anomalies induced by albedo changes due to land ice (LI) variability are taken into account, then one can correct Sp to derive the following specific paleoclimate sensitivity." 6. The sentence on line 5, page 4, appears as a sharp interruption to the logic train before and after that sentence. It should instead be placed at the end of that paragraph and rephrased as "An overview A synthesis of different values estimates of S[CO2,LI] for from both ...."

7. Line 15 page 4: "e.g. because because, e.g."

8. Line 18 page 4: "through efficacy factors ( $\epsilon_{[LI]}$ ), which demands. This requires a reformulation"

9. Line 20 page 4: "to clearly distinguish them the sensitivities from the former ones those of the PALAEOSENS project, in which the radiative forcing of the different processes had identical weights were assigned identical efficacies."

This sentence has been rephrased to:

'This serves to clearly distinguish these newly-derived sensitivities from those of the PALAEOSENS project in which efficacy was not taken into account, implying that identical radiative forcing of different processes leads to identical temperature changes.'

10. Line 25 page 4: "by land ice changes ( $\varepsilon_{[LI]}$ ), using a slightly different definition than the following formulation which is based on, but modified from Hansen et al. (2005)" 11. Line 22 page 5: The sentence "CLIMBER-2 combined a 2.5 statistical-dynamical..." seems something is missing after 2.5. Did the authors mean "2.5 degree"?

Corrected to '... 2.5-dimensional ...'

12. Line 5 page 5: Add comma after "Similarly"

Corrected to 'As before, ...'

13. Line 3 page 5: "leaving 217 data points as indicated in Fig 1c,d."

14. Sentence beginning on line 18 page 7: change it to something like "For our first attempt at compensating paleoclimates sensitivity for slow processes other than  $CO_2$  changes we strive to deduce the same  $S^{\epsilon}_{[CO2,LI]}$ , inferred above, from experiment OIC in which both  $CO_2$  and land ice cover vary over time".

**This sentence has been rewritten as:**

'Now, we apply our approach to the results of experiment OIC, in which both  $CO_2$  and land ice cover vary over time, with the aim of deducing the sole effect of  $CO_2$  changes on global temperature.'

15. Line 23 page 7: "Between .... there are some outlying values caused by outliers resulted from division of by small numbers (not shown on Fig. 2b)."

16. Line 29 page 7: "....is more linear than that of between ..."

17. Line 31 page 7 "in the simulated domain through the entire 5 million year interval."

18. Line 11 page 8: "Similarly as before (Köhler et al., 2018), we detect Similar to Köhler et al., 2018, we too detect"

19. Line 15 page 8: the value of  $\Delta R_{[CO2]}$  should be -2.04

20. Line 15 page 8: "the LGM value (here taken here as the mean...)"

21. Line 21 page 8: "we first scale them it by a factor"

**This sentence has been removed.**

22. Line 23 page 8: "Note that this scaling still assumes unit efficacy for <del>all other</del> process other than land ice changes"

This sentence has been corrected as suggested by the reviewer, and replaced to the method section.

23. Line 24 page 8: "Then, after After multiplying by"

**This sentence has been removed.**

24. Line 24 page 8: Units should be Wm-2

This sentence has been removed.

25. Line 11 page 9: "that the ratio of the radiative forcing change  $AR_{[CO2]}$  between the LGM (185 ppm CO2) and the PI (280 ppm CO2), to the change between the PI and  $\frac{2 \times CO_2}{2}$  a 2 X PI case is

26. Line 16 page 9: "significant on at the 95% level"

Answer to points 25 and 26: Section 3.3 has been removed from the manuscript, see our answer to the fourth general comment of the reviewer.

27. Conclusions section, Lines 26, 28, 30:  $\varepsilon_{[CO2,LI]}$  is a new symbol not previously defined. It seems like a mistake and the authors likely meant  $\varepsilon_{[LI]}$

28. The yellow star in Fig 4 is barely visible against the cyan background. Please change it to something dark, maybe black.

All the colors in Fig. 4 have been changed for better visibility.

**Reviewer #2**

My initial thoughts on seeing this paper were very positive in the sense that, given the uncertainty of information in the paleorecord, and the difficulty of using state of the art models to make very long runs, all progress in the area of better defining climate sensitivity as it relates to past climates is worthwhile.

My optimism remained through the first parts of the paper, but by the end I have to admit that I am lost and really do not understand what the authors are trying to do and what they have discovered.

We thank the reviewer for considering our work. We are pleased the reviewer sees merit in the aim of our study. Along the helpful comments provided, we have thoroughly rewritten and restructured the manuscript to get our message across more clearly.

Most importantly we have improved the readability of the manuscript in the following manners:

- We have removed Section 3.3 from the manuscript, because this section only served as a further illustration of the importance of the effect of land ice changes that is already found in Sect. 3.2. As such, it is not essential to the main storyline of our manuscript.
- We have included a brief introduction to the method and results sections, in which we explain the aim of the section. The results sections end with a statement and discussion of the gained insights.
- We have split Sect. 2.1 (the method section) into three parts, describing 1) the PALAEOSENS approach used so far in earlier studies, 2) our main refinement: the inclusion of the efficacy of land ice changes, and 3) a small refinement that unifies the dependent variable in cross-plots of radiative forcing and global temperature anomalies.
- We have relocated Sects. 2.2.1 and 2.2.2, so that first the modelling results are introduced and analyzed straight away, and thereafter the proxy-inferred dataset is introduced and analyzed.
- We have renamed the variable 'CO2-equivalent temperature change' ( $\Delta T_{[CO2-equiv]}$ ). In the revised manuscript, we have more accurately named it 'the global temperature change (with respect to PI) stripped of the influence of land ice changes ( $\Delta T_{[-LI]}$ )'.

The authors introduce a variable, DTe[CO2 - equiv] but do not explain why this is useful or interesting.

The introduction of this variable, named  $\Delta T_{[-LI]}$  in the revised manuscript (see below), serves to unify the dependent variable in cross-plots of radiative forcing and global temperature anomalies. This makes our calculated  $S^{\epsilon}_{[CO2,LI]}$  more readily comparable to other specific paleoclimate sensitivities, where more and/or different long-term processes are considered. This step is a small refinement compared to our main refinement of including the efficacy of land ice changes. This has been clarified in the revised manuscript by splitting the method section into three parts.

What I would have done is take equation (9), replace X with LI and then explore all the elements of that equation. This would show us how S varies with DRLI and DRCO2, as well a DTg and one could consider how much of the state+forcing+efficacy dependence of S[CO2] is accounted for by considering land ice with and without considering efficacy.

This has been done in detail in Köhler et al. (2010) for the old approach (equivalent to  $\varepsilon_{[LI]} = 1$  in the refined approach). The inclusion of an efficacy factor for land ice changes does not qualitatively change this analysis, it just linearly amplifies (when  $\varepsilon_{[LI]} > 1$ ) or diminishes (when  $\varepsilon_{[LI]} < 1$ ) the effect of radiative forcing by land ice changes. We therefore focus directly on the effect of  $\varepsilon_{[LI]}$  on  $S^{\varepsilon}_{[CO2,LI]}$ .

I can see that Figures 2 and 3 represent some kind of sensitivity-like variable, but I cannot grasp its meaning.

Indeed, in Figures 2 and 3 we showed the main results of our manuscript: the influence of the deduced  $\epsilon_{Lul}$  on  $S^{\epsilon}_{[CO2,Lu]}$ .

Basically, DTe[CO2 - equiv] is not, as you suggest in equation 14 simply a function of DRCO2 but also depends on Tg and DRLI.

What we meant here is that we make a regression to the scattered data of  $\Delta R_{[CO2]}$  and the variable DTe[CO2 - equiv] (now called  $\Delta T_{[-LI]}$ ).  $\Delta T_{[-LI]}$  comprises the influences of  $\Delta R_{[LI]}$ ,  $\Delta R_{[CO2]}$  and  $\Delta T_g$ . To clarify this, in the revised manuscript we have named this function *regfunc* (instead of *g*).

I hope that the remedy is a better explanation of the reasons behind the derivations in section 1 and also better explanation of the insight that you gain from the results.

Other points.

P1L10 "Recently, it has been shown that simulations of models that have been integrated over a few centuries are not yet in equilibrium, and from longer climate simulations a higher ECS can be deduced (Knutti et al., 2017)."

This needs rephrasing. It has been well known since before dynamical oceans were included in climate models that the equilibrium time of the ocean is of the order of thousands of years. Since the invention of the AOGCM, ad-hoc methods have been introduced to try to estimate equilibrium climate sensitivity without running the models to equilibrium. What recent work has been doing is assessing the accuracy of such approximations.

We have removed this sentence from the manuscript, since it was not essential to - and therefore distracting from - the storyline.

P2L23 "likewise as several earlier studies" -> "as in several earlier studies"

This sentence has been removed.

Eq(10) This equation suggests to me that DTg-DT[LI]=DT[CO2]. Maybe I misunderstand, but it seems to me that DTg=DT[LI]+DT[CO2]+DT[X]+Z, where DT[X] is the influence of all the other forcings and Z represents cross terms (ie nonlinearities).

In this manuscript, we aim to calculate specific climate sensitivity  $S^{\epsilon}_{[CO2,LI]}$ , which only compensates paleoclimate sensitivity ( $S^{p}$ ) for the influence of land ice cover changes. To deduce the efficacy of land ice changes, we relate its effect on global temperature changes to that of all other processes combined. This is in line with our calculation of  $S^{a}$  from  $S^{\epsilon}_{[CO2,LI]}$  by multiplying by a factor of 0.64, which implies unit efficacy for all other processes than land ice changes. As stated in the text, this is a source of uncertainty to be investigated in future research.

P3L5 "Similarly as in the old approach," Not English

Corrected to: 'As before, ...'

Eq(13) Looks like a minus sign between "CO2" and "equiv".

We realize that calling this variable 'CO2-equivalent temperature change' ( $\Delta T_{[CO2-equiv]}$ ) was confusing. In the revised manuscript we have therefore more accurately named it 'the global temperature change (with respect to PI) stripped of the influence of land ice changes ( $\Delta T_{[-LI]}$ )'.

P5L12 "A functional relationship between TE[CO2–equiv] and R[CO2] (T[CO2–equiv]= g(R[CO2])) can be obtained by least squares regressions of higher-order polynomial to the scattered data of these variables."

It is not clear which variables are "these variables".

This sentence has been rephrased as:

'Now, we quantify  $S^{\epsilon}_{[CO2,LI]}$  by performing a least-squares regression (regfunc) through scattered data from  $\Delta T^{\epsilon}_{[-LI]}$  and  $\Delta R_{[CO2]}$ .'

Sections 2 and 3

I think the paper order should be 2.2.1, 3.1 then 2.2.2, 3.2. The way it is presented is just confusing. Present the whole of the simple modelling case and then move on to the databased case.

We have followed the suggestion of the reviewer.

P7L22 "and again fit a second order polynomial to the scattered data of T" [CO2–equiv]"

Which experiment?

Here, we analyse the results of experiment OIC. We have clarified this in the revised manuscript.

**P8L12 "Similarly as before" Not English**

**This sentence has been corrected.**

Table 1 State which paper each "published ECS" comes from.

Section 3.3 has been removed from the manuscript, see our answer to the comment of the reviewer below.

I prefer to write reviews before reading what other reviewers have posted, as I feel I will be too easily influenced, so I did not read the other reviewer's comment until now. I am encouraged to see that the other reviewer also found the paper very difficult to follow. This increases my optimism that there is hope that with better explanation in critical areas, and reorganisation to improve the storyline, that the paper may become both comprehensible and publishable.

Reviewer #1 was mostly concerned about Sect. 3.3. During the process of revising the manuscript, we have come to the realization that this section only served as a further illustration of the importance of the effect of land ice changes that is already found in Sect. 3.2. As such, it is not essential to the main storyline of our manuscript. To improve the clarity of the paper as a whole, we have therefore decided to remove it from the manuscript.

[revised manuscript text omitted]
_{[\rm CO_2,X]} = \frac{\Delta T_{\rm g}}{\Delta R_{[\rm CO_2]}} \left(1 - \frac{\Delta R_{[\rm X]}}{\Delta R_{[\rm CO_2]} + \Delta R_{[\rm X]}}\right) = \frac{\Delta T_{\rm g}}{\Delta R_{[\rm CO_2]} + \Delta R_{[\rm X]}} = \frac{\Delta T_{\rm g}}{\Delta R_{[\rm CO_2,X]}}.$$
(6)

If, for instance, only the most important slow feedback in the climate system, namely radiative forcing anomalies induced by albedo changes due to land ice (LI) variability are taken into account, then one can correct  $S^p$  to derive the following specific climate sensitivity:

$$S_{[\rm CO_2,LI]} = \frac{\Delta T_{\rm g}}{\Delta R_{[\rm CO_2]} + \Delta R_{\rm [LI]}} = \frac{\Delta T_{\rm g}}{\Delta R_{[\rm CO_2,LI]}}.$$
(7)

Using this approach, several studies performed a least-squares regression through scattered data from paleotemperature and 10 radiative forcing records (Martínez-Botí et al., 2015; Friedrich et al., 2016; Köhler et al., 2015, 2017b, 2018) relating  $\Delta T_{\rm g}$ to  $\Delta R_{\rm [CO_2,LI]}$  in a time-independent manner, from which  $S_{\rm [CO_2,LI]}$  could be determined. In this way, a state dependency of  $S_{\rm [CO_2,LI]}$  as function of background climate has been deduced for those data which are best approximated by a non-linear function. Furthermore, the quantification of  $S_{\rm [CO_2,LI]}$  for those state-dependent cases has been formalized in Köhler et al. (2017b). A synthesis of estimates of  $S_{\rm [CO_2,LI]}$  from both colder- and warmer-than-present climates has been compiled by von der Heydt et al. (2016).

To obtain  $S^a$ , one needs to multiply  $S_{[CO_2,LI]}$  by a factor of 0.64 that accounts for the influence of other long-term processes, namely vegetation, aerosol and non-CO2 greenhouse gas changes (PALAEOSENS Project Members, 2012). Finally, we obtain the equivalent ECS by multiplying  $S^a$  by 3.7 W m-2, the radiative forcing perturbation representing a CO2 doubling (Myhre et al., 1998).

**20 2.2 Refinement 1: Taking the efficacy of land ice changes into account**

5

The validity of the PALAEOSENS approach to calculate f is contingent on the notion that identical global-average radiative forcing changes leads to identical global temperature responses, regardless of the processes involved. However, it has been demonstrated that the horizontal and vertical distribution of the radiative forcing affects the resulting temperature response (e.g. Stuber et al., 2005; Hansen et al., 2005; Yoshimori et al., 2011; Stap et al., 2018) because, e.g. different fast feedbacks

25 are triggered depending on the location of the forcing. To address this issue, Hansen et al. (2005) introduced the concept of 'efficacy' factors, which we will explore further in this study. These factors ( $\varepsilon_{[X]}$ ) relate the strength of the temperature response to radiative forcing caused by a certain process X ( $\Delta T_{[X]}/\Delta R_{[X]}$ ), to a similar ratio caused by CO2 radiative forcing  $(\Delta T_{[CO_2]}/\Delta R_{[CO_2]})$ . This introduction of efficacy requires a reformulation of f as  $f_{\varepsilon}$ :

$$f_{\varepsilon} = \frac{\varepsilon_{[\mathrm{X}]} \Delta R_{[\mathrm{X}]}}{\Delta R_{[\mathrm{CO}_2]} + \varepsilon_{[\mathrm{X}]} \Delta R_{[\mathrm{X}]}} = 1 - \frac{\Delta R_{[\mathrm{CO}_2]}}{\Delta R_{[\mathrm{CO}_2]} + \varepsilon_{[\mathrm{X}]} \Delta R_{[\mathrm{X}]}},\tag{8}$$

and hence also of  $S_{[\mathrm{CO}_2,\mathrm{X}]}$  as  $S^{\varepsilon}_{[\mathrm{CO}_2,\mathrm{X}]}$ :

$$S_{[\rm CO_2,X]}^{\varepsilon} = \frac{\Delta T_{\rm g}}{\Delta R_{[\rm CO_2]} + \varepsilon_{[\rm X]} \Delta R_{[\rm X]}}.$$
(9)

In these reformulations, where in principal  $\varepsilon_{[X]}$  can take any value, we introduce the superscript  $\varepsilon$ . This serves to clearly distinguish these newly-derived sensitivities from those of the PALAEOSENS project in which efficacy was not taken into account, implying that identical radiative forcing of different processes leads to identical temperature changes.

To calculate  $S_{[CO_2,LI]}^{\varepsilon}$ , we constrain the efficacy factor for radiative forcing by land ice changes ( $\varepsilon_{[LI]}$ ), using the following formulation, which is based on, but slightly modified from Hansen et al. (2005):

$$\frac{\Delta T_{\rm [LI]}}{\Delta R_{\rm [LI]}} = \varepsilon_{\rm [LI]} \frac{\Delta T_{\rm g} - \Delta T_{\rm [LI]}}{\Delta R_{\rm [CO_2]}}.$$
(10)

This leads to:

5

20

$$\varepsilon_{\rm [LI]} = \frac{\omega}{1-\omega} \frac{\Delta R_{\rm [CO_2]}}{\Delta R_{\rm [LI]}},\tag{11}$$

where  $\omega$  represents the fractional relative influence of land ice changes on the global temperature change ( $\omega = \Delta T_{\text{[LI]}} / \Delta T_{\text{g}}$ ). If

10  $\varepsilon_{[LI]}$  is assumed to be constant in time (see Sect. 3.2 and 5), it can be calculated using Eq. 11 from data of any specific moment in time, and consequently applied to the whole record of  $\Delta R_{[CO_2]}$  and  $\Delta R_{[LI]}$  (Fig. 1a,c). As before, with this  $\varepsilon_{[LI]}$  a quantification of  $S^{\varepsilon}_{[CO_2,LI]}$  can be obtained by performing a least-squares regression through scattered data from paleotemperature and radiative forcing records, now relating  $\Delta T_g$  to  $(\Delta R_{[CO_2]} + \varepsilon_{[LI]} \Delta R_{[LI]})$  in a time-independent manner.

Note that apart from the formulation based on Hansen et al. (2005) followed here, other formulations of the efficacy factor 15 are possible. For instance, one can define an alternative efficacy factor ( $\varepsilon_{[LI],alt}$ ) such that it relates the effect of land ice changes on global temperature directly to the radiative forcing anomaly caused by CO2 changes, leading to:

$$S_{[\rm CO_2,X],\rm alt}^{\varepsilon} = \frac{\Delta T_{\rm g}}{\Delta R_{[\rm CO_2]} + \varepsilon_{[\rm LI],\rm alt} \Delta R_{[\rm CO_2]}}.$$
(12)

In this alternative case, the efficacy factor  $\varepsilon_{[LI],alt}$  relates to our original  $\varepsilon_{[LI]}$  as:

$$\varepsilon_{\rm [LI],alt} = \varepsilon_{\rm [LI]} \frac{\Delta R_{\rm [LI]}}{\Delta R_{\rm [CO_2]}}.$$
(13)

This implies that if  $\varepsilon_{[LI]}$  is indeed constant, any non-linearity in the relation between  $\Delta R_{[CO_2]}$  and  $\Delta R_{[LI]}$  would demand a more complex formulation of the alternative efficacy factor  $\varepsilon_{[LI],alt}$  (e.g. via a higher-order polynomial). Since we find such a non-linearity in our data (Fig. 2), using an F test to determine that a second order polynomial is a significantly (p value < 0.0001) better fit to the data than a linear function, we refrain from following this alternative formulation further.

5

**2.3 Refinement 2: Unifying the dependent variable**

In the cross-plots of radiative forcing and global temperature anomalies used to calculate  $S_{[CO_2,LI]}^{\varepsilon}$ , the radiative forcing on the x-axis is caused by a combination of CO2 and land-ice changes. To more readily compare  $S_{[CO_2,LI]}^{\varepsilon}$  to other specific paleoclimate sensitivities  $S_{[CO_2,X]}^{\varepsilon}$ , where more and/or different long-term processes are considered, the dependent variable has to be unified. Here, we therefore reformulate our equation to get  $\Delta R_{[CO_2]}$  in the nominator, enabling the use of cross-plots that now have  $\Delta R_{[CO_2]}$  on the x-axis.

 $S_{[\mathrm{CO}_2,\mathrm{X}]}^{\varepsilon} = \frac{\Delta T_{\mathrm{g}}}{\Delta R_{[\mathrm{CO}_2]} + \varepsilon_{[\mathrm{X}]} \Delta R_{[\mathrm{X}]}} = \frac{\Delta T_{\mathrm{g}}}{\Delta R_{[\mathrm{CO}_2]}} \frac{\Delta R_{[\mathrm{CO}_2]}}{\Delta R_{[\mathrm{CO}_2]} + \varepsilon_{[\mathrm{X}]} \Delta R_{[\mathrm{X}]}} = \frac{\Delta T_{[-\mathrm{X}]}^{\varepsilon}}{\Delta R_{[\mathrm{CO}_2]}}.$

Here,  $\Delta T_{[-X]}^{\varepsilon}$  is the global temperature change (with respect to PI) stripped of the inferred influence of processes X, defined as:

(14)

$$\Delta T_{[-X]}^{\varepsilon} := \Delta T_{g} \frac{\Delta R_{[CO_{2}]}}{\Delta R_{[CO_{2}]} + \varepsilon_{[X]} \Delta R_{[X]}}.$$
(15)

Hence, for the calculation of  $S^{\varepsilon}_{[CO_2,LI]}$  we use:

5

$$\Delta T_{[-\mathrm{LI}]}^{\varepsilon} := \Delta T_{\mathrm{g}} \frac{\Delta R_{[\mathrm{CO}_2]}}{\Delta R_{[\mathrm{CO}_2]} + \varepsilon_{[\mathrm{LI}]} \Delta R_{[\mathrm{LI}]}}.$$
(16)

10 Now, we quantify  $S^{\varepsilon}_{[CO_2,LI]}$  by performing a least-squares regression (regfunc) through scattered data from  $\Delta T^{\varepsilon}_{[-LI]}$  and  $\Delta R_{[CO_2]}$ . We use the precondition that no change in CO2 is related to no change in  $\Delta T^{\varepsilon}_{[-LI]}$ , meaning the regression intersects the y-axis at the origin ((x, y) = (0,0)). Following Köhler et al. (2017b), for any non-zero  $\Delta R_{[CO_2]}$ , we calculate  $S^{\varepsilon}_{[CO_2,LI]}$  as:

$$S_{[\text{CO}_2,\text{LI}]}^{\varepsilon} \bigg|_{\Delta R_{[\text{CO}_2]}} = \frac{\text{regfunc}}{\Delta R_{[\text{CO}_2]}} \bigg|_{\Delta R_{[\text{CO}_2]}}.$$
(17)

If  $\Delta R_{[CO_2]} = 0 \,\mathrm{W \, m^{-2}}$ , as is among others the case for pre-industrial conditions,  $S_{[CO_2, LI]}^{\varepsilon}$  is quantified as:

$$S_{[\text{CO}_2,\text{LI}]}^{\varepsilon} \bigg|_{\Delta R_{[\text{CO}_2]}=0} = \frac{\delta(\text{regfunc})}{\delta(\Delta R_{[\text{CO}_2]})} \bigg|_{\Delta R_{[\text{CO}_2]}=0}.$$
(18)

15 Equations 17 and 18 yield a quantification of  $S_{[CO_2,LI]}^{\varepsilon}$ , 
[revised manuscript text omitted]
_{\rm PA} = \Delta T_{\rm NH} / \Delta T_{\rm g}$ ) as follows: at the LGM,  $f_{\rm PA} = 2.7$  is taken from the average of PMIP3 model data (Braconnot et al., 2012), while at the mid-

[revised manuscript text omitted]

**5 Conclusions**

- 10 We have incorporated the concept of a constant efficacy factor (Hansen et al., 2005), that interrelates the global temperature responses to radiative forcing caused by land ice changes and  $CO_2$  changes, into our framework of calculating specific pale-oclimate sensitivity  $S^{\varepsilon}_{[CO_2,LI]}$ . The aim of this effort has been to overcome the problem that land ice and  $CO_2$  changes can lead to significantly different global temperature responses, even when they induce the same global-average radiative forcing. Firstly, we have shown the importance of considering efficacy differences by applying our new approach to results of 5-Myr
- 15 CLIMBER-2 simulations (Stap et al., 2018), where the separate effects of land ice changes and CO2 changes can be isolated. In the results of these simulations, the error from assuming the efficacy factor to be constant in time is negligible. Thereafter, we have used our new approach to reanalyse an 800-kyr proxy-inferred paleoclimate dataset (Köhler et al., 2015). We have inferred a range in the land ice change efficacy factor  $\varepsilon_{\text{[LI]}}$  from the 0.46±0.14 (mean ±1 $\sigma$ ) relative impact of land ice changes on the LGM temperature anomaly simulated by a 12-member climate model ensemble (Shakun, 2017). The thusly obtained
- 20 efficacy factor  $\varepsilon_{[LI]}$  is smaller than unity, implying that the impact on global temperature per unit of radiative forcing is less strong for land ice changes than for CO2 changes. Consequently, our derived PI  $S_{[CO_2,LI]}^{\varepsilon}$  of  $2.45_{-0.56}^{+0.53}$  KW-1 m2 is ~50% larger than the result of the old approach. The uncertainty in this estimate is only caused by the implemented range in  $\varepsilon_{[LI]}$ . The equivalent  $S^a$  and ECS corresponding to this  $S_{[CO_2,LI]}^{\varepsilon}$  are  $1.6_{-0.4}^{+0.3}$  KW-1 m2 and  $5.8 \pm 1.3$  K per CO2 doubling respectively.

Data availability. The CLIMBER-2 dataset is available at https://doi.pangaea.de/10.1594/PANGAEA.887427, and the proxy-inferred pale oclimate dataset is available at https://doi.pangaea.de/10.1594/PANGAEA.855449, from the PANGAEA database. For more information or data, please contact the authors.

**Appendix A: Influence of the polar amplification factor**

In the analysis performed in Sect. 4.2, we have used a global temperature record that was obtained from northern high-latitude temperature anomalies using a polar amplification factor  $f_{PA}$  that varies from 2.7 at the coldest to 1.6 at the warmest conditions

30 (Sect. 4.1). However, recent climate model simulations of the Pliocene using updated paleogeographic boundary conditions

show that in warmer times polar amplification could have been nearly the same as in colder times (Kamae et al., 2016; Chandan and Peltier, 2017). We therefore repeat the analysis using the same range in  $\varepsilon_{\text{[LI]}}$  and the same dataset, but with an applied constant  $f_{\text{PA}} = 2.7$  over the entire past 800 kyr to generate  $\Delta T_g$  ( $\Delta T_{g2}$  in Köhler et al. (2015)).

- The constant polar amplification used here counteracts increasing state dependency towards low temperatures, as the temper-5 ature differences are no longer amplified by changing polar amplification. Hence,  $S_{[CO_2,LI]}^{\varepsilon}$  is smaller at PI,  $1.96_{-0.44}^{+0.42}$  K W-1 m2 compared to  $2.45_{-0.56}^{+0.53}$  K W-1 m2 using the variable  $f_{PA}$ , but diminishes less strongly towards colder conditions (Fig. A1a cf. Fig. 4a). As before, the old approach (equivalent to the new approach using  $\varepsilon_{[LI]} = 1$ ), yields a lower PI  $S_{[CO_2,LI]}$  of 1.34 K W-1 m2 (Fig. A1b). The PI  $S_{[CO_2,LI]}^{\varepsilon}$  
[revised manuscript text omitted]

---

## Referee Report (RR1)

**Second review of Stap et al. 2018**

The revised manuscript is greatly improved from its original form with regards to the scientific material and with regards to the clarification of the author's methodology. I am providing my comments on the current version of the manuscript. Most of these are minor technical corrections/ language suggestions.

**Comments and suggested technical corrections**

1. The abstract does not include results on the findings of the efficacy factor. Please incorporate the efficacy factor estimate into the abstract.
2. In the beginning of the abstract the authors start by discussing the ECS from climate simulations and then they mention about correcting them to compare to paleodata. But I don't think this is what the paper is about; isn't the whole method of the paper to instead correct paleodata to compare it more directly to ECS from climate models?
3. Page 1 line 6: "This renders the prevailing approach"
4. Page 1 line 13: "does not consider differences in efficacy"
5. Page 2 line 4: "fast feedbacks e.g. through such as those involving changes to ……. changes.
6. Page 2 line 5: "climate models, as for instance"
7. Page line 10: The authors say the long term changes "are not taken into account in the quantification of ECS", but I think what they mean is that they "are not taken into account in the course of estimating ECS from proxy data."
8. Page 2 line 14: "global globally averaged"
9. Page 2 line 17: There are two things wrong with this sentence. Firstly, the "sole effect" of CO2 is not simply the ECS, it also includes the response of land ice and vegetation to the increased radiative forcing, but which operate on timescales belong the ECS. One wouldn't want to give the impression that those processed do not involve CO2. Secondly, the ECS is not the "sole effect" of CO2 also in the sense that it includes the primary effect from CO2 but also the secondary effect of feedbacks from short-term processes. I think it should say "…. the effect of CO2 changes as the accompanying short-term feedbacks, as described by the ECS can be estimated"
10. Sentence on page 2 spanning lines 27—29 is convoluted. Please rephrase it
11. Page 2 line 29: "Hence, In this manner, we can assess"
12. Page 3 line 4: "in this section, we recapitulate first summarize the approach to obtaining climate sensitivity from paleo data that has been used in"
13. Page 3 line 6: "We also then discuss the our main refinement we make in this study to that approach, ……. refinement that is meant to unify unifies the dependent variables …"
14. Page 3 lines 10-11: This is the third repeat of the definition of ECS (previously mentioned at the start of the Abstract and the Introduction). Please remove it.
15. Page 3 line 12: "such as those involving changes to ice sheet ….. changes are kept constant"
16. Page 3 line 14: "taking the ratio … over to ….. leads to yields"
17. Page 3 line 20: "perturbation record"

18. In the comment spanning pages 3 and 4 that discusses the use of the PALEOSENS members' approach to estimating climate sensitivity from paleodata, also mention the use of the approach to estimating sensitivity within modelling studies (e.g. the PALEOSENS paper itself and Chandan and Peltier, 2018, CP).
19. Page 4 lines 2—3: This is not the best way to phrase it. I like what the authors had in the original version of the manuscript. Please revert back to that version.
20. Page 4 Line 11: In this way the course of those studies"
21. The first paragraph on page 6 should be re-written as it is very raw in its current form
22. Page 6 line 5: "dependent variables have has to be unified. ..... nominator denominator"
23. Page 6 Lines 17-19 are a repeat of what was said in lines 16—18 on page 4. Please unify the underlying material into one appropriate location.
24. Page 7 Line 7: "In brief The simulation are could be forced by solar insolation which changes due to orbital (O) variations, and further by land ice (I) change..."
25. Page 4 Line 10 "reference experiment (OIC) all input data these factors are varied"
26. Page 8 Line 19: "cannot be directly obtained (e.g. from proxy-based datasets) and is hence a-priori unknown"
27. Page 8 Line 23: "The investigated dataset to be investigated contains ..... for the past 800 kyr. Although it the dataset covers the past 5 Myr, here we focus only on the ... constrained by high-fidelity ice core measurements of CO2 data within ice cores"
28. Page 8 Line 28: "The revised formulations of for $\Delta R_{CO_2}$ following from Etminan et al 2016 leads to "
29. Page 8 Line 31: "ice sheet model ANICE (de Boer et al. 2014) . ANICE was forced by .... to a reference PI climate. The temperature anomalies were obtained from a benthic"
30. Page 9 Line 6: "function of the NH temperature"
31. Removed the reference from line 7 on page 9, as its not needed in the context of that comment and it is already referred to twice in the preceding 6 lines.
32. Page 9 Line 9: "Therefore, these results are here considered to be more similar to those of from proxy-based reconstructions than of those from climate model-based simulations"
33. Page 9 Line 18: "Shakun (2017) compiled the simulated model based estimates of the relative impact ...... using a an ensemble of 12-member climate models ensemble and found a range of estimated w to be ..."
34. Page 9 Line 24: "CLIMBER-2 results (section 3). The difference may be related either to the fact"
35. Page 10 Line 14: "Finally, we have shown assessed the importance usefulness of considering"

---

## Author Response (AR2)

ALFRED-WEGENER-INSTITUT HELMHOLTZ-ZEÑTRUM FÜR POLAR-UND MEERESFORSCHUNG

Alfred Wegener Institute, PO Box 12 01 61, 27515 Bremerhaven, Germany

То

Dr. Daniel Kirk-Davidoff Handling Editor of Earth System Dynamics

9 May 2019 Subject: Resubmission of manuscript #esd-2018-88

Dear Dr. Kirk-Davidoff,

We wish to resubmit our manuscript entitled 'Including the efficacy of land ice changes in deriving climate sensitivity from paleodata', co-authored by L.B. Stap, P. Köhler and G. Lohmann, for consideration of publication in *Earth System Dynamics*. The manuscript represents a revision of our earlier submitted manuscript of the same name (MS No.: esd-2018-88) along the remaining reviewer comments.

Along the comments of reviewer #2, we have included more discussion on the various sources of uncertainty in our analysis. As you requested, we have given the conversion factor 0.64 a variable name ( $\phi$ ), and we have done the same for the factor 3.71 W m-2 to convert Sa to ECS ( $\Delta R_{2xCO_2}$ ). The effects of the uncertainties in these factors on Sa and ECS is now quantified in the manuscript. Furthermore, we have addressed all comments of both reviewers concerning clarity of the text and grammar.

We submit a color-coded revised manuscript, as well as a detailed point-bypoint response to the comments of the reviewers. If you have any further questions, please do not hesitate to contact us.

Yours sincerely,

Lennert Stap

Dr. Lennert B. Stap Postdoctoral researcher Bussestraße 24 27570 Bremerhaven +49(471)4831-1721 lennert.stap@awi.de

Alfred Wegener Institute Helmholtz Centre for Polar and Marine Research

BREMERHAVEN

Am Handelshafen 12 27570 Bremerhaven Germany Phone +49 471 4831-0 Fax +49 471 4831-1149 www.awi.de

Public law institution

Head Office: Am Handelshafen 12 27570 Bremerhaven Germany Phone +49 471 4831-0 Fax +49 471 4831-1149 www.awi.de

Board of Governors: MinDir Dr. Karl Eugen Huthmacher Board of Directors: Prof. Dr. Antje Boetius (Director) Dr. Karstén Wurr (Administrative Director) Dr. Uwe Nixdorf (Vice Director) Prof. Dr. Karen H. Wiltshire (Vice Director)

Bank account: Commerzbank AG, Bremerhaven BIC/Swift COBADEFF292 IBAN DE12292400240349192500 Tax-Id-No. DE 114707273

'Including the efficacy of land ice changes in deriving climate sensitivity from paleodata' by L.B. Stap, P. Köhler and G. Lohmann. Submitted for potential publication by Earth System Dynamics

**REPLY TO THE COMMENTS BY THE REVIEWERS**

Color coding: Black – comments by reviewers Green – reply by authors Purple – changes made to the manuscript (line numbers refer to the color-coded revised manuscript)

**Reviewer #1**

The revised manuscript is greatly improved from its original form with regards to the scientific material and with regards to the clarification of the author's methodology. I am providing my comments on the current version of the manuscript. Most of these are minor technical corrections/language suggestions.

We thank the reviewer for considering our manuscript a second time. We have implemented all the suggestions, except where indicated below.

**Comments and suggested technical corrections**

1. The abstract does not include results on the findings of the efficacy factor. Please incorporate the efficacy factor estimate into the abstract.

**Changed to:**

**Page 1, lines 11-13:**

We apply our refined approach to a proxy-inferred paleoclimate dataset, using  $\epsilon_{[LI]}=0.45^{+0.34}$ -0.20 based on a multi-model assemblage of simulated relative influences of land ice changes on the Last Glacial Maximum temperature anomaly.

2. In the beginning of the abstract the authors start by discussing the ECS from climate simulations and then they mention about correcting them to compare to paleodata. But I don't think this is what the paper is about; isn't the whole method of the paper to instead correct paleodata to compare it more directly to ECS from climate models?

We agree with the reviewer, and we have therefore rephrased this sentence to:

**Page 1, lines 6-7:**

Hence, climate sensitivity derived from paleodata has to be compensated for these processes, when comparing it to the ECS of climate models.

3. Page 1 line 6: "This renders the prevailing approach"

4. Page 1 line 13: "does not consider differences in efficacy"

**Changed to:**

**Page 1 line 15-16:**

Consequently, our obtained ECS estimate of 5.8  $\pm$  1.3K, where the uncertainty reflects the implemented range in  $\epsilon_{[LI]}$ , is ~50% higher than when differences in efficacy are not considered.

5. Page 2 line 4: "fast feedbacks e.g. through such as those involving changes to ...... changes.
6. Page 2 line 5: "climate models, as for instance"

7. Page line 10: The authors say the long term changes "are not taken into account in the quantification of ECS", but I think what they mean is that they "are not taken into account in the course of estimating ECS from proxy data."

**Changed to:**

**Page 2, lines 9-11:**

Among these are long-term processes (or slow feedbacks) such as changes in vegetation, dust, and, arguably most importantly, land ice changes, which are kept constant in the climate model runs used to calculate ECS.

**8. Page 2 line 14: "global globally averaged"**

9. Page 2 line 17: There are two things wrong with this sentence. Firstly, the "sole effect" of CO2 is not simply the ECS, it also includes the response of land ice and vegetation to the increased radiative forcing, but which operate on timescales belong the ECS. One wouldn't want to give the impression that those processed do not involve CO2. Secondly, the ECS is not the "sole effect" of CO2 also in the sense that it includes the primary effect from CO2 but also the secondary effect of feedbacks from short-term processes. I think it should say ".... the effect of CO2 changes as the accompanying short-term feedbacks, as described by the ECS can be estimated"

10. Sentence on page 2 spanning lines 27-29 is convoluted. Please rephrase it

**Changed to:**

**Page 2, lines 27-31 (similar at page 6 line 21 to page 7 line 2):**

We first illustrate our refined approach by applying it to transient simulations over the past 5 Myr using CLIMBER-2 (Stap et al., 2018), obtaining a quantification of the effect on global temperature of  $CO_2$  changes and the accompanying short-term feedbacks from a simulation forced by both land ice and  $CO_2$  changes. We compare this result to a simulation where  $CO_2$  changes are the only operating long-term process.

**11. Page 2 line 29: "Hence, In this manner, we can assess"**

12. Page 3 line 4: "in this section, we recapitulate first summarize the approach to obtaining climate sensitivity from paleo data that has been used in"

13. Page 3 line 6: "We also then discuss the our main refinement we make in this study to that approach, ...... refinement that is meant to unify unifies the dependent variables ..."

14. Page 3 lines 10-11: This is the third repeat of the definition of ECS (previously mentioned at the start of the Abstract and the Introduction). Please remove it.

15. Page 3 line 12: "such as those involving changes to ice sheet ..... changes are kept constant"
16. Page 3 line 14: "taking the ratio ... over to ..... leads to yields"

17. Page 3 line 20: "perturbation record"

18. In the comment spanning pages 3 and 4 that discusses the use of the PALEOSENS members' approach to estimating climate sensitivity from paleodata, also mention the use of the approach to estimating sensitivity within modelling studies (e.g. the PALEOSENS paper itself and Chandan and Peltier, 2018, CP).

**Changed to:**

**Page 3 line 24 – page 4 line 3:**

To obtain *f*, PALAEOSENS Project Members (2012) proposed an approach, which has subsequently been used in numerous studies aiming to constrain climate sensitivity from paleodata (e.g. von der Heydt et al., 2014; Martínez-Botí et al., 2015; Köhler et al., 2015, 2017b, 2018; Friedrich et al., 2016), and paleoclimate modelling studies (e.g. PALAEOSENS Project Members, 2012; Friedrich et al., 2016; Chandan and Peltier, 2018).

19. Page 4 lines 2—3: This is not the best way to phrase it. I like what the authors had in the original version of the manuscript. Please revert back to that version.

We have changed this sentence back to the original version.

**Page 4, lines 3-4:**

They suggested to quantify the influence of the long-term processes (X) by the radiative forcing change they induce ( $\Delta R_{[X]}$ ), relative to the total radiative forcing perturbation:

**20. Page 4 Line 11: In this way the course of those studies"**

21. The first paragraph on page 6 should be re-written as it is very raw in its current form

Changed to:

**Page 5 line 22 – Page 6 line 2:**

To calculate  $S^{\epsilon}_{[CO2,LI]}$ , previous studies have used cross-plots of global temperature anomalies and radiative forcing. The latter is caused by a combination of CO2 and land-ice changes, which is cumbersome if one wants to compare  $S^{\epsilon}_{[CO2,LI]}$ , to other specific paleoclimate sensitivities  $S^{\epsilon}_{[CO2,LI]}$ , where more and/or different long-term processes are considered. Here, we therefore reformulate our quantification of  $S^{\epsilon}_{[CO2,X]}$  to unify the dependent variable as  $\Delta R_{[CO2]}$ .

22. Page 6 line 5: "dependent variables have has to be unified. ..... nominator denominator"

Changed, see previous comment.

23. Page 6 Lines 17-19 are a repeat of what was said in lines 16—18 on page 4. Please unify the underlying material into one appropriate location.

We have removed the redundant information at lines 16-18 on page 4, and rephrased the paragraph at page 6:

**Page 6 lines 13-19:**

To obtain Sa, one needs to multiply SE[CO2,LI] by a conversion factor  $\varphi = 0.64 \pm 0.07$  (1 $\sigma$ uncertainty) that accounts for the influence of other long-term processes, namely vegetation, aerosol and non-CO2 greenhouse gas changes (PALAEOSENS Project Members, 2012). Note that this multiplication by  $\varphi$  ignores any possible state-dependencies in  $\varphi$  and assumes unit efficacy for processes other than land ice changes. Because a comprehensive analysis of the efficacy and state-dependency of these other processes is beyond the scope of this study, it is a source of uncertainty to be investigated in future research. Finally, we obtain the equivalent ECS by multiplying Sa by  $\Delta_{R_{2xCO2}} = 3.71 \pm 0.37$  Wm-2 (1 $\sigma$ -uncertainty), the radiative forcing perturbation representing a CO2 doubling (Myhre et al., 1998).

24. Page 7 Line 7: "In brief The simulation are could be forced by solar insolation which changes due to orbital (O) variations, and further by land ice (I) change..."

25. Page 4 Line 10 "reference experiment (OIC) all input data-these factors are varied"
26. Page 8 Line 19: "cannot be directly obtained (e.g. from proxy-based datasets) and is hence apriori unknown"

We specifically mean the proxy-based datasets here, hence we have changed this sentence to:

**Page 8, lines 21-23:**

Other than for climate model simulations, in proxy-based datasets the influence of land ice changes on global temperature perturbations cannot be directly obtained, and is hence a-priori unknown.

27. Page 8 Line 23: "The investigated dataset to be investigated contains ..... for the past 800 kyr. Although it the dataset covers the past 5 Myr, here we focus only on the ... constrained by high-fidelity ice core-measurements of CO2 data within ice cores"

28. Page 8 Line 28: "The revised formulations of for  $\Delta R_{CO2}$  following from Etminan et al 2016 leads to "

29. Page 8 Line 31: "ice sheet model ANICE (de Boer et al. 2014) . ANICE was forced by .... to a reference PI climate. The temperature anomalies were obtained from a benthic"

30. Page 9 Line 6: "function of the NH temperature"

31. Removed the reference from line 7 on page 9, as its not needed in the context of that comment and it is already referred to twice in the preceding 6 lines.

32. Page 9 Line 9: "Therefore, these results are here considered to be more similar to those of from proxy-based reconstructions than of those from climate model-based simulations"

Changed to (also because of a comment by reviewer #2):

**Page 9, lines 4-7:**

The ANICE results are here considered to be proxy-inferred, because, unlike climate models, ANICE is not constrained by climatic boundary conditions such as insolation and greenhouse gases. The temperature anomalies follow directly from a benthic  $\delta^{18}$ O stack (Lisiecki and Raymo, 2005) using an inverse technique.

33. Page 9 Line 18: "Shakun (2017) compiled the simulated model based estimates of the relative impact ..... using a an ensemble of 12-member-climate models ensemble and found a range of estimated w to be ..."

34. Page 9 Line 24: "CLIMBER-2 results (section 3). The difference may be related either to the fact"

35. Page 10 Line 14: "Finally, we have shown assessed the importance usefulness of considering"

**Reviewer #2**

The manuscript is sufficiently improved that I can follow what the authors are trying to do, and attempt a review.

Overall, within the practical constraints of the state-of-the-art, I think the work is reasonable enough, although some of the sources of uncertainty need better highlighting. I have a few comments of varying degrees of potential importance.

We thank the reviewer for a second consideration of our work. At the suggestion of the reviewer, we have included more discussion of uncertainty sources, as described below.

**p2.**

"Ceteris paribus" - the language of the journal is English and superfluous Latin should, in my opinion, be avoided. Not all readers have a first language that has a Latin base. I Googled the meaning, and do not understand why you don't use English here.

We have removed the whole clause containing this expression, because we felt it was redundant.

p3. "Repeat" not "recapitulate"

Changed to 'summarize' at the suggestion of reviewer #1.

**р4.**

I suspect that the 0.64 is highly uncertain and state dependent. It would be good if you gave a rough estimate of your uncertainty for this value.

The conversion factor (named  $\varphi$  in the revised manuscript) 0.64 is uncertain in three ways: 1) The PALEOSENS paper from which it is taken, gives a  $1\sigma$ -uncertainty of 0.07. We have incorporated the effect of this uncertainty in our estimate of Sa in the revised manuscript. 2) It ignores any possible state-dependency of this factor, which is now mentioned in the manuscript. 3) It does not take into account efficacy differences for long-term processes other than land ice changes. As we feel that a comprehensive analysis of the efficacy of other processes is beyond the scope of this work, we refer to this fact as a source of uncertainty to be investigated in future research.

We have furthermore included a brief quantification and discussion of the influence of the  $1\sigma$ uncertainty of 0.37 in the factor (now called  $\Delta R_{2xCO_2}$  in the revised manuscript) 3.71 Wm-2 to convert Sa to ECS.

**Page 6 lines 13-19:**

To obtain Sa, one needs to multiply S $\epsilon$ [CO2,LI] by a conversion factor  $\phi = 0.64 \pm 0.07$  (1 $\sigma$ uncertainty) that accounts for the influence of other long-term processes, namely vegetation, aerosol and non-CO2 greenhouse gas changes (PALAEOSENS Project Members, 2012). Note that this multiplication by  $\phi$  ignores any possible state-dependencies in  $\phi$  and assumes unit efficacy for processes other than land ice changes. Because a comprehensive analysis of the efficacy and state-dependency of these other processes is beyond the scope of this study, it is a source of uncertainty to be investigated in future research. Finally, we obtain the equivalent ECS by multiplying Sa by  $\Delta R_{2xCO_2} = 3.71 \pm 0.37$  Wm-2 (1 $\sigma$ -uncertainty), the radiative forcing perturbation representing a CO2 doubling (Myhre et al., 1998).

**Page 10, lines 5-8:**

Our inferred PI S $\epsilon$ [CO2,LI] is equivalent to an Sa of 1.6+0.3-0.4 KW-1m2, when considering only the uncertainty caused by the implemented range in  $\epsilon$ [LI], and to an Sa of 1.6+0.1-0.2 KW-1m2, when only considering the uncertainty in the conversion factor  $\phi$ . The equivalent ECS is 5.8 ± 1.3 K per CO2 doubling, when only considering the uncertainty caused by the implemented range in  $\epsilon$ [LI], and 5.8 ± 0.6 K per CO2 doubling, when only considering the uncertainty in the conversion factor  $\Delta R_{2xCO_2}$ .

**p5.**

On p4, you state that, to convert from S\_[LI,CO2] to S^a you should multiply by 0.64. Therefore DT\_G on in eqn 10 should also be multiplied by 0.64. This may solve all your problems in terms of reconciling CLIMBER and the data as, if I understand correctly, you need to include this multiplicative factor for the data based analyses, but not for the CLIMBER analysis.

The factor (now called  $\varphi$ ) to convert  $S^{\epsilon}_{[CO2,LI]}$  to  $S^{a}$  accounts for the long-term processes other than CO2 and ice-sheet changes. Indeed, this factor has to be included in the data-based analysis, but not in the CLIMBER-2 analysis as these long-term processes are ignored in CLIMBER-2. However, in Eq. 10 we relate the change in temperature caused by land ice-changes to the one caused by all other processes, aiming first to calculate  $S^{\epsilon}_{[CO2,LI]}$ . Hence, here  $\Delta T_{g}$  should not be multiplied by  $\varphi$ .

**Ρ5**

"Any specific moment", "in time" ... I do not think this is what you mean. Equation 1 relates to the steady state! This also causes me to ask what time averaging is used for the CLIMBER analysis on p7-8...? (Maybe this is already stated somewhere but I do not recall seeing this mentioned in the manuscript...?)

Here, we mean any time step in the record of  $\Delta R_{[CO2]}$  and  $\Delta R_{[LI]}$ , as is now indicated in the revised manuscript.

**Page 5, lines 9-10:**

If  $\epsilon_{[LI]}$  is assumed to be constant in time (see Sect. 3.2), it can be calculated using Eq. 11 from data of any time step in the record of  $\Delta R_{[CO2]}$  and  $\Delta R_{[LI]}$ , and consequently applied to the whole record (Fig. 1a,c).

The time averaging of the CLIMBER-2 results was already indicated in the manuscript to be 1,000 years.

*Page 7, line 19:*(...), after averaging to 1,000 year temporal resolution (Fig. 1a,b).

р6.

I don't think this is really a "validation" as such - more an "illustration" perhaps? Clearly CLIMBER is very simplified and other models may have quite different result. A sentence explaining why this information is not forthcoming from GCMs would be appropriate (ie you are forced to use a simplified model because of the CPU overhead).

This section is now referred to as an 'illustration' rather than a 'validation' of our approach throughout the text. We have furthermore included a sentence explaining why we use results of an intermediate complexity climate model rather than a GCM.

**Page 7, lines 5-7:**

Currently, long (~ $10^5$  to ~ $10^6$  years) integrations of state-of-the-art climate models, such as general circulation models and Earth system models, are not yet not feasible due to limited computer power. This gap can be filled by using models of reduced complexity (Claussen et al., 2002; Stap et al., 2017).

P7. "Note that, in the case S... is equal to... The fit further shows decreasing..." This seems silly, and confusing, especially since you then use it as a reason for referring in a confusing way to S\_CO2,LI later on. Just call it S^E\_[CO2]. It is surely clear to anyone who has managed to get this far through the paper what is meant.

We have deliberately called it  $S^{\epsilon}_{[CO2,LI]}$  because we compare it to another way of obtaining  $S^{\epsilon}_{[CO2,LI]}$  later on. This should keep people from thinking we are comparing apples to oranges.

Ρ8

Fig 3b looks to be less linear not more linear than 3a

Indeed, it should be less linear, enhancing  $S^{\epsilon}_{[CO2,LI]}$ . We have fixed this mistake in the revised manuscript.

**Page 8, lines 10-11:**

The relationship between  $\Delta T^{\epsilon}_{[-LI]}$  and  $\Delta R_{[CO2]}$  (Fig. 3b) is less linear than that between  $\Delta T_{[OC]}$  and  $\Delta R_{[CO2]}$  (Fig. 3a), hence the state dependency of  $S^{\epsilon}_{[CO2,LI]}$  is enhanced.

Again we have an "any moment in time". Rephrase please.

Changed to 'any time step of the record'.

The section "In principal [which should be "principle", but these days we rely on copy-editing to pick up this kind of thing] ... (Fig3c)." Should be moved to p7 where the LGM is first mentioned in this context.

**We have moved this part to page 7 line 32 to page 8 line 2.**

Please remove all the "old approach" / "new approach" labels. Give the approaches names, or reference the equations, or other papers. Otherwise people do not know what you are

talking about, and they will be even more lost in the future when there is a new new approach and your approach becomes the old approach!

We agree that this could cause confusion. Throughout the revised manuscript, we therefore refer to the 'old' method as the PALAEOSENS approach (that does not consider efficacy differences). The 'new' method is now referred to as 'our (refined) approach'.

ANICE is used to convert from temperature to radiation. After seeing some of the ice model inter-comparison results I suspect this is a major source of uncertainty. If you agree, it would be good to see this at least mentioned.

Please see our answer to the next comment.

р9

"Therefore, these results...similar to the of proxy data..." I do not know what you mean. Why?

We meant to say that the ANICE temperature results are more directly related to the ( $\delta^{18}$ O) proxy, because they are not constrained by climatic boundary conditions. We have now clarified this. Of course, using only a single model leads to model dependency of the results, which is now acknowledged in the revised manuscript.

**Page 9, lines 4-7:**

The ANICE results are here considered to be proxy-inferred, because, unlike climate models, ANICE is not constrained by climatic boundary conditions such as insolation and greenhouse gases. The temperature anomalies follow directly from a benthic  $\delta^{18}$ O stack (Lisiecki and Raymo, 2005) using an inverse technique. Nevertheless, the results are model-dependent and therefore subject to uncertainty.

P10 and throughout. The uncertainty ranges are quite strange because they exclude an awful lot of known uncertainty. Maybe you could be more careful about stating in the Abstract and Conclusion what these ranges include and do not include in terms of known uncertainties. Also, what are the quoted ranges - 1 standard deviation? 2,3? 95%??

Because, in this manuscript, we focus on the effect of the efficacy factor for land ice changes ( $\epsilon_{[LI]}$ ) on climate sensitivity, the uncertainties we provide in the abstract and conclusions are only caused by the implemented range in  $\epsilon_{[LI]}$ . This is mentioned in the manuscript.

**Page 1 line 15-16:**

Consequently, our obtained ECS estimate of 5.8  $\pm$  1.3K, where the uncertainty reflects the implemented range in  $\epsilon_{\text{[LI]}}$ , is ~50% higher than when differences in efficacy are not considered.

Page 11, lines 1-2:

The uncertainty in these estimates is only caused by the implemented range in  $\epsilon_{\mbox{\tiny [LI]}}.$

As described in our reply to a previous comment of the reviewer, we have furthermore included the effects of the uncertainty in the factors to convert  $S^{\epsilon}_{[CO2,LI]}$  to Sa and Sa to ECS (now called  $\phi$  and  $\Delta R_{2xCO_2}$  respectively) on our estimates of Sa and ECS. Additionally, we note that any possible state-dependency of  $\phi$  is ignored.

**Page 6 lines 13-19:**

To obtain Sa, one needs to multiply S $\epsilon$ [CO2,LI] by a conversion factor  $\phi = 0.64 \pm 0.07$  (1 $\sigma$ uncertainty) that accounts for the influence of other long-term processes, namely vegetation, aerosol and non-CO2 greenhouse gas changes (PALAEOSENS Project Members, 2012). Note that this multiplication by  $\phi$  ignores any possible state-dependencies in  $\phi$  and assumes unit efficacy for processes other than land ice changes. Because a comprehensive analysis of the efficacy and state-dependency of these other processes is beyond the scope of this study, it is a source of uncertainty to be investigated in future research. Finally, we obtain the equivalent ECS by multiplying Sa by  $\Delta R_{2xCO_2} = 3.71 \pm 0.37$  Wm-2 (1 $\sigma$ -uncertainty), the radiative forcing perturbation representing a CO2 doubling (Myhre et al., 1998).

**Page 10, lines 5-8:**

Our inferred PI S $\epsilon$ [CO2,LI] is equivalent to an Sa of 1.6+0.3-0.4 KW-1m2, when considering only the uncertainty caused by the implemented range in  $\epsilon$ [LI], and to an Sa of 1.6+0.1-0.2 KW-1m2, when only considering the uncertainty in the conversion factor  $\phi$ . The equivalent ECS is 5.8 ± 1.3 K per CO2 doubling, when only considering the uncertainty caused by the implemented range in  $\epsilon$ [LI], and 5.8 ± 0.6 K per CO2 doubling, when only considering the uncertainty in the conversion factor  $\Delta R_{2xCO_2}$ .

A comprehensive analysis of the effect of the uncertainties in the proxy-inferred records has already been performed by Köhler et al. (2015). We feel that repeating that exercise would distract from the primary focus of this paper, and therefore instead refer to this earlier study with respect to this issue.

**Page 10, lines 15-18:**

[revised manuscript text omitted]
_{[\rm CO_2,X]} = \frac{\Delta T_{\rm g}}{\Delta R_{[\rm CO_2]}} \left(1 - \frac{\Delta R_{[\rm X]}}{\Delta R_{[\rm CO_2]} + \Delta R_{[\rm X]}}\right) = \frac{\Delta T_{\rm g}}{\Delta R_{[\rm CO_2]} + \Delta R_{[\rm X]}} = \frac{\Delta T_{\rm g}}{\Delta R_{[\rm CO_2,X]}}.$$
(6)

If, for instance, only the most important slow feedback in the climate system, namely radiative forcing anomalies induced by albedo changes due to land ice (LI) variability are taken into account, then one can correct  $S^p$  to derive the following specific climate sensitivity:

$$S_{[\rm CO_2,LI]} = \frac{\Delta T_{\rm g}}{\Delta R_{[\rm CO_2]} + \Delta R_{[\rm LI]}} = \frac{\Delta T_{\rm g}}{\Delta R_{[\rm CO_2,LI]}}.$$
(7)

- Using this approach, several studies performed a least-squares regression through scattered data from paleotemperature and radiative forcing records (Martínez-Botí et al., 2015; Friedrich et al., 2016; Köhler et al., 2015, 2017b, 2018) relating ΔTg to ΔR[CO2,LI] in a time-independent manner, from which S[CO2,LI] could be determined. In the course of those studies, a state dependency of S[CO2,LI] as function of background climate has been deduced for those data which are best approximated by a non-linear function. Furthermore, the quantification of S[CO2,LI] for those state-dependent cases has been formalized in Köhler
  et al. (2017b). A synthesis of estimates of S[CO2,LI] from both colder- and warmer-than-present climates has been compiled by
- von der Heydt et al. (2016).

**2.2 Refinement 1: Taking the efficacy of land ice changes into account**

The validity of the PALAEOSENS approach to calculate f is contingent on the notion that identical global-average radiative forcing changes leads to identical global temperature responses, regardless of the processes involved. However, it has been

20 demonstrated that the horizontal and vertical distribution of the radiative forcing affects the resulting temperature response (e.g. Stuber et al., 2005; Hansen et al., 2005; Yoshimori et al., 2011; Stap et al., 2018) because, e.g. different fast feedbacks are triggered depending on the location of the forcing. To address this issue, Hansen et al. (2005) introduced the concept of 'efficacy' factors, which we will explore further in this study. These factors (ε[X]) relate the strength of the temperature response to radiative forcing caused by a certain process X (ΔT[X]/ΔR[X]), to a similar ratio caused by CO2 radiative forcing
25 (ΔT[CO2]/ΔR[CO2]). This introduction of efficacy requires a reformulation of f as fε:

$$f_{\varepsilon} = \frac{\varepsilon_{[\mathrm{X}]} \Delta R_{[\mathrm{X}]}}{\Delta R_{[\mathrm{CO}_2]} + \varepsilon_{[\mathrm{X}]} \Delta R_{[\mathrm{X}]}} = 1 - \frac{\Delta R_{[\mathrm{CO}_2]}}{\Delta R_{[\mathrm{CO}_2]} + \varepsilon_{[\mathrm{X}]} \Delta R_{[\mathrm{X}]}},\tag{8}$$

and hence also of  $S_{[CO_2,X]}$  as  $S_{[CO_2,X]}^{\varepsilon}$ :

$$S_{[\rm CO_2,X]}^{\varepsilon} = \frac{\Delta T_{\rm g}}{\Delta R_{[\rm CO_2]} + \varepsilon_{[\rm X]} \Delta R_{[\rm X]}}.$$
(9)

In these reformulations, where in principle  $\varepsilon_{[X]}$  can take any value, we introduce the superscript  $\varepsilon$ . This serves to clearly distinguish these newly-derived sensitivities from those of the PALAEOSENS approach in which efficacy was not taken into account, implying that identical radiative forcing of different processes leads to identical temperature changes.

5

To calculate  $S_{[CO_2,LI]}^{\varepsilon}$ , we constrain the efficacy factor for radiative forcing by land ice changes ( $\varepsilon_{[LI]}$ ), using the following formulation, which is based on, but slightly modified from Hansen et al. (2005):

$$\frac{\Delta T_{\rm [LI]}}{\Delta R_{\rm [LI]}} = \varepsilon_{\rm [LI]} \frac{\Delta T_{\rm g} - \Delta T_{\rm [LI]}}{\Delta R_{\rm [CO_2]}}.$$
(10)

This leads to:

$$\varepsilon_{[\mathrm{LI}]} = \frac{\omega}{1-\omega} \frac{\Delta R_{[\mathrm{CO}_2]}}{\Delta R_{[\mathrm{LI}]}},\tag{11}$$

where ω represents the fractional relative influence of land ice changes on the global temperature change (ω = ΔT[LI]/ΔTg). If ε[LI] is assumed to be constant in time (see Sect. 3.2), it can be calculated using Eq. 11 from data of any time step in the record
of ΔR[CO2] and ΔR[LI], and consequently applied to the whole record (Fig. 1a,c). As before, with this ε[LI] a quantification of Sε[CO2,LI] can be obtained by performing a least-squares regression through scattered data from paleotemperature and radiative forcing records, now relating ΔTg to (ΔR[CO2] + ε[LI] ΔR[LI]) in a time-independent manner.

Note that apart from the formulation based on Hansen et al. (2005) followed here, other formulations of the efficacy factor are possible. For instance, one can define an alternative efficacy factor ( $\varepsilon_{[LI],alt}$ ) such that it relates the effect of land ice changes on global temperature directly to the radiative forcing anomaly caused by CO2 changes, leading to:

$$S_{[\rm CO_2,X],alt}^{\varepsilon} = \frac{\Delta T_{\rm g}}{\Delta R_{[\rm CO_2]} + \varepsilon_{[\rm LI],alt} \Delta R_{[\rm CO_2]}}.$$
(12)

In this alternative case, the efficacy factor  $\varepsilon_{[LI],alt}$  relates to our original  $\varepsilon_{[LI]}$  as:

$$\varepsilon_{\rm [LI],alt} = \varepsilon_{\rm [LI]} \frac{\Delta R_{\rm [LI]}}{\Delta R_{\rm [CO_2]}}.$$
(13)

This implies that if  $\varepsilon_{[LI]}$  is indeed constant, any non-linearity in the relation between  $\Delta R_{[CO_2]}$  and  $\Delta R_{[LI]}$  would demand a more complex formulation of the alternative efficacy factor  $\varepsilon_{[LI],alt}$  (e.g. via a higher-order polynomial). Since we find such a non-linearity in our data (Fig. 2), using an F test to determine that a second order polynomial is a significantly (p value < 0.0001) better fit to the data than a linear function, we refrain from following this alternative formulation further.

20

15

**2.3 Refinement 2: Unifying the dependent variable**

To calculate  $S_{[CO_2,LI]}^{\varepsilon}$ , previous studies have used cross-plots of global temperature anomalies and radiative forcing. The latter is caused by a combination of CO2 and land-ice changes, which is cumbersome if one wants to compare  $S_{[CO_2,LI]}^{\varepsilon}$  to

other specific paleoclimate sensitivities  $S^{\varepsilon}_{[\text{CO}_2,\text{X}]}$ , where more and/or different long-term processes are considered. Here, we therefore reformulate our quantification of  $S^{\varepsilon}_{[\text{CO}_2,\text{LI}]}$  to unify the dependent variable as  $\Delta R_{[\text{CO}_2]}$ .

$$S_{[\text{CO}_2,\text{X}]}^{\varepsilon} = \frac{\Delta T_{\text{g}}}{\Delta R_{[\text{CO}_2]} + \varepsilon_{[\text{X}]} \Delta R_{[\text{X}]}} = \frac{\Delta T_{\text{g}}}{\Delta R_{[\text{CO}_2]}} \frac{\Delta R_{[\text{CO}_2]}}{\Delta R_{[\text{CO}_2]} + \varepsilon_{[\text{X}]} \Delta R_{[\text{X}]}} = \frac{\Delta T_{[-\text{X}]}^{\varepsilon}}{\Delta R_{[\text{CO}_2]}}.$$
(14)

Here,  $\Delta T_{[-X]}^{\varepsilon}$  is the global temperature change (with respect to PI) stripped of the inferred influence of processes X, defined as:

$$\Delta T_{[-X]}^{\varepsilon} := \Delta T_{g} \frac{\Delta R_{[CO_{2}]}}{\Delta R_{[CO_{2}]} + \varepsilon_{[X]} \Delta R_{[X]}}.$$
(15)

5 Hence, for the calculation of  $S^{\varepsilon}_{[CO_2,LI]}$  we use:

$$\Delta T_{[-\mathrm{LI}]}^{\varepsilon} := \Delta T_{\mathrm{g}} \frac{\Delta R_{[\mathrm{CO}_2]}}{\Delta R_{[\mathrm{CO}_2]} + \varepsilon_{[\mathrm{LI}]} \Delta R_{[\mathrm{LI}]}}.$$
(16)

Now, we quantify  $S^{\varepsilon}_{[\text{CO}_2,\text{LI}]}$  by performing a least-squares regression (regfunc) through scattered data from  $\Delta T^{\varepsilon}_{[-\text{LI}]}$  and  $\Delta R_{[\text{CO}_2]}$ . We use the precondition that no change in CO2 is related to no change in  $\Delta T^{\varepsilon}_{[-\text{LI}]}$ , meaning the regression intersects the y-axis at the origin ((x, y) = (0, 0)). Following Köhler et al. (2017b), for any non-zero  $\Delta R_{[\text{CO}_2]}$ , we calculate  $S^{\varepsilon}_{[\text{CO}_2,\text{LI}]}$  as:

$$S_{[\text{CO}_2,\text{LI}]}^{\varepsilon} \bigg|_{\Delta R_{[\text{CO}_2]}} = \frac{\text{regfunc}}{\Delta R_{[\text{CO}_2]}} \bigg|_{\Delta R_{[\text{CO}_2]}}.$$
(17)

10 If  $\Delta R_{[CO_2]} = 0 \text{ Wm}^{-2}$ , as is among others the case for pre-industrial conditions,  $S_{[CO_2, LI]}^{\varepsilon}$  is quantified as:

$$S_{[\text{CO}_2,\text{LI}]}^{\varepsilon} \bigg|_{\Delta R_{[\text{CO}_2]}=0} = \frac{\delta(\text{regfunc})}{\delta(\Delta R_{[\text{CO}_2]})} \bigg|_{\Delta R_{[\text{CO}_2]}=0}.$$
(18)

[revised manuscript text omitted]

25 ΔR[CO2] = -2.04 W m-2 and ΔR[LI] = -3.88 W m-2, yields ε[LI] = 0.45+0.34-0.20. Implementing this range for ε[LI] in Eq. 16, we calculate ΔTe[-LI] over the whole 800-kyr period. Fitting second order polynomials by least-squares regression to the scattered data of ΔTe[-LI] and ΔR[CO2], we infer a PI Se[CO2,LI] of 2.45+0.53-0.56 KW-1 m2 (Fig. 4a). The substantial uncertainty given here only reflects the 1σ uncertainty in ε[LI]. Similar to Köhler et al. (2018), we also detect a state dependency with decreasing Se[CO2,LI] towards colder climates for this dataset, more strongly so in case of lower ε[LI]. This state dependency is opposite to the one found in the CLIMBER-2 results (Sect. 3). The difference may be related either to the fact that fast climate feedbacks are too linear, or that some slow feedbacks are underestimated in intermediate complexity climate models like CLIMBER-2 (see Köhler et al., 2018, for a detailed discussion). At ΔR[CO2] = -2.04 W m-2, the LGM value, Se[CO2,LI] is only 1.45+0.33-0.37 K W-1 m2. The PALAEOSENS approach, which does not consider efficacy and is therefore equivalent to our approach using ε[LI] = 1, yields S[CO2,LI] = 1.66 K W-1 m2 for PI, and S[CO2,LI] = 0.93 K W-1 m2 for the LGM (Fig. 4b).

The specific paleoclimate sensitivities we find using the refined approach are hence generally larger than those obtained when neglecting efficacy differences. This is because, for the range of the impact of land ice changes on the LGM temperature anomaly implemented ( $\omega = 0.46 \pm 0.14$ ), the efficacy factor  $\varepsilon_{[LI]}$  is smaller than unity. In other words, these land ice changes contribute comparatively less per unit radiative forcing to the global temperature anomalies than the CO2 changes.

- 5 Our inferred PI  $S^{\varepsilon}_{[CO_2,LI]}$  is equivalent to an  $S^a$  of  $1.6^{+0.3}_{-0.4}$  KW-1m2, when only considering the uncertainty caused by the implemented range in  $\varepsilon_{[LI]}$ , and to an  $S^a$  of  $1.6^{+0.1}_{-0.2}$  KW-1m2, when only considering the uncertainty in the conversion factor  $\phi$ . The equivalent ECS is  $5.8 \pm 1.3$  K per CO2 doubling, when only considering the uncertainty caused by the implemented range in  $\varepsilon_{[LI]}$ , and  $5.8 \pm 0.6$  K per CO2 doubling, when only considering the uncertainty in the conversion factor  $\Delta R_{2xCO_2}$ . The ECS we find is thus on the high end of the results of other approaches to obtain ECS (Knutti et al., 2017), e.g. the 2.0 to
- 10 4.3 K 95%-confidence range from a large model ensemble (Goodwin et al., 2018), and the 2.2 to 3.4 K 66% confidence range from an emerging constraint from global temperature variability and CMIP5 (Cox et al., 2018). Hence, the low end of our ECS estimate is in the best agreement with these other estimates. This could mean that the influence the relative influence of land ice changes on the LGM temperature anomaly is on the high side, or possibly higher than, the  $0.46 \pm 0.14$  range we consider here. Alternatively, the conversion factor  $\phi = 0.64 \pm 0.07$  we use to convert  $S_{[CO_2,LI]}$  to  $S^a$  is an overestimation, which could
- 15 be caused by a larger-than-unity efficacy of long-term processes besides  $CO_2$  and land ice changes. We have focused primarily on the effect of  $\varepsilon_{[LI]}$  on  $S_{[CO_2,LI]}^{\varepsilon}$  in this analysis, and therefore we have for simplicity ignored uncertainties in the investigated proxy-inferred records themselves. A comprehensive description of these uncertainties and their influence on the calculated climate sensitivity can be found in Köhler et al. (2015).

**5 Conclusions**

- We have incorporated the concept of a constant efficacy factor (Hansen et al., 2005), that interrelates the global temperature responses to radiative forcing caused by land ice changes and CO2 changes, into our framework of calculating specific pale-oclimate sensitivity S€[CO2,LI]. The aim of this effort has been to overcome the problem that land ice and CO2 changes can lead to significantly different global temperature responses, even when they induce the same global-average radiative forcing. Firstly, we have assessed the usefulness of considering efficacy differences by applying our refined approach to results of 5-Myr CLIMBER-2 simulations (Stap et al., 2018), where the separate effects of land ice changes and CO2 changes can be
- isolated. In the results of these simulations, the error from assuming the efficacy factor to be constant in time is negligible. Thereafter, we have used our approach to reanalyse an 800-kyr proxy-inferred paleoclimate dataset (Köhler et al., 2015). We have inferred a range in the land ice change efficacy factor  $\varepsilon_{[LI]}$  from the relative impact of land ice changes on the LGM temperature anomaly simulated by a 12-member climate model ensemble (Shakun, 2017). The thusly obtained efficacy factor
- 30  $\varepsilon_{[LI]} = 0.45^{+0.34}_{-0.20}$  is smaller than unity, implying that the impact on global temperature per unit of radiative forcing is less strong for land ice changes than for CO2 changes. Consequently, our derived PI  $S^{\varepsilon}_{[CO_2,LI]}$  of  $2.45^{+0.53}_{-0.56}$  KW-1 m2 is ~50% larger than when efficacy differences are neglected. The equivalent  $S^a$  and ECS corresponding to this  $S^{\varepsilon}_{[CO_2,LI]}$  are  $1.6^{+0.3}_{-0.4}$  KW-1 m2

and  $5.8 \pm 1.3$  K per CO2 doubling respectively. The uncertainty in these estimates is only caused by the implemented range in  $\varepsilon_{\text{[LI]}}$ .

*Data availability.* The CLIMBER-2 dataset is available at https://doi.pangaea.de/10.1594/PANGAEA.887427, and the proxy-inferred paleoclimate dataset is available at https://doi.pangaea.de/10.1594/PANGAEA.855449, from the PANGAEA database. For more information or data, please contact the authors.

**Appendix A: Influence of the polar amplification factor**

5

10

15

In the analysis performed in Sect. 4.2, we have used a global temperature record that was obtained from northern high-latitude temperature anomalies using a polar amplification factor  $f_{\rm PA}$  that varies from 2.7 at the coldest to 1.6 at the warmest conditions (Sect. 4.1). However, recent climate model simulations of the Pliocene using updated paleogeographic boundary conditions show that in warmer times polar amplification could have been nearly the same as in colder times (Kamae et al., 2016; Chandan and Peltier, 2017). We therefore repeat the analysis using the same range in  $\varepsilon_{\rm [LI]}$  and the same dataset, but with an applied constant  $f_{\rm PA} = 2.7$  over the entire past 800 kyr to generate  $\Delta T_q$  ( $\Delta T_{q2}$  in Köhler et al. (2015)).

The constant polar amplification used here counteracts increasing state dependency towards low temperatures, as the temperature differences are no longer amplified by changing polar amplification. Hence,  $S_{[CO_2,LI]}^{\varepsilon}$  is smaller at PI,  $1.96_{-0.44}^{+0.42}$  KW-1 m2 compared to  $2.45_{-0.56}^{+0.53}$  KW-1 m2 using the variable  $f_{
[revised manuscript text omitted]